# Single-cell, whole-embryo phenotyping of mammalian developmental disorders

Xingfan Huang[1,2,21], Jana Henck[3,4,21], Chengxiang Qiu[1,21], Varun K. A. Sreenivasan[3], Saranya Balachandran[3], Oana V. Amarie[5], Martin Hrabě de Angelis[5,6,7], Rose Yinghan Behncke[8,9], Wing-Lee Chan[8,9], Alexandra Despang[4,9], Diane E. Dickel[10], Madeleine Duran[1], Annette Feuchtinger[11], Helmut Fuchs[5], Valerie Gailus-Durner[5], Natja Haag[12], Rene Hägerling[4,8,9], Nils Hansmeier[4,8,9], Friederike Hennig[4], Cooper Marshall[1,13], Sudha Rajderkar[10], Alessa Ringel[4,8], Michael Robson[4], Lauren M. Saunders[1], Patricia da Silva-Buttkus[5], Nadine Spielmann[5], Sanjay R. Srivatsan[1], Sascha Ulferts[8,9], Lars Wittler[4], Yiwen Zhu[7], Vera M. Kalscheuer[4], Daniel M. Ibrahim[4,9], Ingo Kurth[12], Uwe Kornak[14], Axel Visel[10], Len A. Pennacchio[10], David R. Beier[13,15,16], Cole Trapnell[1,13,17], Junyue Cao[18✉], Jay Shendure[1,13,17,19✉] & Malte Spielmann[3,4,20✉]

Mouse models are a critical tool for studying human diseases, particularly developmental disorders[1]. However, conventional approaches for phenotyping may fail to detect subtle defects throughout the developing mouse[2]. Here we set out to establish single-cell RNA sequencing of the whole embryo as a scalable platform for the systematic phenotyping of mouse genetic models. We applied combinatorial indexing-based single-cell RNA sequencing[3] to profile 101 embryos of 22 mutant and 4 wild-type genotypes at embryonic day 13.5, altogether profiling more than 1.6 million nuclei. The 22 mutants represent a range of anticipated phenotypic severities, from established multisystem disorders to deletions of individual regulatory regions[4,5]. We developed and applied several analytical frameworks for detecting differences in composition and/or gene expression across 52 cell types or trajectories. Some mutants exhibit changes in dozens of trajectories whereas others exhibit changes in only a few cell types. We also identify differences between widely used wild-type strains, compare phenotyping of gain- versus loss-of-function mutants and characterize deletions of topological associating domain boundaries. Notably, some changes are shared among mutants, suggesting that developmental pleiotropy might be 'decomposable' through further scaling of this approach. Overall, our findings show how single-cell profiling of whole embryos can enable the systematic molecular and cellular phenotypic characterization of mouse mutants with unprecedented breadth and resolution.

For more than 100 years, the laboratory mouse (*Mus musculus*) has served as the quintessential animal model for studying human diseases[1]. For developmental disorders in particular, mice have been transformative, as a mammalian system that is nearly ideal for genetic analysis and in which the embryo is readily accessible[6].

At its inception, mouse genetics relied on spontaneous or induced mutations resulting in visible physical defects that could then be mapped. However, gene-targeting techniques later paved the way for 'reverse genetics' (that is, analysing the phenotypic effects of intentionally engineered mutations)[6]. Through systematic efforts such as the International Knockout Mouse Consortium, knockout (KO) models are now available for thousands of genes[7]. Furthermore, with genome editing[8,9], it is increasingly practical to delete individual regulatory elements[10].

Phenotyping has also grown more sophisticated. Conventional investigations of developmental syndromes typically focus on one organ at a

[1]Department of Genome Sciences, University of Washington, Seattle, WA, USA. [2]Paul G. Allen School of Computer Science & Engineering, University of Washington, Seattle, WA, USA. [3]Institute of Human Genetics, University Medical Center Schleswig-Holstein, University of Lübeck & Kiel University, Lübeck, Germany. [4]Max Planck Institute for Molecular Genetics, Berlin, Germany. [5]Institute of Experimental Genetics, German Mouse Clinic, Helmholtz Zentrum München, German Research Center for Environmental Health (GmbH), Neuherberg, Germany. [6]Chair of Experimental Genetics, TUM School of Life Sciences, Technische Universität München, Freising, Germany. [7]German Center for Diabetes Research (DZD), Neuherberg, Germany. [8]Institute of Medical Genetics and Human Genetics of the Charité, Berlin, Germany. [9]Berlin Institute of Health at Charité – Universitätsmedizin Berlin, BCRT, Berlin, Germany. [10]Lawrence Berkeley National Laboratory, Berkeley, CA, USA. [11]Core Facility Pathology & Tissue Analytics, Helmholtz Zentrum München, German Research Center for Environmental Health (GmbH), Neuherberg, Germany. [12]Institute for Human Genetics and Genomic Medicine, Medical Faculty, RWTH Aachen University, Aachen, Germany. [13]Brotman Baty Institute for Precision Medicine, University of Washington, Seattle, WA, USA. [14]Institute of Human Genetics, University Medical Center Göttingen, Göttingen, Germany. [15]Center for Developmental Biology & Regenerative Medicine, Seattle Children's Research Institute, Seattle, WA, USA. [16]Department of Pediatrics, University of Washington, Seattle, WA, USA. [17]Allen Discovery Center for Cell Lineage Tracing, Seattle, WA, USA. [18]Laboratory of Single-Cell Genomics and Population Dynamics, The Rockefeller University, New York, NY, USA. [19]Howard Hughes Medical Institute, Seattle, WA, USA. [20]DZHK (German Centre for Cardiovascular Research), partner site Hamburg/Lübeck/Kiel, Lübeck, Germany. [21]These authors contributed equally: Xingfan Huang, Jana Henck, Chengxiang Qiu. ✉e-mail: jcao@rockefeller.edu; shendure@uw.edu; malte.spielmann@uksh.de

specific stage (for example, combining expression analyses, histology and imaging to investigate a visible malformation)[1]. The Mouse Clinic, involving a battery of standardized tests, reflects a more systematic approach[11], but phenotypes detected through such tests (for example, behavioural and electrophysiological) may require years of additional work to link them to molecular and cellular correlates. Furthermore, it is often the case that an intentionally engineered mutation results in no detectable abnormality[12]. In such instances, it remains unknown whether there is truly no phenotype, or whether the methods used are simply insufficiently sensitive. In sum, phenotyping has become 'rate limiting' in mouse genetics.

Single-cell molecular profiling offers a potential path to overcome such barriers. As a first step, we and others have applied single-cell RNA sequencing (scRNA-seq) to profile wild-type mouse development at the scale of the whole embryo[3,13–18]. Applying scRNA-seq to mouse mutants, several groups have successfully unravelled how specific mutations affect transcriptional networks and lead to altered cell fate decisions in individual organs[19]. However, there is still no clear framework for analysing such data at the whole-embryo scale.

## scRNA-seq of 101 mouse embryos

We set out to establish whole-embryo scRNA-seq as a scalable framework for the systematic molecular and cellular phenotyping of mouse genetic models. We collected 103 mouse embryos, including 22 different mutants and 4 wild-type strains at embryonic day (E)13.5, and generally 4 replicates per strain (Fig. 1a). Mutants were chosen to represent a spectrum of phenotypic severity ranging from established pleiotropic disorders to KOs of individual regulatory elements.

We grouped mutants into four categories (Fig. 1a and Supplementary Table 1). The first category, pleiotropic mutants, consisted of embryos with KOs of developmental genes expressed in several organs (*Ttc21b* KO, *Carm1* KO and *Gli2* KO), and two mutations of the *Sox9* regulatory landscape suspected to have pleiotropic effects (*Sox9* topological associating domain (TAD) boundary knock-in; *Sox9* regulatory inversion (INV))[5,20–22]. The second category, developmental disorder mutants, consisted of embryos intended to model specific human diseases (*Scn11a* gain of function (GOF), *Ror2* knock-in, *Gorab* KO and *Cdkl5* −/Y (hemizygous))[23–25]. The third category consisted of embryos with mutations of loci associated with human disease (*Scn10a/Scn11a* double KO, *Atp6v0a2* KO, *Atp6v0a2^R755Q* and *Fat1* TAD KO)[26,27]. The fourth category consisted of embryos with prospective deletions of *cis*-regulatory elements, including of TAD boundaries near developmental transcription factors (*Smad3*, *Tbx5*, *Neurog2*, *Sim1*, *Smad7*, *Dmrt1*, *Tbx3* and *Twist1*)[4]. As a positive control, this fourth category includes a ZRS distal enhancer (zone of polarizing activity regulatory sequence) KO mutant, which specifically fails to develop distal limb structures[28]. Except for *Scn11a* GOF, all mutants were homozygous.

To validate staging, we leveraged our previous mouse organogenesis cell atlas (MOCA), which spans E9.5 to E13.5 (ref. 3). After doublet filtering, we profiled 1,671,245 nuclei (16,226 ± 9,289 per embryo; 64,279 ± 18,530 per strain; median unique molecular identifier count: 843 per cell; median genes detected: 534; 75% duplication rate). Below we refer to this dataset as the mouse mutant cell atlas (MMCA).

Applying principal component analysis (PCA) to 'pseudobulk' profiles of the embryos resulted in two groups corresponding to genetic background (Fig. 1b), with FVB embryos clustering separately from C57BL/6J, G4 and BALB/c embryos. However, embryos corresponding to individual mutants did not cluster separately, suggesting that none was affected with severe, global aberrations. A single outlier (embryo 104) was aberrant with respect to cell recovery (*n* = 1,047) and appearance (Extended Data Fig. 1a).

To validate staging, we leveraged our previous mouse organogenesis cell atlas (MOCA), which spans E9.5 to E13.5 (ref. 3). PCA of pseudobulk profiles of 61 wild-type embryos from MOCA resulted in a first component (principal component 1 (PC1)) strongly correlated with developmental age (Fig. 1c). Projecting pseudobulk profiles of the 103 MMCA embryos to this embedding resulted in most MMCA embryos clustering with E13.5 MOCA embryos along PC1, consistent with accurate staging. However, five MMCA embryos seemed closer to E11.5 or E12.5 MOCA embryos. Four of these were retained as their delay might be explained by their mutant genotype, whereas one wild-type embryo (C57BL/6; embryo 41) was designated a second outlier. We removed cells from the two outlier embryos (embryos 104 and 41) as well as cells with high proportions of reads mapping to the mitochondrial genome (>10%) or ribosomal genes (>5%). This left 1,627,857 cells, derived from 101 embryos (Fig. 1d).

To facilitate data integration, we projected cells from all genotypes to a wild-type-derived 'reference embedding' (Methods and Extended Data Fig. 1b,c). Altogether, we identified 13 major trajectories, 8 of which were further stratified into 59 sub-trajectories (Fig. 1e, Extended Data Fig. 2a and Supplementary Table 2), generally covering the expected cell trajectories at this stage of development. These were also generally consistent with our annotations of MOCA, albeit with some corrections as described elsewhere[17,29]. Greater granularity for some cell types is probably a consequence of the deeper sampling of E13.5 cells in these new data (Extended Data Fig. 2b).

## Mutant-specific variation in cell-type composition

In analysing these data, we pursued three approaches: quantification of gross differences in cell-type composition (this section); investigation of more subtle differences in the distribution of cell states within annotated trajectories and sub-trajectories; and analysis of the extent to which phenotypic features are shared between mutants.

To systematically assess cell-type compositional differences, we first examined the proportions of cells assigned to each of 13 major trajectories. These proportions were mostly consistent across genotypes (Extended Data Fig. 3a), but some mutants exhibited substantial differences. For example, compared to wild-type C57BL/6, the proportion of cells in the neural tube trajectory decreased from 37.3% to 33.7% and 32.6% in the *Gli2*-KO and *Ttc21b*-KO strains, respectively, and the proportion of cells in the mesenchymal trajectory decreased from 44.1% to 37.1% in the *Gorab*-KO strain. These changes are broadly consistent with the gross phenotypes associated with these mutations[20,25,30], but are subject to the caveat of substantial interindividual heterogeneity (Extended Data Fig. 3b). We also observe differences in major-trajectory composition between the four wild-type strains. For example, wild-type FVB and G4 mice consistently had fewer mesenchymal and more neural tube cells than wild-type BALB/c and C57BL/6 embryos (Extended Data Fig. 3c). We further checked for technical effects (for example, experimental batch) that might confound cell-type recovery rates (Extended Data Fig. 4a–c).

We next sought to investigate compositional differences at the level of sub-trajectories. For each combination of background and sub-trajectory, we carried out regression to identify mutations that were nominally predictive of the proportion of cells falling in that sub-trajectory (uncorrected *P* value < 0.05; beta-binomial regression; Methods). Across 22 mutants, this analysis highlighted 300 nominally significant changes (Fig. 2a and Supplementary Table 3). Owing to the limited number of replicate embryos per strain, our power to definitively call such changes is limited, particularly in the smaller trajectories (Methods and Extended Data Fig. 4d). Nevertheless, two patterns are noteworthy, as follows.

First, *Atp6v0a2* KO and *Atp6v0a2^R755Q*, distinct mutants of the same gene[26], exhibit highly consistent patterns of change, both with respect to which sub-trajectories are nominally significant as well as the direction and magnitude of changes. The consistency supports the validity of this analytical approach.

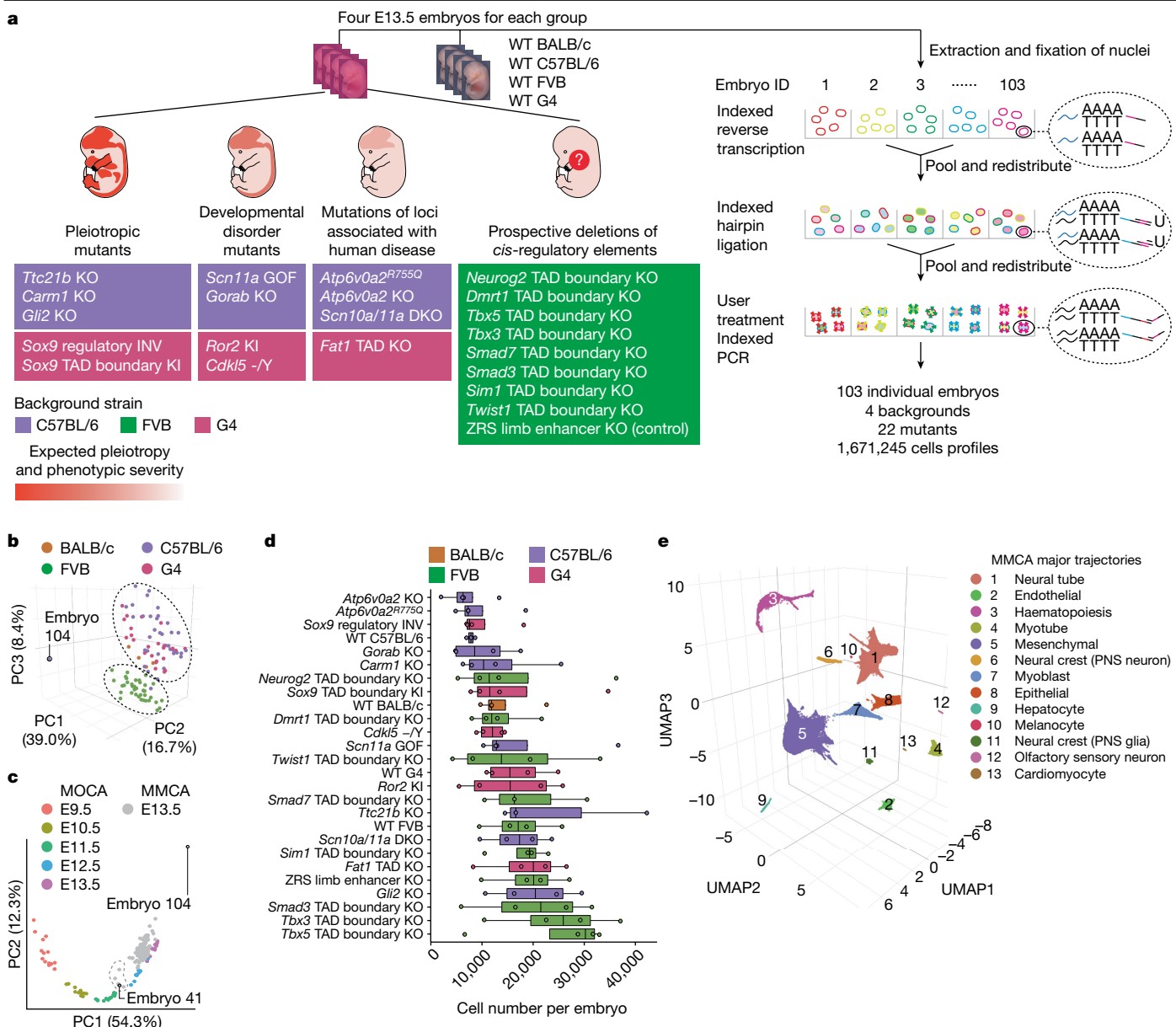

**Fig. 1 | Single-cell transcriptional profiling of 103 whole mouse embryos staged at E13.5. a**, Categories of mutants (left) analysed by whole-embryo profiling with sci-RNA-seq3 (right). WT, wild type. KI, knock-in; DKO, double KO. −/Y, hemizygous **b**, Embeddings of pseudobulk RNA-seq profiles of MMCA embryos in PCA space with visualization of the top three PCs. Datapoints are coloured by background strain of the embryo. The black dashed ovals highlight two major groups corresponding to FVB versus other backgrounds. Embryo 104 is a clear outlier. **c**, Embeddings of pseudobulk RNA-seq profiles of MOCA[3] and MMCA embryos in PCA space defined solely by MOCA, with MMCA embryos (grey) projected onto it. The top two PCs are visualized. Coloured points correspond to MOCA embryos of different stages (E9.5–E13.5), and grey

points to MMCA embryos (E13.5). The dashed line highlights five MMCA embryos that are co-localized with E11.5 or E12.5 embryos from MOCA. Three are from *Scn11a*-GOF (embryos 33, 34 and 36), *Carm1*-KO (embryo 101) and wild-type (embryo 41) C57BL/6 strains. **d**, Number of cells profiled per embryo for each strain. Centre lines show medians; box limits indicate 25th and 75th percentiles; replicates (*n* = 3 for wild-type C57BL/6, *n* = 4 for all others) are represented by dots. Genotypes are listed by median cell number in ascending order. **e**, Three-dimensional UMAP visualization of wild-type subset of MMCA dataset (215,575 cells from 15 embryos). Cells are coloured by major-trajectory annotation. PNS, peripheral nervous system.

Second, mutants varied considerably with respect to the number of sub-trajectories nominally significant for compositional differences. At the higher extreme, 30 of 54 sub-trajectories were nominally altered by the *Sox9* regulatory INV mutation, consistent with the wide-ranging roles of SOX9 in development[31,32]. At the lower extreme, TAD boundary KOs exhibited very few changes, consistent with the paucity of gross phenotypes in such mutants[12]. Nonetheless, all TAD boundary KOs did show some nominal changes, including specific ones (for example, the lung epithelial and liver hepatocyte trajectories

were specifically decreased in *Dmrt1* and *Tbx3* TAD boundary KOs, respectively).

There were a few extreme examples (for example, in which a sub-trajectory seemed to be fully lost). For example, *Ttc21b*, which encodes a cilial protein and whose KO is associated with brain, bone and eye phenotypes[20,33], exhibited a marked reduction in retinal neuron ($\log_2$[ratio] = −7.16; Fig. 2b), lens ($\log_2$[ratio] = −2.40) and retina epithelium ($\log_2$[ratio] = −1.65) trajectories (Extended Data Fig. 5a–c). Validations through haematoxylin and eosin staining

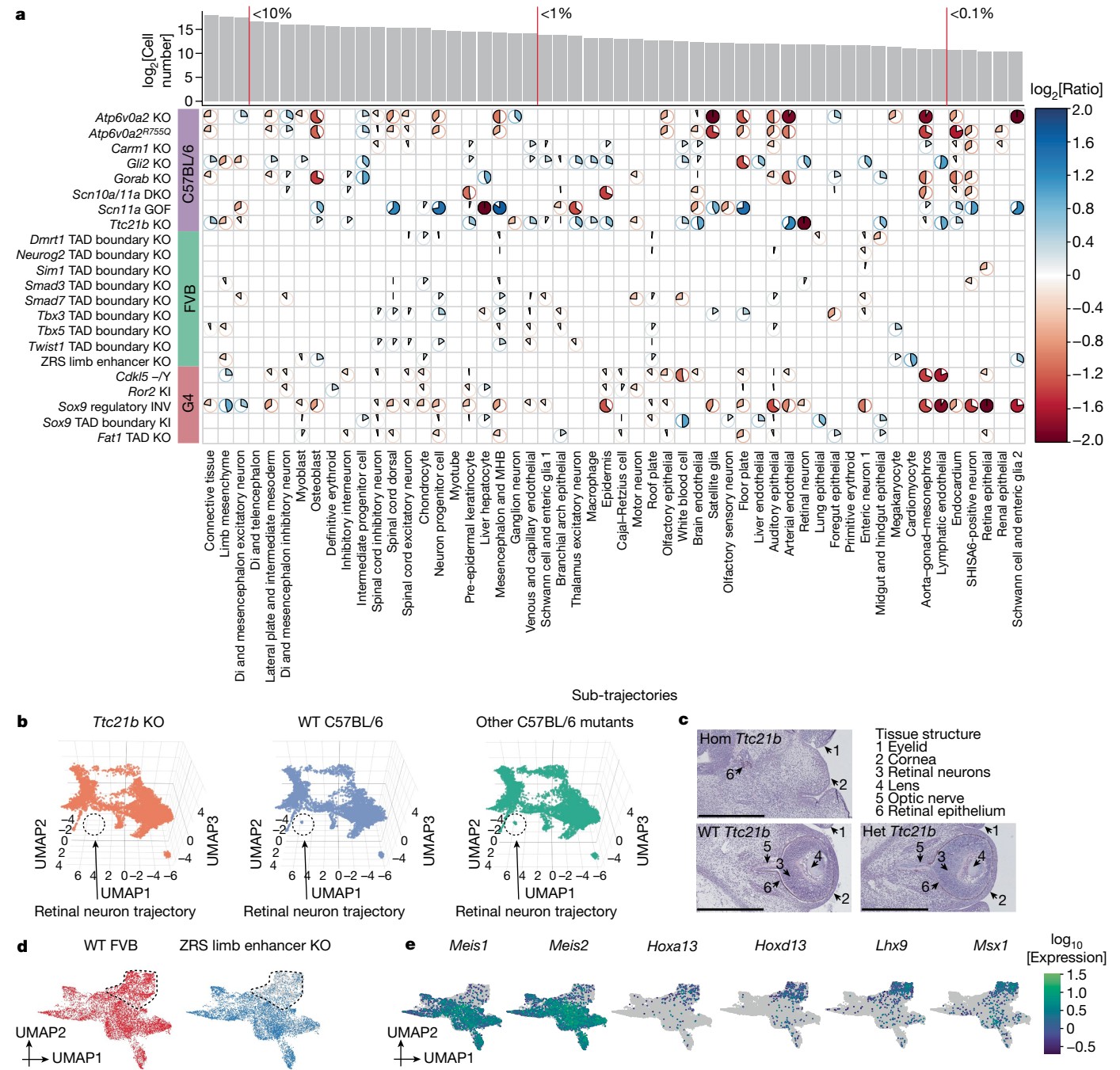

**Fig. 2 | Cell composition changes for individual mutants across developmental trajectories. a**, The heatmap shows log₂-transformed ratios of cell proportions between each mutant type (*y* axis) versus the pooled reference (consisting of wild type and other mutants from the same strain; cell counts from replicates were merged) across individual sub-trajectories (*x* axis). Only nominally significant results are shown (Methods). The pie colour and direction correspond to whether the log₂-transformed ratio is >0 (blue, clockwise) or <0 (red, anticlockwise); the pie size and colour intensity correspond to the scale of the log₂-transformed ratio. log₂-transformed ratios that were >2 or <−2 were manually set to 2 or −2. The number of cells assigned to each sub-trajectory and selected thresholds of proportions (red vertical lines) are shown above. **b**, Three-dimensional UMAP visualization of the neural tube trajectory, highlighting cells from either *Ttc21b*-KO (left) or wild-type (middle) C57BL/6,

or other C57BL/6 mutants (right), after downsampling a uniform number of cells per plot. **c**, Haematoxylin and eosin staining of the developing eye of homozygous mutant (Hom), heterozygous mutant (Het) and wild-type *Ttc21b* E13.5 embryos (Methods). Structures are lost exclusively in homozygous mutants. Scale bars, 500 μm. **d**, UMAP visualization of co-embedded cells of the limb mesenchyme trajectory from ZRS limb enhancer KO and wild-type FVB embryos. The same UMAP is shown twice, highlighting FVB wild-type (left) or ZRS limb enhancer KO (right) cells. A subpopulation exhibiting extreme loss in the ZRS limb enhancer KO is circled. **e**, The same UMAP as in **d**, coloured by expression of marker genes that seem specific to proximal (*Meis1* and *Meis2*) or distal (*Hoxa13*, *Hoxd13*, *Lhx9* and *Msx1*) limb development (Supplementary Table 4). MHB, midbrain–hindbrain boundary; Di, diencephalon.

support these patterns, as the homozygous *Ttc21b* mutant exhibits a visible collapse in structures that are detectable within the wild-type eye at E13.5. Specifically, the retinal neurons, lens and optic nerve were missing in the homozygous mutant (Fig. 2c). The retinal epithelium was delocalized and reduced as well (Fig. 2c and Extended Data Fig. 5c).

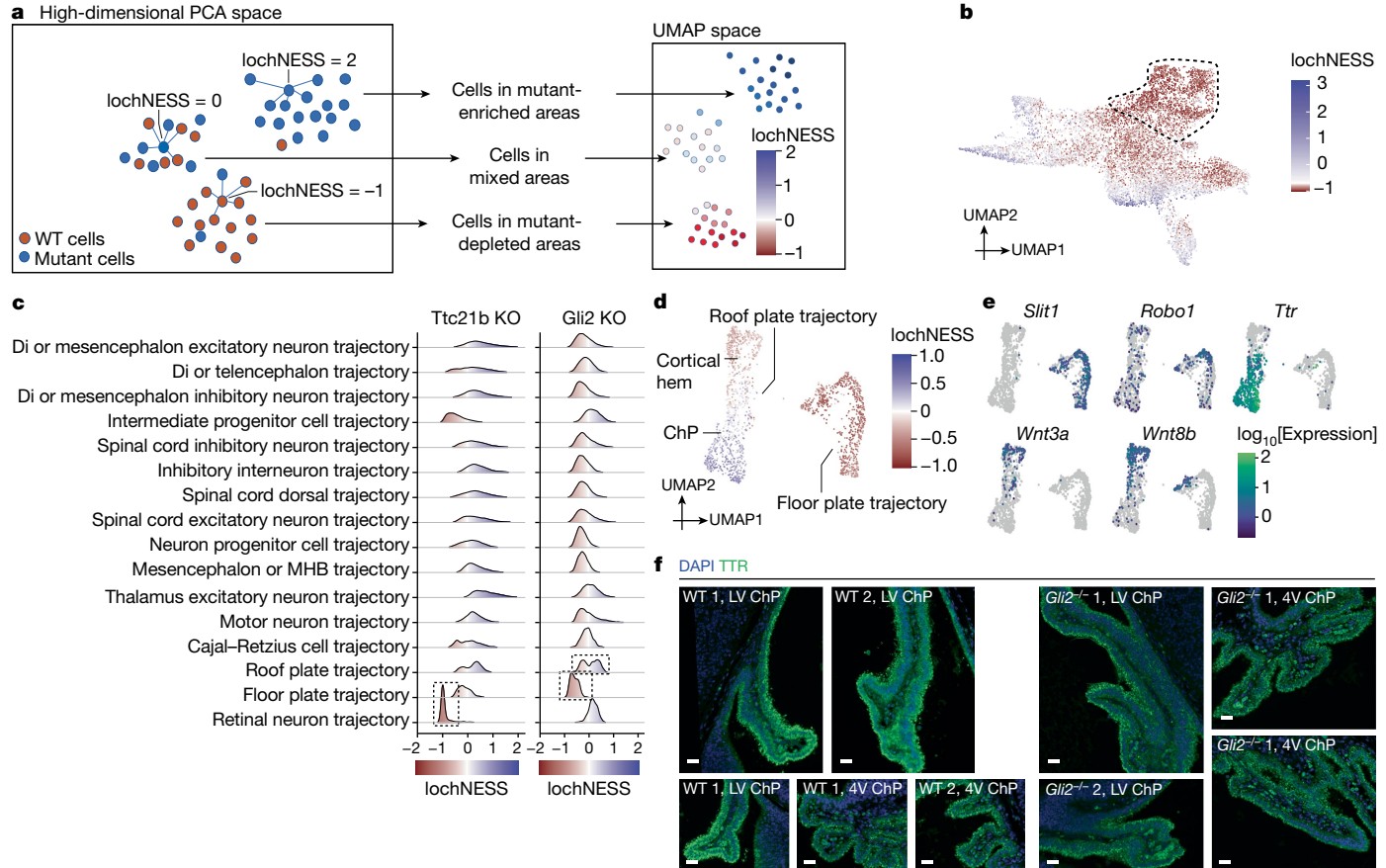

**Fig. 3 | LochNESS highlights mutant-related changes. a**, A schematic of lochNESS calculation and visualization. **b**, UMAP of the limb mesenchyme trajectory from ZRS limb enhancer KO and wild-type cells, coloured by lochNESS. The colour scale is centred at the median. Cells corresponding to a subset of ZRS limb enhancer KO cells with more extreme loss in Fig. 2d are outlined with a dashed line. **c**, Distribution of lochNESS in the neural tube sub-trajectories of *Ttc21b*-KO and *Gli2*-KO mutants. The dashed outlines highlight shifted distributions of the retinal neuron sub-trajectory of the *Ttc21b*-KO mutant and the floor and roof plate sub-trajectories of the *Gli2*-KO mutant. **d**, UMAP of co-embedded cells of floor plate and roof plate sub-trajectories from the *Gli2*-KO mutant and pooled wild type, coloured by lochNESS. **e**, The same as in **d**, but coloured by selected marker gene expression. **f**, Immunofluorescence staining of TTR (ChP marker) in brain regions (LV, lateral ventricle; 4V, fourth ventricle) in sections from the wild type and the *Gli2*-KO mutant (Methods). Scale bars, 50 μm.

We next examined the positive control, the ZRS limb enhancer KO, a well-studied mutant that shows a loss of the distal limb structure at birth[28]. Eight sub-trajectories were nominally altered in this mutant, mostly mesenchymal. Although the reduction in limb mesenchymal cells was modest (24% or log_2[ratio] = −0.39), co-embedding of limb mesenchyme cells from ZRS limb enhancer KO and wild-type FVB embryos identified a subpopulation that specifically expressed markers of the distal mesenchyme of the early embryonic limb bud, such as *Hoxa13* and *Hoxd13*, that was markedly affected (Fig. 2d,e and Extended Data Fig. 5d). Such heterogeneity was not observed for the seven other nominally altered sub-trajectories (Extended Data Fig. 5e), consistent with the specificity of this phenotype.

## Transcriptional heterogeneity within cell types

We next sought to develop a more sensitive approach for detecting deviations in transcriptional programs within cell-type trajectories. For this, we developed the local cellular heuristic neighbourhood enrichment specificity score (lochNESS), a score calculated on the basis of the 'neighbourhood' of each cell in a sub-trajectory co-embedding of a given mutant (all replicates) versus a pooled wild type (all replicates of all backgrounds). Briefly, lochNESS takes aligned PC features of each sub-trajectory and finds *k* nearest neighbours for each cell from other embryos. For each mutant cell, we compute the fold change of the observed versus expected number of mutant cells in its neighbourhood (Fig. 3a and Methods; similar methods developed independently in ref. 34).

Visualization of lochNESS in the embedded space highlights areas with enrichment or depletion of mutant cells. For example, returning to the ZRS limb enhancer KO embryos, we observe markedly low lochNESS in the distal mesenchyme of the early embryonic limb bud (Figs. 2d and 3b). This highlights the value of lochNESS, as within a sub-trajectory (limb mesenchyme), an effect is both detected and assigned to a subset of cells in a label-agnostic fashion.

Globally, the distribution of lochNESS is unremarkable for some mutants (for example, most TAD boundary KOs) but aberrant for others (for example, pleiotropic mutants such as *Sox9* regulatory INV; Extended Data Fig. 6a). After carrying out additional quality control checks (Methods and Extended Data Fig. 6b–d), we examined lochNESS for each mutant in each sub-trajectory. Consistent with earlier analyses, our data show low lochNESS for the retinal neuron sub-trajectory in *Ttc21b*-KO mice (Fig. 3c and Extended Data Fig. 6e). We also observe a strong shift towards low lochNESS for the floor plate sub-trajectory in *Gli2*-KO mice, and a subtle change for the roof plate trajectory, which is forming opposite to the floor plate along the dorsal–ventral axis of the developing neural tube[35] (Fig. 3c and Extended Data Fig. 6e).

To explore this further, we extracted and reanalysed cells corresponding to the floor plate and roof plate (Extended Data Fig. 7a). Within the

floor plate, *Gli2*-KO cells consistently exhibited low lochNESS (Fig. 3d). However, there were only a handful of differentially expressed genes between the wild type and the mutant, and no significantly enriched pathways. For example, genes such as *Robo1* and *Slit1*, involved in neuronal axon guidance, are specifically expressed in the floor plate relative to the roof plate (Fig. 3e), but are not differentially expressed between wild-type and *Gli2*-KO cells of the floor plate. Alternatively, our failure to detect substantial differential expression may be due to power, as there were fewer floor plate cells in the *Gli2* KO (about 60% reduction). Overall, these observations are consistent with the established role of *Gli2* in floor plate induction, its role as an activator of SHH in dorso-ventral patterning of the neural tube and the previous demonstration that *Gli2* KOs fail to properly induce a floor plate[35,36].

Less expectedly, we identified two subpopulations of roof-plate-derivative cell types, one depleted and the other enriched in *Gli2*-KO embryos (Fig. 3d and Extended Data Fig. 7a–c). To annotate these subpopulations, we examined genes whose expression was predicted by lochNESS (Methods). The mutant-enriched group of roof plate cells was marked by cilial genes and *Ttr*, a marker for the choroid plexus (ChP), whereas the mutant-depleted group was marked by WNT-related genes (for example, *Rspo1/2/3* and *Wnt3a/8b/9a*) indicating it to be a region close to the ChP of the lateral ventricle, namely the cortical hem (Fig. 3e, Extended Data Fig. 7d and Supplementary Tables 4 and 5). We also mapped the three clusters shown in Extended Data Fig. 7a to spatial transcriptomic data from E13.5 mouse embryos[37] (Extended Data Fig. 7e). Supporting our annotations, cluster 1 mapped to the floor of the neural tube, cluster 2 next to the lateral ventricle ChP, and cluster 3 to the ChP (both in the lateral (anterior) and fourth (posterior) ventricles). We then examined marker genes that further separate lateral ventricle and fourth ventricle ChP and found that in addition to the roof plate marker *Lmx1a*, cluster 3 expresses the fourth ventricle marker *Meis1* and cluster 2 expresses the lateral ventricle markers *Otx1* and *Emx2* (Extended Data Fig. 7f and Supplementary Table 4).

To experimentally validate these observations, we examined developmental progression of the neural tube and brain in E13.5 *Gli2*-KO mutant and wild-type embryos. In coronal sections of the mutant, we observed severe developmental defects including deformed forebrain lobes and delayed neural tube development (Extended Data Fig. 8a). Immunofluorescence imaging of *Pax6* expression revealed a severely disturbed shape of the neural tube, confirming the well-described 'dorsalization' phenotype of the neural tube (Extended Data Fig. 8b), and consistent with marked reductions in the proportion of floor plate cells in the *Gli2*-KO mutant (Fig. 3d). Turning to the less expected observation of increased ChP, we found that the lateral ventricle as well as the fourth ventricle exhibited a disturbed pattern of staining of *Ttr* expression. Whereas the wild type shows inner and outer *Ttr* signal within the single cell layer, the mutant exhibited a 'double DAPI' layer, indicating a disordered tissue organization (Fig. 3f and Extended Data Fig. 8c,d). Adjusting for the overall smaller size of *Gli2*-KO mutants at E13.5, we quantified cells positive for *Ttr* expression in the lateral and fourth ventricle, and found a proportional increase in the mutant relative to the wild type (Supplementary Table 6), again consistent with the marked increase in the proportion of ChP cells in this mutant (Fig. 3d). In summary, we could confirm both the expected reduction in floor plate and the unexpected increase in roof-plate-derived ChP in the mutant. Of note, the relatively subtle and opposing effects on these roof plate subpopulations were missed by our original analysis of cell-type proportions, and uncovered only by the granularity of lochNESS.

LochNESS distributions can be systematically screened to identify sub-trajectories exhibiting mutant-specific shifts. For example, although all TAD boundary KO mutants have similarly unremarkable global lochNESS distributions, when we plot these distributions by sub-trajectory, a handful of shifted distributions are evident (Extended Data Fig. 9a,b). For example, multiple epithelial sub-trajectories, including pre-epidermal keratinocyte, epidermis, branchial arch and lung epithelial trajectories, are most shifted in *Tbx3* TAD boundary KO cells, with further analyses preliminarily supporting a role for *Tbx3* in epidermal and lung development[38] (Methods, Extended Data Fig. 9c,d and Supplementary Table 7).

## Mutant-specific and mutant-shared effects

Pleiotropy, wherein a single gene influences multiple, unrelated traits, is a pervasive phenomenon in developmental genetics, and yet remains poorly understood[39]. Although here we have 'whole-embryo' molecular profiling of just 22 mutants, we sought to investigate whether we could distinguish between mutant-specific and mutant-shared effects within each major trajectory. In brief, within a co-embedding of cells from all embryos from a given background, we computed *k* nearest neighbours as in Fig. 3a, and then calculated the observed versus expected ratio of each genotype among a cell's *k* nearest neighbours. The 'similarity score' between one genotype versus all others is defined as the mean of these ratios across cells of the genotype (Methods). To assess whether any observed similarities or dissimilarities are robust, we can also calculate similarity scores between individual embryos. For example, for the mesenchymal trajectory of C57BL/6 mutants, similarity scores are generally higher for pairwise comparisons of individuals with the same genotype (Fig. 4a and Extended Data Fig. 10a–c). Pairs of individuals with the *Scn11a*-GOF mutation exhibited the most extreme similarity scores, consistent with our earlier observation that they clustered with E12.5 rather than E13.5 embryos (Fig. 1c). Following further analysis, we believe that the most parsimonious explanation is incorrect staging of these litters, rather than mutation-specific, global developmental delay (Extended Data Fig. 10d–g and Supplementary Note 1).

We also observed that the similarity scores between three mutants (*Atp6v0a2* KO, *Atp6v0a2*[R755Q] and *Gorab* KO) were consistent with shared effects, in the mesenchymal, epithelial, endothelial, hepatocyte and neural crest (peripheral nervous system glia) trajectories in particular; in other major trajectories, such as neural tube and haematopoiesis, the *Atp6v0a2* KO and *Atp6v0a2*[R755Q] exhibited high similarity scores with one another, but not with the *Gorab* KO (Fig. 4a and Extended Data Fig. 10a,c,f). In human patients, mutations in *ATP6V0A2* and *GORAB* cause overlapping connective tissue disorders, which is reflected in the misregulation of the mesenchymal trajectory of *Atp6v0a2* and *Gorab* mutants[25,26]. However, only the ATP6V0A2-related disorder exhibits a prominent central nervous system phenotype, consistent with the changes in the neural tube trajectory seen only in *Atp6v0a2* mutants (Extended Data Fig. 10a,c,f).

To further explore phenotypic sharing between these mutants, we co-embedded cells of the lateral plate and intermediate mesoderm sub-trajectory from C57BL/6 strains. We resolved the identity of most subclusters using marker genes and spatial mapping, identifying multiple subsets for which *Atp6v0a2*-KO, *Atp6v0a2*[R755Q] and *Gorab*-KO mice are similarly distributed compared to other C57BL/6 genotypes (Fig. 4b and Extended Data Fig. 11). Some subsets are enriched for cells from these mutants (for example, proepicardium, hepatic mesenchyme and lung mesenchyme) whereas others are depleted (for example, gastrointestinal smooth muscle; Fig. 4c,d and Supplementary Table 4). Although individually subtle, the consistent shifts in cell-type proportions between the two *Atp6v0a2*- and *Gorab*-KO mutants across these subsets of mesenchyme derived from lateral plate mesoderm presumably underlie their high mesenchymal similarity scores (Fig. 4c).

Altogether, these analyses illustrate how the joint analysis of mutants subjected to whole-embryo scRNA-seq can reveal sharing of molecular and cellular phenotypes. This includes global similarity (*Atp6v0a2* KO versus *Atp6v0a2*[R755Q]) as well as instances in which specific aspects of phenotypes are shared between previously unrelated mutants (*Atp6v0a2* mutants versus *Gorab* KO).

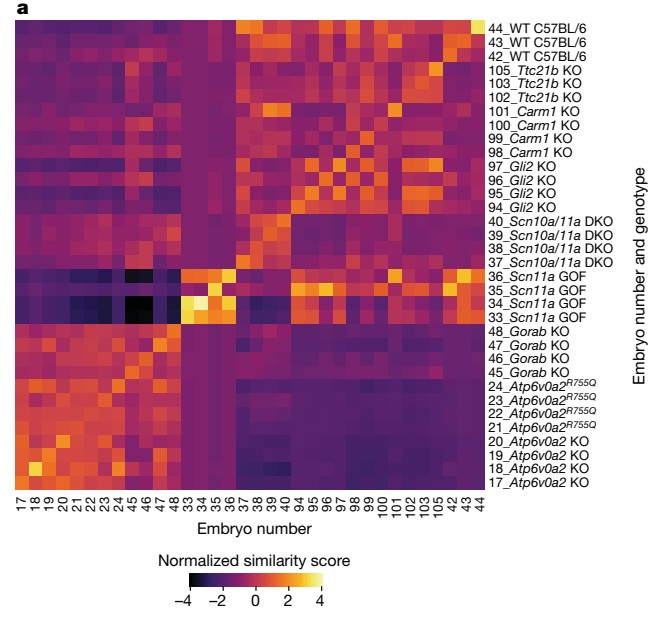

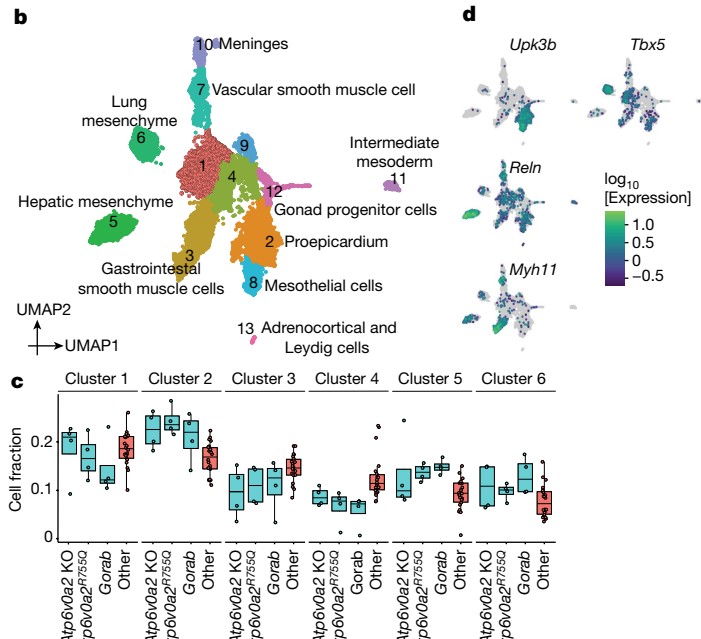

**Fig. 4 | Similarity scores identify mutant-shared and mutant-specific effects. a**, A heatmap showing similarity scores between individual C57BL/6 embryos in the mesenchymal trajectory. The rows and columns are grouped by genotype and labelled by embryo id and genotype. **b**, UMAP of the lateral plate and intermediate mesoderm sub-trajectory for mutants from the C57BL/6 background strain, coloured and labelled by subcluster and detailed cell type (marker genes in Supplementary Table 4). **c**, Boxplots showing the composition of the top six subclusters for individual *Atp6v0a2*-KO, *Atp6v0a2^{R755Q}* and *Gorab*-KO embryos (blue, $n = 4$ each genotype) and other C57BL/6 embryos (red, $n = 23$). Centre lines show medians; box limits indicate 25th and 75th percentiles; replicates are represented by dots. **d**, The same as in **b**, but coloured by log-transformed expression of selected marker genes.

## Mesenchymal stalling in a *Sox9* regulatory mutant

About half of the mutants profiled here model disruptions of regulatory, rather than coding, sequences. Among these, the *Sox9* regulatory INV mutant stands out in having a markedly shifted lochNESS distribution, particularly in mesenchyme (Fig. 5a and Extended Data Fig. 6a). *Sox9* encodes a pleiotropic transcription factor crucial for development of the skeleton, the brain, sex determination and other systems, orchestrated by a complex regulatory landscape[40–42]. This particular mutant features an inversion of a 1-megabase upstream region bearing several distal enhancers and a TAD boundary, essentially relocating these elements into a TAD with *Kcnj2*, which encodes a potassium channel[5] (Fig. 5b). Like the *Sox9* KO, the homozygous *Sox9* regulatory INV is perinatally lethal, with extensive skeletal phenotypes including digit malformation, a cleft palate, bowing of bones and delayed ossification. In addition to the loss of 50% of *Sox9* expression, the inversion causes pronounced misexpression of *Kcnj2* in the digit anlagen in a wild-type *Sox9* pattern[5]. However, the extent to which *Kcnj2* and *Sox9* are misexpressed elsewhere, as well as the molecular and cellular correlates of the widespread skeletal phenotype, have yet to be deeply investigated.

At the level of mesenchymal sub-trajectories, shifts in the lochNESS distribution for *Sox9* regulatory INV were consistently observed, but limb mesenchyme and connective tissue were particularly enriched for cells with extremely high lochNESS (Fig. 5a, right). Notably, two of three major enhancers (E250 and E195) known to drive *Sox9*-mediated chondrogenesis in mesenchymal stem cells are located within the inverted region[40] (Fig. 5b). Cell-type composition analysis (Fig. 2a) showed that *Sox9* regulatory INV mutants harbour considerably larger numbers of cells classified as limb mesenchyme, at the expense of osteoblasts, lateral plate and intermediate mesoderm, chondrocytes and connective tissue trajectories. This shift can also be seen in a uniform manifold approximation and projection (UMAP) embedding (Fig. 5c), a topic we revisit further below.

These changes in cell-type composition were accompanied by reduced expression of *Sox9* and increased expression of *Kcnj2* in bone (aggregate of chondrocyte, osteoblast and limb mesenchyme; Extended Data Fig. 12a), although the number of cells expressing *Kcnj2* was generally low. This suggests that the *Sox9* regulatory inversion is resulting in increased *Kcnj2* expression (through *Sox9* enhancer adoption) and *Sox9* reduction (through boundary repositioning) not only in the digit anlagen, but in skeletal mesenchyme more generally. To validate this, we carried out RNA in situ hybridization (RNAscope) on sections of developing bones of the rib cage at E13.5, comparing a heterozygous *Sox9* regulatory INV mouse with a wild-type littermate. Consistent with our scRNA-seq data derived from homozygous mutants, our data show a *Sox9*-patterned increase in *Kcnj2* levels, together with losses in *Sox9* expression, in the developing bone (Fig. 5d and Extended Data Fig. 12b).

As the inverted *Sox9* regulatory region also hosts multiple enhancers active in other tissues (for example, E161 in lung and E239 in cerebral cortex)[40], we wondered whether these patterns were also seen in other tissues. Indeed, both scRNA-seq and RNAscope quantification show increased *Kcnj2* levels in all other tissues examined. Whereas reductions in *Sox9* expression, clear in bone, were not observed in most other tissues by scRNA-seq, RNAscope showed *Sox9* reductions in telencephalon and lung as well (Extended Data Fig. 12a,b). Taken together, these data suggest marked changes in mesenchyme due to reduced *Sox9*, together with broader increases in *Kcnj2* expression. As expected on the basis of the role of *Sox9* in chondrogenesis, hallmark pathways related to chondrocyte proliferation and differentiation[43–46] were downregulated; less expectedly, several immune-related pathways were upregulated (Extended Data Fig. 12c).

To explore the apparent accumulation of limb mesenchyme in the *Sox9* regulatory INV (Fig. 5c and Extended Data Fig. 12d) in more detail, we reanalysed mutant and wild-type cells from the limb mesenchyme sub-trajectory, which revealed subpopulations of condensing mesenchyme, perichondrium and undifferentiated mesenchyme (Extended

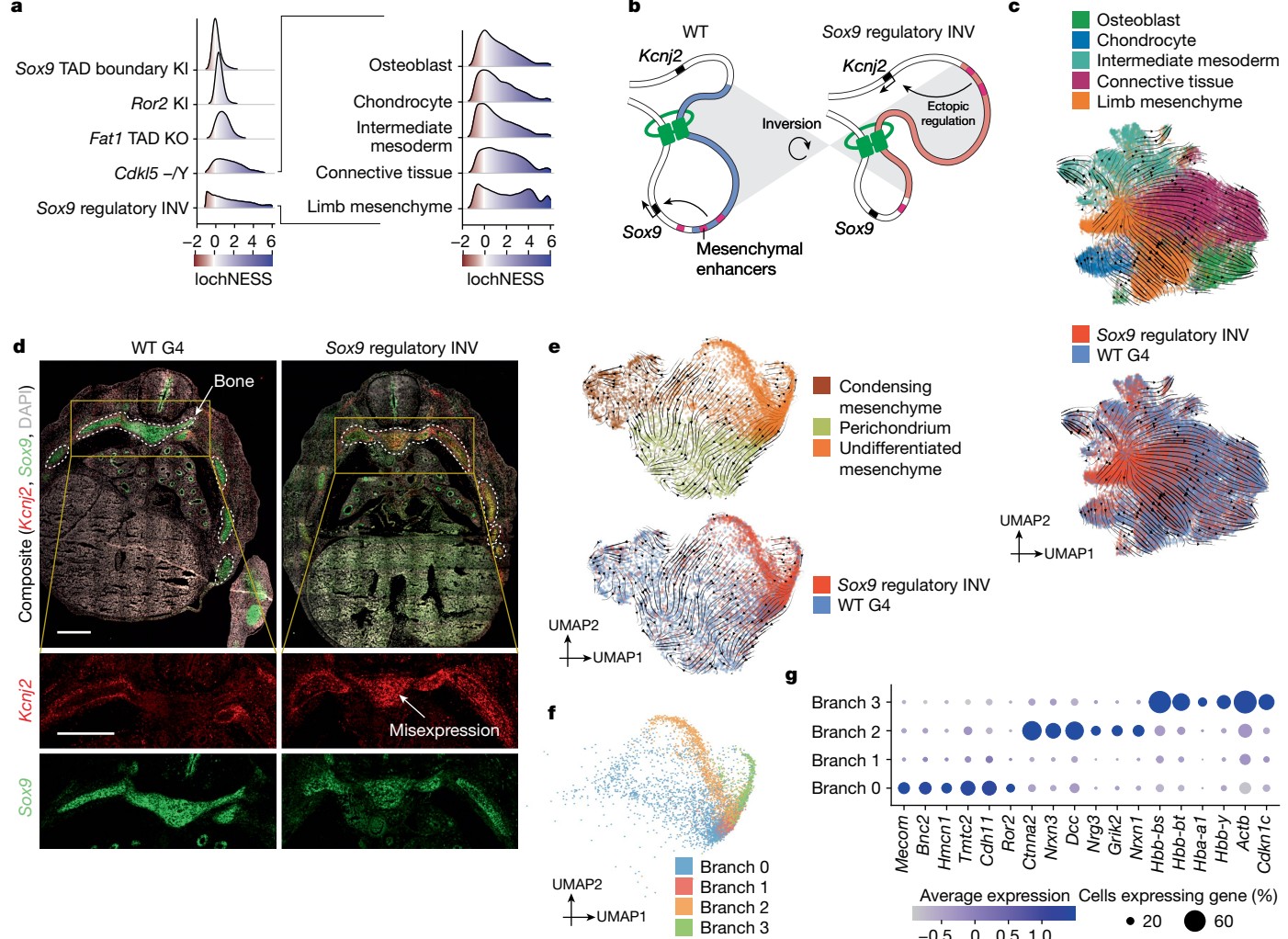

**Fig. 5 | Apparent stalling and redirection of mesenchyme differentiation in the *Sox9* regulatory INV mutant. a**, LochNESS distributions for all G4 mutants in the mesenchymal trajectory (left) and the *Sox9* regulatory INV mutant in mesenchymal sub-trajectories (right). **b**, Model of the *Sox9* regulatory INV mutation depicting ectopic *Kcnj2* regulation through enhancer adoption. **c**, RNA velocity of mesenchymal G4 wild-type and *Sox9* regulatory INV cells coloured by sub-trajectories (top) or genotype (bottom). **d**, *Sox9* regulatory INV heterozygous mutant and littermate wild-type RNAscope images (red: *Kcnj2*; green: *Sox9*), with insets below highlighting a region corresponding to developing bone (area outlined with white dots line). Scale bars, 500 μm. **e**, RNA velocity of G4 wild-type and *Sox9* regulatory INV cells in the limb mesenchymal sub-trajectory labelled by annotation (top) or genotype (bottom). **f**, The same as in **e**, but coloured by branch number. **g**, Dot plot of the top differentially expressed genes in the four branches shown in **f**.

Data Fig. 12e,f). RNA velocity analyses suggested that most limb mesenchyme 'accumulation' in mutant embryos is due to cells that are delayed or stalled in an undifferentiated or stem-like state (Fig. 5c,e and Extended Data Fig. 12e). This accumulation is even more apparent in integrated views of the limb mesenchyme sub-trajectory, for which we observe branches that are highly enriched for *Sox9* regulatory INV mutant cells, within undifferentiated mesenchyme (Fig. 5e and Extended Data Fig. 12g,h).

To investigate these branches further, we subclustered undifferentiated mesenchyme cells from the mutant and wild type (Fig. 5f,g). Notably, the most differentially expressed genes in 'branch 2' were largely neuronal (for example, several neurexins and neuregulin 3), an observation supported by gene set enrichment analysis (Extended Data Fig. 12i,j). A cellular composition analysis revealed that these neuronal-like cells were not restricted to the *Sox9* regulatory INV mutant, but also found in wild-type embryos, albeit much less frequently (Extended Data Fig. 12g,h). To validate this unexpected 'neural-like' branch of mesenchymal cells as well as to assess their anatomical distribution, we mapped these cells to spatial transcriptomic data from E13.5 mouse embryos[37]. Strikingly, this analysis placed

branch 2 cells along the neural tube and the brain regions (Extended Data Fig. 13a). To address concerns that artefacts might arise from mapping single-cell data onto non-single-cell spatial maps, we also integrated our data with sci-space[47] spatial transcriptomic data (E14.5), as these retain single-nucleus resolution. The results are consistent, in that branch 2 mesenchymal cells are enriched in brain regions, branch 0 cells are enriched in limb bud regions, and branch 1 and 3 cells are diffusely distributed but largely excluded from brain regions (Extended Data Fig. 13b).

Taken together, these analyses support the validity of this neural-like subset of mesenchyme (present in the wild type and increased in *Sox9* regulatory INV mutants). The observation is consistent with the reports that mesenchymal stem cells can be differentiated to neuronal states in vitro[48].

## Discussion

Here we set out to establish whole-embryo scRNA-seq as a new paradigm for the systematic, scalable phenotyping of mouse developmental mutants. On data obtained for 22 mutants in a single experiment,

we developed analytical approaches to identify deviations in cell-type composition, subtle differences in gene expression within cell types (lochNESS), and sharing of sub-phenotypes between mutants (similarity scores). Overall, the results are encouraging, and show how systematic, outcome-agnostic computational analyses of data obtained at the whole-embryo scale may in some cases reveal molecular and cellular phenotypes that are missed by conventional phenotyping.

We emphasize that the concurrent analysis of many mutants proved essential to the contextualization of particular observations (that is, to understand how specific or nonspecific any apparent deviation really was) against a background of dozens of genotypes and more than 100 embryos. This also enabled us to discover shared aspects of phenotypes between previously unrelated genotypes (for example, between *Gorab* and *Atp6v0a2* mutants). Looking forward, profiling of additional mouse mutants might enable the further 'decomposition' of developmental pleiotropy, a poorly understood phenomenon, into 'basis vectors'.

The diverse mutants analysed yielded a variety of results that speak to the utility of whole-embryo scRNA-seq for phenotyping. For example, an abnormal eye phenotype in *Ttc21b* mutants was previously described, but considered probably to be secondary to a more general craniofacial defect[20,33]. The scRNA-seq analysis of E13.5 *Ttc21b* mutants demonstrated that multiple retinal cell trajectories were essentially absent. Detailed histological analysis confirmed this, suggesting that the eye abnormality is probably not a secondary effect, but rather that the overactive SHH signalling has a primary effect on retinal development in this mutant.

The utility of pursuing whole-embryo scRNA-seq was also demonstrated by an unexpected finding of both a depleted and an enriched cell population of roof plate cell derivatives in the *Gli2*-KO mutant. The 'dorsalization' of the neural tube in the absence of SHH signalling is well described[20,35,36] and was confirmed in our histological analysis of this line (Extended Data Fig. 8). However, there have been no described changes in the roof plate or its derivatives so far in *Gli2*-KO mice[36]. By contrast, whole-embryo scRNA-seq uncovered that derivatives of the roof plate depict changes in composition (a primary finding) and tissue development (a finding based on secondary validation) in the mutant, illustrating how this approach can potentially yield new insight into even well-studied developmental pathways. However, owing to our dataset capturing only one time point, whether *Gli2* misexpression causes the structural change directly in the derivative tissue or earlier during roof plate formation remains elusive.

Our MMCA has limitations. First, we profiled only four replicates per mutant at a single developmental time point. On the basis of a simulation analysis of the analytical approach that considers only cell proportions, four replicates of each mutant is probably sufficient to detect modest changes in abundant cell types (for example, a 10% change for cell types at 10% abundance) but only large changes in rarer cell types (for example, a 25% change in cell types at 1% abundance; Extended Data Fig. 4b). As such, to detect more subtle changes in model organisms such as mice for which very large numbers of replicates are not feasible, more sophisticated strategies such as lochNESS, which is not based on counts of cell types but rather directly considers the distribution of cells derived from different genotypes in a complex embedding, may be essential. It is important to note that our cell composition analysis, which includes both wild-type and mutant cells from the same strain to generate a pooled reference, assumes that the cell-type proportions of non-wild-type genotypes are roughly consistent, at least on the whole, with those of wild-type cells. This assumption may be more problematic in studies of biologically related mutants. Of note, in concurrently published studies in this issue, a similar approach is taken for genetic and environmental perturbations in zebrafish (ref. 49), such that dozens to hundreds of replicate embryos of each genotype can be profiled and phenotypic variability quantified.

Second, profiling only a small fraction of cells present in E13.5 embryos potentially limits sensitivity. However, for any given mutant,

we had more than 1.5 million cells from other genotypes (wild type or other mutants), which facilitated the detection of mutant-specific phenotypes for rare cell types (for example, in the retina (*Ttc21b* KO) and roof plate (*Gli2* KO)).

Third, we were not able to explore all mutants in detail, nor to thoroughly investigate other aspects of the data (for example, the differences between wild-type strains). In the future, we anticipate that community input and domain expertise will be essential to extract full value from these data. To facilitate this, we created an interactive browser that allows exploration of mutant-specific effects on gene expression in trajectories and sub-trajectories, together with the underlying data (https://atlas.gs.washington.edu/mmca_v2/). Additionally, some of the phenotypes identified here have probably not been described before owing to the lack of resolution of conventional phenotyping. New secondary validation strategies need to be developed to confirm subtle defects in molecular programs or subtle changes in the relative proportions of specific cell types. A promising approach would be to complement whole-embryo scRNA-seq with rapidly advancing methods for whole-mouse-body antibody labelling and three-dimensional imaging[50].

Fourth, our results emphasize the importance of a well-matched control; although data from our wild-type embryos could be reused as control data for future studies of additional mutants, that risks batch effects, and a safer strategy would be to always include a well-matched, 'in-line' wild-type control while profiling mutant embryos.

In 2011, the International Mouse Phenotyping Consortium set out to drive towards the 'functionalization' of every protein-coding gene in the mouse, by generating thousands of KO mouse lines[51]. In principle, the whole-embryo scRNA-seq phenotyping approach presented here could be extended to all Mendelian genes or even to all 20,000 mouse gene KOs.

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

## Methods

### Data reporting

No statistical methods were used to predetermine sample size. Embryos used in experiments were randomized before sample preparation. Investigators were blinded to group allocation during data collection and analysis. Embryo collection and sci-RNA-seq3 analysis were carried out by different researchers in different locations.

### Embryo collection

Mutants were generated through conventional gene-editing tools and breeding or tetraploid aggregation and collected at the embryonic stage E13.5, calculated from the day of vaginal plug (noon = E0.5). Collection and whole-embryo dissection were carried out as previously described[52]. The embryos were immediately snap-frozen in liquid nitrogen and shipped to the Shendure Lab (University of Washington) in dry ice. Sets of animals with the same genotype were either all male or half male–half female. All animal procedures were carried out in accordance with institutional, state and government regulations.

### Isolation and fixation of nuclei

Snap-frozen embryos were processed as previously described[3]. Briefly, the frozen embryos were cut into small pieces with a blade and further dissected by resuspension in 1 ml ice-cold cell lysis buffer (10 mM Tris-HCl, pH 7.4, 10 mM NaCl, 3 mM $MgCl_2$, 0.1% IGEPAL CA-630, 1% SUPERase In and 1% BSA) in a 6-cm dish. After adding another 3 ml of cell lysis buffer, the sample was strained (40 µm) into a 15-ml Falcon tube and centrifuged to a pellet (500g, 5 min). By resuspending the sample with another 1 ml of cell lysis buffer, the isolation of nuclei was ensured. The nuclei were pelleted again (500g, 5 min) and then washed and fixed in 10 ml 4% paraformaldehyde (PFA) for 15 min on ice. The fixed nuclei were pelleted (500g, 3 min) and washed twice in the nucleus suspension buffer (500g, 5 min). The nuclei finally were resuspended in 500 µl nucleus suspension buffer and split into two tubes, each containing 250 µl of sample. The tubes were flash frozen in liquid nitrogen and stored in a −80 °C freezer, until further use for library preparation. The embryo preparation was carried out randomly for nuclei isolation to avoid batch effects.

### sci-RNA-seq3 library preparation and sequencing

The library preparation was carried out as previously described[53]. In short, the fixed nuclei were permeabilized, sonicated and washed. Nuclei from each mouse embryo were then distributed into several individual wells in four 96-well plates. We split samples into four batches (about 25 samples randomly selected in each batch) for sci-RNA-seq3 processing. The ID of the reverse transcription well was linked to the respective embryo for downstream analysis. In a first step, the nuclei were then mixed with oligo-dT primers and dNTP mix, denatured and placed on ice; afterwards, they were processed for reverse transcription including a gradient incubation step. After reverse transcription, the nuclei from all wells were pooled with the nuclei dilution buffer (10 mM Tris-HCl, pH 7.4, 10 mM NaCl, 3 mM $MgCl_2$, 1% SUPERase In and 1% BSA), spun down and redistributed into 96-well plates containing the reaction mix for ligation. The ligation proceeded for 10 min at 25 °C. Afterwards, nuclei again were pooled with nuclei suspension buffer, spun down and washed and filtered. Next, the nuclei were counted and redistributed for second strand synthesis, which was carried out at 16 °C for 3 h. Afterwards, tagmentation mix was added to each well, and tagmentation was carried out for 5 min at 55 °C. To stop the reaction, DNA binding buffer was added and the sample was incubated for another 5 min. Following an elution step using AMPure XP beads and elution mix, the samples were subjected to PCR amplification to generate sequencing libraries.

Finally after PCR amplification, the resulting amplicons were pooled and purified using AMPure XP beads. The library was analysed by electrophoresis and the concentration was calculated using Qubit (Invitrogen). The library was sequenced on the NovaSeq platform (Illumina; read 1: 34 cycles, read 2: 100 cycles, index 1: 10 cycles, index 2: 10 cycles).

### Processing of sequencing reads

Read alignment and cell × gene expression count matrix generation was carried out on the basis of the pipeline that we developed for sci-RNA-seq3 (ref. 3) with the following minor modifications: base calls were converted to fastq format using Illumina's bcl2fastq v2.20 and demultiplexed on the basis of PCR i5 and i7 barcodes using the maximum-likelihood demultiplexing package deML[54] with default settings. Downstream sequence processing and cell × gene expression count matrix generation were similar to sci-RNA-seq[55] except that the reverse transcription (RT) index was combined with the hairpin adaptor index, and thus the mapped reads were split into constituent cellular indices by demultiplexing reads using both the RT index and the ligation index (Levenshtein edit distance (ED) < 2, including insertions and deletions). Briefly, demultiplexed reads were filtered on the basis of the RT index and ligation index (ED < 2, including insertions and deletions) and adaptor-clipped using trim_galore v0.6.5 with default settings. Trimmed reads were mapped to the mouse reference genome (mm10), using STAR v2.6.1d[56] with default settings and gene annotations (GENCODE VM12 for mouse). Uniquely mapping reads were extracted, and duplicates were removed using the unique molecular identifier (UMI) sequence (ED < 2, including insertions and deletions), reverse transcription (RT) index, hairpin ligation adaptor index and read 2 end-coordinate (that is, reads with UMI sequence less than 2 ED, RT index, ligation adaptor index and tagmentation site were considered duplicates). Finally, mapped reads were split into constituent cellular indices by further demultiplexing reads using the RT index and ligation hairpin (ED < 2, including insertions and deletions). To generate the cell-x-gene expression count matrix, we calculated the number of strand-specific UMIs for each cell mapping to the exonic and intronic regions of each gene with Python v2.7.13 HTseq package[57]. For multi-mapped reads, reads were assigned to the closest gene, except in cases in which another intersected gene fell within 100 base pairs of the end of the closest gene, in which case the read was discarded. For most analyses, we included both expected-strand intronic and exonic UMIs in the cell-x-gene expression count matrix.

The single-cell gene count matrix included 1,941,605 cells after cells with low quality (UMI ≤ 250 or detected gene ≤ 100) were filtered out. Each cell was assigned to its original mouse embryo on the basis of the reverse transcription barcode. We applied three strategies to detect potential doublet cells. As the first strategy, we split the dataset into subsets for each individual, and then applied the scrublet v0.1 pipeline[58] to each subset with parameters (min_count = 3, min_cells = 3, vscore_percentile = 85, n_pc = 30, expected_doublet_rate = 0.06, sim_doublet_ratio = 2, n_neighbors = 30, scaling_method = 'log') for doublet score calculation. Cells with doublet scores above 0.2 were annotated as detected doublets (5.5% in the whole dataset).

As the second strategy, we used an iterative clustering strategy based on Seurat v3 (ref. [59]) to detect the doublet-derived subclusters for cells. Briefly, gene count mapping to sex chromosomes was removed before clustering and dimensionality reduction, and then genes with no count were filtered out and each cell was normalized by the total UMI count per cell. The top 1,000 genes with the highest variance were selected. The data was log-transformed after adding a pseudocount, and scaled to unit variance and zero mean. The dimensionality of the data was reduced by PCA (30 components) first and then with UMAP, followed by Louvain clustering carried out on the 10 PCs (resolution = 1.2). For Louvain clustering, we first fitted the top 10 PCs to compute a neighbourhood graph of observations (k.param = 50) followed by clustering the cells into subgroups using the Louvain algorithm. For UMAP visualization, we directly fitted the PCA matrix with min_distance = 0.1.

For subcluster identification, we selected cells in each major cell type and applied PCA, UMAP and Louvain clustering similarly to the major cluster analysis. Subclusters with a detected doublet ratio (by Scrublet) above 15% were annotated as doublet-derived subclusters.

We found that the above Scrublet and iterative clustering-based approach is limited in marking cell doublets between abundant cell clusters and rare cell clusters (for example, less than 1% of the total cell population); thus, we applied a third strategy to further detect such doublet cells. Briefly, cells labelled as doublets (by Scrublet) or from doublet-derived subclusters were filtered out. For each cell, we retained only protein-coding genes, long intergenic noncoding RNA genes and pseudogenes. Genes expressed in fewer than 10 cells and cells expressing fewer than 100 genes were further filtered out. The downstream dimension reduction and clustering analysis were carried out with Monocle v3 (ref. 3). The dimensionality of the data was reduced by PCA (50 components) first on the top 5,000 most highly variable genes and then with UMAP (max_components = 2, n_neighbors = 50, min_dist = 0.1, metric = 'cosine'). Cell clusters were identified using the Leiden algorithm implemented in Monocle v3 (resolution = $1 \times 10^{-6}$). Next, we took the cell clusters identified by Monocle v3 and first computed differentially expressed genes across cell clusters with the top_markers function of Monocle v3 (reference_cells = 1,000). We then selected a gene set combining the top 10 gene markers for each cell cluster (filtering out genes with fraction_expressing <0.1 and then ordering by pseudo_R2). Cells from each main cell cluster were selected for dimension reduction by PCA (10 components) first on the selected gene set of top cluster-specific gene markers, and then by UMAP (max_components = 2, n_neighbors = 50, min_dist = 0.1, metric = 'cosine'), followed by clustering identification using the Leiden algorithm implemented in Monocle v3 (resolution = $1 \times 10^{-4}$). Subclusters showing low expression levels of markers specific for target cell clusters and enriched expression levels of markers specific for non-target cell clusters were annotated as doublet-derived subclusters and filtered out in visualization and downstream analysis. Finally, after removing the potential doublet cells detected by either of the above three strategies, 1,671,270 cells were retained for further analyses.

## Whole-mouse-embryo analysis

As described previously[3], each cell could be assigned to the mouse embryo from which it derived on the basis of its reverse transcription barcode. After removing doublet cells and another 25 cells that were poorly assigned to any mouse embryo, 1,671,245 cells from 103 individual mouse embryos were retained (a median of 13,468 cells per embryo). UMI counts mapping to each sample were aggregated to generate a pseudobulk RNA-seq profile for each sample. Each cell's counts were normalized by dividing them by the estimated size factor, and then the data were $\log_2$-transformed after adding a pseudocount followed by carrying out the PCA. The normalization and dimension reduction were carried out in Monocle v3.

We previously used sci-RNA-seq3 to generate the MOCA dataset, which profiled about 2 million cells derived from 61 wild-type B6 mouse embryos staged between stages E9.5 and E13.5. The cleaned dataset, including 1,331,984 high-quality cells, was generated by removing cells with <400 detected UMIs as well as doublets (http://atlas.gs.washington.edu/mouse-rna). UMI counts mapping to each sample were aggregated to generate a pseudobulk RNA-seq profile for each embryo. Each cell's counts were normalized by dividing them by the estimated size factor, and then the data were $\log_2$-transformed after adding a pseudocount, followed by PCA. The PCA space was retained, and then the embryos from the MMCA dataset were projected onto it.

## Cell clustering and annotation

After removing doublet cells, genes expressed in fewer than 10 cells and cells expressing fewer than 100 genes were further filtered out. We also filtered out low-quality cells on the basis of the proportion of reads mapping to the mitochondrial genome (MT%) or ribosomal genome (Ribo%) (specifically, filtering cells with MT% > 10 or Ribo% > 5). We then removed cells from two embryos that were identified as outliers on the basis of the whole-mouse-embryo analysis (embryo 41 and embryo 104). This left 1,627,857 cells (median UMI count 845; median genes detected 539) from 101 individual embryos that were retained for all subsequent analyses.

To eliminate the potential heterogeneity between samples due to different mutant types and genotype backgrounds, we sought to carry out the dimensionality reduction on a subset of cells from the wild-type mice (including 15 embryos with 215,575 cells, 13.2% of all cells) followed by projecting all remaining cells, derived from the various mutant embryos, onto this same embedding. These procedures were carried out using Monocle v3. In brief, the dimensionality of the subset of data from the wild-type mice was reduced by PCA, retaining 50 components, and all remaining cells were projected onto that PCA embedding space. Next, to mitigate potential technical biases, we combined all cells from wild-type and mutant mice and applied the align_cds function implemented in Monocle v3, with MT%, Ribo% and log-transformed total UMI of each cell as covariates. We took the subset of cells from wild-type mice, using their 'aligned' PC features to carry out UMAP (max_components = 3, n_neighbors = 50, min_dist = 0.01, metric = 'cosine') by uwot v0.1.8, followed by saving the UMAP space. Cell clusters were identified using the Louvain algorithm implemented in Monocle v3 on three dimensions of UMAP features, resulting in 13 isolated major trajectories (Fig. 1e). We then projected all of the remaining cells from mutant mouse embryos onto the previously saved UMAP space and predicted their major-trajectory labels using a $k$-nearest-neighbour ($k$-NN) heuristic. Specifically, for each mutant-derived cell, we identified its 15 nearest-neighbour wild-type-derived cells in UMAP space and then assigned the major trajectory with the maximum frequency within that set of 15 neighbours as the annotation of the mutant cell. We calculated the ratio of the maximum frequency to the total as the assigned score. Of note, more than 99.9% of the cells from the mutant mice had an assigned score greater than 0.8. The cell-type annotation for each major trajectory was based on expression of the known marker genes (Supplementary Table 2).

Within each major trajectory, we repeated a similar strategy, but with slightly adjusted PCA and UMAP parameters. For the major trajectories with more than 50,000 cells, we reduced the dimensionality by PCA to 50 PCs; for the other major trajectories of more than 1,000 cells, we reduced the dimensionality by PCA to 30 PCs; for the remaining major trajectories, we reduced the dimensionality by PCA to 10 PCs. UMAP was carried out with max_components = 3, n_neighbors = 15, min_dist = 0.1, metric = 'cosine'. For the mesenchymal trajectory, we observed a notable separation of cells by their cell-cycle phase in the UMAP embedding. We calculated a g2m index and an $s$ index for individual cells by aggregating the log-transformed normalized expression for marker genes of the G2M phase and the S phase and then included them in the align_cds function along with the other factors. Applying these procedures to all of the major trajectories, we identified 64 sub-trajectories in total. Similarly, after assigning each cell from the mutant mice with a sub-trajectory label, we calculated the ratio of the maximum frequency to the total as the assigned score. Of note, more than 96.7% of the cells from the mutant mice had an assigned score greater than 0.8. The cell-type annotation for each sub-trajectory was also based on the expression of known marker genes (Supplementary Table 2).

## Identification of correlated cell trajectories between datasets

To identify correlated cell trajectories between MOCA and MMCA datasets, we first calculated an aggregate expression value for each gene in each cell trajectory by summing the log-transformed normalized UMI counts of all cells of that trajectory. For consistency during the comparison to MOCA, we manually regrouped the cells from the MMCA dataset into 10 cell trajectories, by merging the olfactory sensory

neuron trajectory into the neural crest (peripheral nervous system neuron) trajectory, merging the myotube trajectory, the myoblast trajectory and the cardiomyocyte trajectory into the mesenchymal trajectory, and splitting the hepatocyte trajectory into the lens epithelial trajectory and the liver hepatocyte trajectory. Next, for the two datasets, we applied non-negative least-squares regression to predict gene expression in a target trajectory ($T_a$) in dataset A based on the gene expression of all trajectories ($M_b$) in dataset B: $T_a = \beta_{0a} + \beta_{1a}M_b$, based on the union of the 3,000 most highly expressed genes and 3,000 most highly specific genes in the target trajectory. We then switched the roles of datasets A and B; that is, predicting the gene expression of the target trajectory ($T_b$) in dataset B from the gene expression of all trajectories ($M_a$) in dataset A: $T_b = \beta_{0b} + \beta_{1b}M_a$. Finally, for each trajectory $a$ in dataset A and each trajectory $b$ in dataset B, we combined the two correlation coefficients: $\beta = 2(\beta_{ab} + 0.001)(\beta_{ba} + 0.001)$ to obtain a statistic, for which high values reflect reciprocal, specific predictivity. We repeated this analysis on sub-trajectories within each major trajectory.

## Identification of significant cell composition changes in mutant mice using beta-binomial regression

A cell number matrix of all 64 developmental sub-trajectories (rows) and 101 embryos (columns) was created and the cell numbers were then normalized by the size factor of each column that was estimated by the estimate_size_factors function in Monocle v3. Ten sub-trajectories with a mean cell number across individual embryos <10 were filtered out. The beta-binomial regression was carried out using the VGAM package of R. The following code was used: vglm(cbind($n_{celltype}$, $n_{total}$ - $n_{celltype}$) ~ genotype, family = betabinomial), where $n_{celltype}$ refers to the trajectory-specific cell number, and $n_{total}$ refers to the total cell number of that embryo. Of note, embryos from the four different mouse strain backgrounds were analysed independently.

We reason that the power of our strategy to detect the cell proportion changes between different genotypes is affected by three factors: the abundance of a given cell type; the number of replicates in each genotype group; and the effect size. To evaluate power, we carried out a simulation analysis that varied these factors, implemented as follows.

1. We selected the 20 most abundant cell types in wild-type embryos. Their abundances ranged from about 1% to about 20%. The proportions of these cell types served as the basis for our simulations.
2. We simulated ten groups of 'wild-type' samples with 4, 8, 16, …, 40 replicates in each group, wherein each sample consisted of cells drawn from the 20 cell types. For each replicate, the simulated number of cells of each cell type was calculated as the product of: (a) the cell-type proportions, simulated by fitting a Dirichlet model based on the real proportions from step 1; and (b) the total number of cells recovered for that replicate, simulated on the basis of the mean ($n \approx 15,000$) and standard deviation of the cell numbers across replicates in the real dataset.
3. We simulated ten groups of 'mutant' samples by repeating the above step except adding shifts to the numbers of cells within each cell type. The shifting scales were based on different effect sizes. For instance, effect size = 0.1 represents a 10% reduction in the number of cells.
4. We carried out beta-binomial regression (the same test used in Fig. 2a) to test whether the cell-type proportions were significantly changed between simulated 'wild-type' and 'mutant' samples, further checking the results as stratified by cell type (with different abundances), the number of replicates and the effect size.

The results are in line with our hypothesis that the detection power of our strategy varies among comparisons with different effect sizes, sample sizes or cell-type abundances (Extended Data Fig. 4). The main 'take-home' messages are summarized below.

1. Changes of 25% are robustly detectable, even for rare cell types (for example, <2%), with modest numbers of replicates.

2. Changes of 10% are possible to detect, but only for abundant cell types (for example, >5%). More replicates can help in this zone.
3. Changes of 1% are almost impossible to detect with a cell proportion approach, even with very large numbers of replicates.

In general, at the level of single-cell sampling carried out in our study, four samples (corresponding to the number of samples used in the manuscript) would be sufficient to detect a 25% effect size for those cell types present at a 1% proportion in wild-type embryos.

## Defining and calculating lochNESS

To identify local enrichments or depletions of mutant cells, we aim to define a metric for each single cell to quantify the enrichments or depletions of mutant cells in its surrounding neighbourhood. For these analyses, we consider a mutant and a pooled wild type combining all four background strains in a major trajectory as a dataset. For each dataset, we define lochNESS as:

$$\text{lochNESS} = \frac{\frac{\text{number of mutant cells in } k\text{-NNs}}{k}}{\frac{\text{number of mutant cells in dataset}}{N}} - 1,$$

in which $N$ is the total number of cells in the dataset, $k = \frac{\sqrt{N}}{2}$ scales with $N$, and the cells from the same embryo as the cell of interest are excluded from the $k$-NNs. Note that this value is equivalent to the fold change of mutant cell percentage in the neighbourhood of a cell relative to in the whole major trajectory. For implementation, we took the aligned PCs in each sub-trajectory as calculated above, and for each cell in an embryo we find the $k$-NNs in the remaining mutant embryo cells and wild-type cells. We plot the lochNESS in a red–white–blue scale, for which white corresponds to 0 or the median lochNESS, blue corresponds to high lochNESS or enrichments, and red corresponds to low lochNESS or depletions.

At present, we calculate lochNESS using a pooled wild type combining all four background strains to include larger numbers of cells in constructing the $k$-NN graph. If the numbers of cells are sufficient, a wild type from the matched background strain can be used. Additionally, if the numbers of cells are sufficient, one set of lochNESS can be calculated for each wild-type sample separately and the variability between samples can be considered.

## Examining global distributions of lochNESS

Plotting the global distributions of lochNESS for each mutant across all sub-trajectories, we further observed that some mutants (for example, most TAD boundary KOs; $Scn11a$ GOF) exhibit unremarkable distributions (Extended Data Fig. 6a). However, others (for example, $Sox9$ regulatory INV; $Scn10a/11a$ double KO) are associated with a marked excess of high lochNESS, consistent with mutant-specific effects on transcriptional state across many developmental systems. For reference, we simultaneously create a null distribution of lochNESS using random permutation of the mutant and wild-type cell labels, simulating datasets in which the cells are randomly mixed. Of note, we confirmed that repeating the calculation of lochNESS after random permutation of mutant and wild-type labels resulted in bell-shaped distributions centred around zero (Extended Data Fig. 6b). As such, the deviance of lochNESS can be summarized as the average Euclidean distance between lochNESS versus lochNESS under permutation (Extended Data Fig. 6c). In addition, we computed lochNESS between wild types from different background strains and observed minimal variation in cell distribution between the wild type from G4, FVB and BALB/c strains and potential strain-specific distributions in wild-type C57BL/6 mice (Extended Data Fig. 6d).

## Comparing lochNESS with the batch-mixing score the local inverse Simpson index

LochNESS shares conceptual similarities with batch-correcting measurement scores such as the local inverse Simpson index (LISI)[60], which

quantifies the amount of mixing in a cell's neighbourhood by counting the number of batches represented in the neighbourhood. As a direct comparison, we calculated LISI on each mutant with a pooled wild-type reference in PCA space. We calculated LISI with a dynamic perplexity based on the dataset size (perplexity = floor($0.5 \times \sqrt{(N)}/3$ ), $K = 3 \times$ perplexity), similar to our strategy for determining the neighbourhood size for lochNESS. Focusing on the G4 mutants as an example, the results show a correlation between LISI and lochNESS, for which LISI values close to 1 correspond to the more extreme positive or negative values of lochNESS as expected (Extended Data Fig. 6f). LochNESS has several conceptual advantages compared to LISI. First, lochNESS can easily determine whether the mutant sample is enriched or depleted in an area that is not well mixed using the sign of the value (positive = enrichment, negative = depletion), whereas LISI can separate only mixed (scores approaching 2) versus separated (scores approaching 1). Second, lochNESS can be easily extended to comparisons between multiple samples, whereas LISI is relatively restricted to pairwise comparisons. Third, lochNESS considers a dataset-specific neighbourhood size and baseline proportions.

## Identifying lochNESS-associated gene expression changes

To identify gene expression changes associated with mutant-enriched or mutant-depleted areas, we find differentially expressed genes through fitting a regression model for each gene accounting for lochNESS. We use the fit_models() function implemented in Monocle v3 with lochNESS as the model_formula_str. This essentially fits a generalized linear model for each gene: $\log(y_i) = \beta_0 + \beta_i \times x_i$, in which $y_i$ is the gene expression of gene $i$, $\beta_n$ captures the effect of the lochNESS $x_n$ on expression of gene $i$, and $\beta_0$ is the intercept. For each gene $i$, we test whether $\beta_i$ is significantly different from zero using a Wald test, and after testing all genes, we adjust the $P$ values using the Benjamini–Hochberg procedure to account for multiple hypothesis testing. We identify the genes that have adjusted $P$ value < 0.05 and large positive $\beta_i$ values as associated with mutant-enriched areas, and those with large negative $\beta_i$ values as associated with mutant-depleted areas.

## Systematic screening of lochNESS distributions

LochNESS distributions can be systematically screened to identify sub-trajectories exhibiting substantial mutant-specific shifts. For example, although all TAD boundary KO mutants have similarly unremarkable global lochNESS distributions, when we plot these distributions by sub-trajectory, a handful of shifted distributions are evident (Extended Data Fig. 9a). Such deviations, summarized as the average Euclidean distances between lochNESS and lochNESS under permutation, are visualized in Extended Data Fig. 9b. For example, multiple epithelial sub-trajectories, including pre-epidermal keratinocyte, epidermis, branchial arch and lung epithelial trajectories, are most shifted in *Tbx3* TAD boundary KO cells. Co-embeddings of mutant and wild-type cells of these sub-trajectories, together with regression analysis, identify multiple keratin genes as positively correlated with lochNESS, consistent with a role for *Tbx3* in epidermal development[38] (Extended Data Fig. 9c,d and Supplementary Table 7). The lung epithelial cells were separated into two clusters, with the cluster more depleted in *Tbx3* TAD boundary KO cells marked by expression of *Etv5*, which encodes a transcription factor associated with alveolar type II cell development, as well as *Bmp* signalling genes that regulate *Tbx3* during lung development (*Bmp1/4*), and the distal airway markers *Sox9* and *Id2* (Supplementary Table 4). Of note, the shifts that we observed in *Tbx3* TAD boundary KO cells remain preliminary and would need to be confirmed by further validation experiments.

## Spatial mapping with Tangram

We computationally map our dataset onto a spatially resolved transcriptomics dataset, the mouse organogenesis spatiotemporal transcriptomics atlas (MOSTA) generated with Stereo-seq[37]. The atlas has a total of 53 sagittal sections from C57BL/6 mouse embryos from E9.5 to E16.5 in 1-day intervals, and we obtained one section from the most relevant E13.5 data (E13.5_E1S1.MOSTA.h5ad) from the data-sharing website associated with the manuscript: https://db.cngb.org/stomics/mosta/download/. To map the cells for each single cell cluster on the spatially resolved transcriptomics dataset, we used a machine learning-based method called Tangram[61]. Briefly, Tangram is a computational tool that uses a Bayesian approach to infer the spatial locations of cells in a single-cell transcriptomics dataset on the basis of their transcriptomic profiles and the spatial patterns of gene expression in the spatially resolved dataset. The relevant subset of the MMCA data was preprocessed in Scanpy, but the metadata were inherited from the results generated in the section above entitled Cell clustering and annotation. We used Tangram with default parameters to estimate the spatial coordinates of cells from each cluster in the single-cell dataset and visualized results on the coordinates provided by MOSTA. We trained the Tangram model in gpu mode using an NVIDIA A100 GPU. Overall, Tangram provided a powerful method for mapping the cells from the scRNA-seq dataset onto MOSTA, enabling us to infer the spatial locations of different cell clusters of interest within the tissue.

## Calculating mutant and embryo similarity scores

We can extend the lochNESS analysis, which is computed on each mutant and its corresponding wild-type mice, to compute 'similarity scores' between all pairs of individual embryos from the same background strain. We consider all embryos in the same background in a major trajectory as a dataset. For each dataset, we define a 'similarity score' between cell $n$ and embryo $j$ as:

$$\text{similarity score}_{\text{cell }n, \text{embryo }j} = \frac{\frac{\text{no. of cells from embryo } j \text{ in } k\text{-NNs of cell } n}{k}}{\frac{\text{no. of cells from embryo } j \text{ in dataset}}{N}}$$

in which $N$ is the total number of cells in the dataset and $k = \frac{\sqrt{N}}{2}$. We take the mean of the similarity scores across all cells in the same embryo, resulting in an embryo similarity score matrix for which entries are:

$$\text{similarity score}_{\text{embryo }i, \text{embryo }j} = \frac{1}{n_i} \sum_{n=1}^{n_i} \text{similarity score}_{\text{cell }n, \text{embryo }j}$$

in which $n_i$ is the number of cells in embryo $i$.

## Identifying and quantifying developmental delay

To identify potential mutant-related developmental delay, we integrate MMCA with MOCA. We consider a mutant and its corresponding wild type in a sub-trajectory as a dataset. We take the cells from E11.5 to E13.5 with similar annotations from MOCA and co-embed with the MMCA cells. We take the raw counts from both datasets, normalize and process the data together without explicit batch correction as both datasets were generated with sci-RNA-seq3 and were similar in dataset quality. We visualize the co-embedded data in three-dimensional UMAP space and check for developmental delay in the mutant cells (that is, mutant cells embedded closer to early MOCA cells compared to wild-type cells). To quantify the amount of developmental delay, we find k-NNs in MOCA for each cell in MMCA and calculate time score = $\frac{\sum_{n=1}^{k} T_n}{k}$, in which $T_n$ is the developmental time of MOCA cell $n$ in the k-NNs of the MMCA cell. Afterwards, we test whether the average time scores of mutant cells are significantly different from that of wild-type cells using a Student's $t$-test.

## RNAscope in situ hybridization

For RNAscope, embryos were collected at stage E13.5 and fixed for 4 h in 4% PFA in PBS at room temperature. The embryos were washed twice in PBS before incubation in a sucrose series (5%, 10% and finally 15% sucrose (Roth) in PBS) each for 1 h or until the embryos sank to the bottom of the tube. Finally, the embryos were incubated in 15% sucrose in

PBS and O.C.T. (Sakura) in a 1:1 solution before embedding the embryos in O.C.T. in a chilled ethanol bath and storing them at −80 °C until sectioning. The embryos were cut into 5-μm-thick sections on slides for RNAscope.

Simultaneous RNA in situ hybridization was carried out using the RNAscope technology (Advanced Cell Diagnostics (ACD)) and the following probes specific for Mm-K (catalogue number 476261, ACD) and Mm-Sox9-C2 (catalogue number 401051-C2, ACD) on 5-μm sections of the mouse embryos. RNAscope probes were purchased from ACD and designed as described[62]. The RNAscope assay was run on a HybEZ II Hybridization System (catalogue number 321720, ACD) using the RNAscope Multiplex Fluorescent Reagent Kit v2 (catalogue number 323100, ACD) and the manufacturer's protocol for fixed-frozen tissue samples with target retrieval on a hotplate for 5 min. Fluorescent labelling of the RNAscope probes was achieved by using OPAL 520 and OPAL 570 dyes (catalogue numbers FP1487001KT and FP1488001KT, Akoya Biosciences), and stained sections were scanned at ×25 magnification using an LSM 980 with Airyscan 2 (Zeiss).

## Image analysis
For quantitative analysis of the RNAscope images, representative fields of view for each stained section were analysed using the image processing software Fiji[63]. The mRNA signal for each organ of interest was counted in a defined area ($1 \times 1$ mm$^2$), with $n = 6$ per condition. Statistics were calculated using Student' $t$-test and evaluated (not significant, $P > 0.05$; *$P < 0.05$ to $\geq 0.01$; **$P < 0.01$ to $\geq 0.001$; ***$P < 0.001$).

## Ttc21b- and Gli2-mutant fixation for haematoxylin and eosin staining and immunofluorescence
Homozygous and heterozygous Ttc21b mutants and wild-type E13.5 mouse embryos were fixed overnight in 4% PFA at 4 °C. To stop fixation, the samples were transferred into 70% ethanol, washed twice and dehydrated. In the following, the embryos were embedded in paraffin, and cut into 2.5-μm-thick sections.

## Ttc21b-mutant haematoxylin and eosin staining
Histochemical staining was carried out on the eyes of the embryos using haematoxylin and eosin. Slides were scanned with a digital slide scanner (NanoZoomer 2.0HT) and analysed using NDP.view2 software (Hamamatsu Photonics). The following numbers of embryos were processed: 2 wild type; 2 heterozygous Ttc21b ; 4 homozygous Ttc21b.

## Gli2-mutant haematoxylin and eosin staining and immunofluorescence
For the histological analysis, haematoxylin and eosin staining of E13.5 Gli2-KO mouse embryos, and respective wild-type littermates ($n = 4$ and $n = 2$, respectively), was carried out on 4% paraformaldehyde-fixed paraffin-embedded sections (3 μm). Stained paraffin sections were scanned using a digital slide scanner (NanoZoomer 2.0HT) and examined using NDP.view2 software. The cut regions and positions were annotated according to ref. [64].

The spatial abundance patterns of prealbumin as a marker for ChP and PAX6 as a marker for neural tube development were analysed by immunofluorescence, using specific antibodies (rabbit monoclonal (EPR20971) to prealbumin (1:1,000, Abcam) and rabbit polyclonal antibody to PAX6 (1:200, AB2237 Merck Sigma) in an automated BOND Research Detection system. Antibody binding was detected by goat anti-rabbit Alexa Fluor 488-conjugated secondary antibody (Leica, A-11008). Nuclear counterstaining was achieved using 4′,6-diamino-2-phenylindole (DAPI). In negative-control sections, the primary antibodies were omitted and antibody diluent was applied.

Stained embryo sections were scanned with an AxioScan 7 digital slide scanner (Zeiss).

## Fluorescence quantification
Quantification of prealbumin expression cells was carried out using the image analysis software Definiens Developer XD2 (Definiens). The regions of interest (1–4) within the fourth and lateral ventricle ChP were annotated manually in serial sections. The calculated parameter was the ratio of the total number of prealbumin-positive cells over the embryo section area (in micrometres).

## Statistics and reproducibility
Haematoxylin and eosin staining of the developing eye (Fig. 2c) was carried out on homozygous Ttc21b mutants ($n = 4$), heterozygous Ttc21b mutants ($n = 2$) and wild-type E13.5 embryos ($n = 2$). Experiments on the sections were carried out in parallel to ensure consistency.

Haematoxylin and eosin staining of Gli2-mutant and wild-type embryo sections (Extended Data Fig. 8a) was carried out on homozygous Gli2-KO ($n = 4$) and wild-type ($n = 2$) samples. Experiments on the sections were carried out in parallel to ensure consistency.

Immunofluorescence staining of the ChP marker TTR and neural tube marker PAX6 (Fig. 3f and Extended Data Fig. 8b–d) was carried out on sections of homozygous Gli2-KO ($n = 4$) and wild-type ($n = 2$) samples. Immunofluorescence of the same antibody was carried out on all mutants in parallel to ensure consistency.

Sox9 and Kcnj2 expression of heterozygous E13.5 wild-type and Sox9 regulatory INV mutant embryos ($n = 6$ embryos for each condition) was measured by RNAscope image quantification in a defined area ($1 \times 1$ mm$^2$). Statistics were calculated using a two-sided Student' $t$-test and evaluated as follows: not significant, $P > 0.05$; *$P < 0.05$ to $\geq 0.01$; **$P < 0.01$ to $\geq 0.001$; ***$P < 0.001$. RNAscope of the tissue was carried out on all samples in parallel to ensure consistency.

## Clustering and annotation limb mesenchyme trajectory
Seurat v4.0.6 was used for the analysis. Wild-type cells in the limb mesenchyme trajectory from all wild-type mice ($n = 15$ mice, $n = 25,211$ cells) were used to first annotate the cells. The raw counts were log-normalized, after which PCA was carried out with default parameters on the top 2,000 highly variable genes selected using the vst method. Nearest neighbours were computed on the PCA space, with default parameters, except that all of the PCs computed earlier were used. Clustering was carried out using the Louvain community detection algorithm with a resolution of 0.1, resulting in three clusters. Positive marker genes for these clusters were identified using the Wilcoxon rank-sum test, for which only the genes expressed in at least 20% of the cells in either cell group were considered. The clusters were annotated on the basis of biologically relevant markers (Extended Data Fig. 12f). The newly assigned cell annotations for the limb mesenchyme trajectory cells in the wild-type dataset were transferred to the corresponding cells in the Sox9 regulatory INV mutant using the FindTransferAnchors and TransferData functions using default parameters, except that all of the computed PCs were used. A total of 92.3% of the transferred annotations had a score (prediction.score.max) greater than or equal to 0.8.

## Density visualization and RNA velocity analysis
Using Seurat v4.0.6, the raw counts were log-normalized, and PCA was carried out with default parameters on the top highly variable genes 2,000 genes, selected using the vst method. Dimensionality reduction was carried out using PCA with default parameters, after which the UMAP embedding was carried out on all computed PC components. Density plots were created using the stat_2d_density_filled function in ggplot2 v3.3.5. For RNA velocity analysis using scVelo v0.2.4, the total, spliced and unspliced count matrices, along with the UMAP embeddings, were exported as an h5ad file using anndata v0.7.5.2 for R. The count matrices were filtered and normalized using scv.pp.filter_and_normalize, with min_shared_counts = 20 and

n_top_genes = 2,000. Means and variances between 30 nearest neighbours were calculated in the PCA space (n_pcs = 50, to be consistent with the default value in Seurat). The velocities were calculated using default parameters and projected onto the UMAP embedding exported from Seurat.

## Single-sample gene set enrichment analysis

Single-sample gene set enrichment analysis was applied to scRNA-seq data using the escape package in R[65]. The msigdbr and getGeneSets functions were used to fetch and filter the entire hallmark (H; 50 sets) or the signature cell type (C8; 700 sets) *M. musculus* gene sets from MSigDB[66]. enrichIt with default parameters, except for using 10,000 groups and variable number of cores, was carried out on the Seurat object containing data corresponding to the undifferentiated mesenchyme cells from the *Sox9* regulatory INV mutant, after converting the feature names to gene symbols as necessitated by the escape package. The obtained enrichment scores for each gene set were compared between the two branches (Fig. 5f) using the two-sample Wilcoxon test (wilcox_test) with default parameters and adjusted for multiple comparisons using Bonferroni correction.

## Integration and spatial mapping with sci-space data

We integrated our dataset with a spatial transcriptomics dataset on mid-gestational mice (E14.5), based on the sci-space method[47], in which a subset of transcriptionally profiled nuclei have known physical locations in sagittal sections within which they were mapped before scRNA-seq. We used anchor-based integration as implemented by Seurat for a co-embedding of a subset of MMCA and sci-space. For cells in the subset of MMCA, we find the nearest neighbour in sci-space data in the integrated co-embedding, and plot the location of the neighbouring sci-space cell if it is known.

## Reporting summary

Further information on research design is available in the Nature Portfolio Reporting Summary linked to this article.

## Data availability

The data generated in this study can be downloaded in raw and processed forms from the National Center for Biotechnology Information Gene Expression Omnibus under accession number GSE199308. Other intermediate data files and an interactive app to explore our dataset are freely available via https://atlas.gs.washington.edu/mmca_v2/.

## Code availability

All code is freely available through a public GitHub repository at https://github.com/shendurelab/MMCA.

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

**Acknowledgements** We thank S. Mundlos and C. Prada for helpful discussions around data processing and analysis and interpretation of results; all members of the laboratories of J.C., J.S. and M.S. for continuous support and input; V. Suckow for genotyping the *Ror2*-knock-in and *Cdkl5* –/Y mice; and S. Houghtaling and T.-H. Ho for breeding and embryo collection of *Ttc21b, Carm1* and *Gli2* mice. N. Haag and I.K. thank M. Ebbinghaus for help with breeding of *Scn11a*-GOF mice. J.S. and work in the laboratory of J.S. were supported by the Paul G. Allen Frontiers Foundation (Allen Discovery Center grant to J.S. and C.T.), the National Institutes of Health (NIH; grant UM1HG011531 to J.S.), Alex's Lemonade Stand's Crazy 8 Initiative (to J.S.) and the Bonita and David Brewer Fellowship (C.Q.). Work at the E.O. Lawrence Berkeley National Laboratory was supported by US NIH grants to L.A.P. and A.V. (UM1HG009421 and R01HG003988) and carried out under US Department of Energy Contract DE-AC02-05CH11231, University of California. U.K. was supported by the Deutsche Forschungsgemeinschaft (DFG) (KO 2891/6-1) and ERA-Net for Research on Rare Diseases (EUROGLYCAN-omics). D.M.I. was supported by the Deutsche Forschungsgemeinschaft (DFG) (IB 139/1-1 and IB 139/6-1). I.K. was supported by the Deutsche Forschungsgemeinschaft (DFG) (KU1587/3-1, KU1587/10-1). R.H. was supported by the European Union (ERC, PREVENT, 101078827). J.S. is an Investigator of the Howard Hughes Medical Institute. M.S. is a DZHK principal investigator and is supported by grants from the Deutsche Forschungsgemeinschaft (DFG; SP1532/3-1, SP1532/4-1 and SP1532/5-1) and the Deutsches Zentrum für Luft- und Raumfahrt (DLR 01GM1925). D.R.B. was supported by R01HD36404 from the National Institute of Child Health and Human Development. J.C. is supported by the NIH (grant DP2 HG012522-01 and RM1HG011014) and the Rockefeller University. M.H.d.A. and work at the German Mouse Clinic were supported by the German Federal Ministry of Education and Research (Infrafrontier grant 01KX1012); German Center for Diabetes Research (DZD).

**Author contributions** J.C., M.S. and J.S. conceptualized, supervised and secured funding for the project. D.R.B., W.-L.C., A.D., D.E.D., N. Haag, D.I., I.K., F.H., V.M.K., U.K., L.A.P., S.R., A.R., M.R., A.V., L.W. and Y.Z. provided mouse embryos. J.C. and J.H. extracted the nuclei from embryos and carried out the sciRNA-seq experiment. S.U., R.B., R.H., N. Hansmeier and J.H. carried out the RNAscope experiment and image analysis. M.H.d.A., V.G.-D. and H.F. supervised and coordinated the *Ttc21b* and *Gli2* validation experiment; O.V.A. and P.d.S.-B. carried out embryo staining, O.V.A., P.d.S.-B. and J.H. carried out data analysis and interpretation of the *Ttc21b* and *Gli2* validation experiment. X.H., C.Q., J.H., V.K.A.S. and S.B. carried out all computational analyses. C.M. created the interactive web page with guidance from X.H. and J.S. M.D., L.S., S.R.S. and C.T. provided assistance with data analysis and interpretation of results. X.H., C.Q., J.H. and V.K.A.S. wrote the first draft of the manuscript, which was finalized together with J.C., M.S. and J.S. and input from all authors.

**Funding** Open access funding provided by Universität zu Lübeck.

**Competing interests** J.S. is a scientific advisory board member, consultant and/or co-founder of Cajal Neuroscience, Guardant Health, Maze Therapeutics, Camp4 Therapeutics, Phase Genomics, Adaptive Biotechnologies, Scale Biosciences, Pacific Biosciences, Prime Medicine and Sixth Street Capital. All other authors declare no competing interests.

**Additional information**
**Correspondence and requests for materials** should be addressed to Junyue Cao, Jay Shendure or Malte Spielmann.

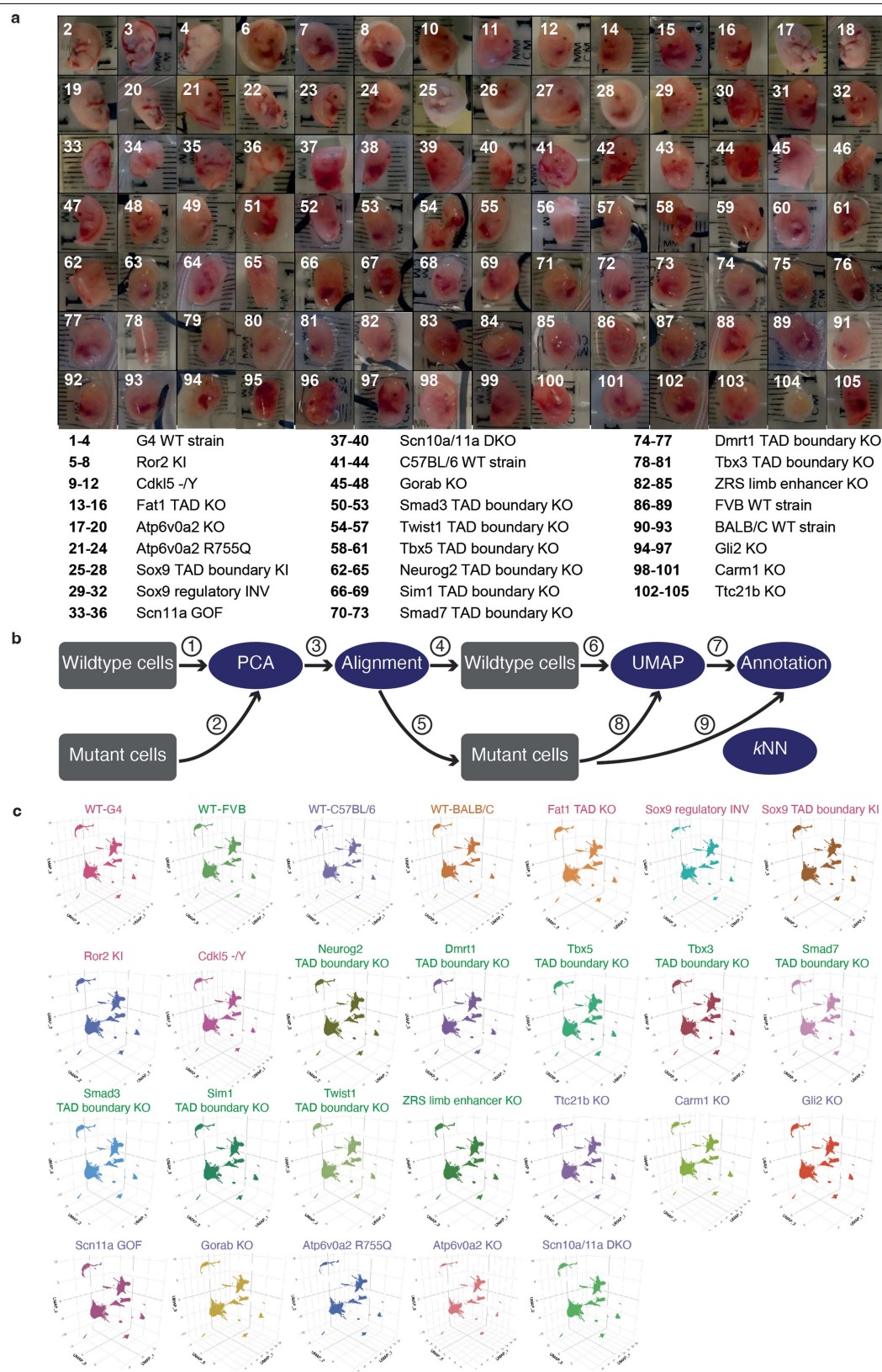

**Extended Data Fig. 1** | See next page for caption.

**Extended Data Fig. 1 | Images of mouse embryos and integrating cells derived from embryos of multiple genetic backgrounds to a single, wildtype-based "reference embedding". a**, 104 embryos (26 genotypes x 4 replicates) were staged at E13.5 and sent by five groups to a single site. #49 was accidentally skipped in our numbering systems. Embryo #70 was lost in transport. Pictures of embryos #1, #5, #9, #13 and #91 were not taken, but the embryos were included in the sci-RNA-seq3 experiment. As discussed in the text, embryos #41 and #104 were labelled as outliers based on computational analyses and their data discarded, while data from the remaining 101 embryos were retained and analysed further. Of note, in addition to the computational analyses suggesting that embryo #104 was an outlier, it was also relatively small in size upon visualisation. **b**, Schematic of approach. We first applied principal components-based dimensionality reduction to cells from wildtype genotypes only (①). We then projected cells from the mutant embryos to this PCA embedding (②). Next, to mitigate potential biases from technical factors, we applied the *align_cds* function in *Monocle*/v3, with the MT%, Ribo%, and log-transformed total UMIs of each cell as covariates (③). We then split wildtype and mutant cells again (④ & ⑤), and applied the UMAP algorithm to wildtype cells only using their "aligned" PC features (⑥), followed by Louvain clustering and manual annotation of individual clusters based on marker gene expression to identify major trajectories, and then iterative clustering and annotation to identity and annotate sub-trajectories (⑦). Finally, cells from mutant embryos were projected to this wildtype-based UMAP embedding, again using their aligned PC features (⑧). Major trajectory labels were assigned to mutant cells via a *k*-nearest neighbour (*k*-NN) heuristic, and these last steps were repeated to further assign sub-trajectory labels to mutant cells (⑨). **c**, 3D UMAP visualisations of cells from each wildtype or mutant background within the shared "reference embedding" resulting from the aforedescribed procedures.

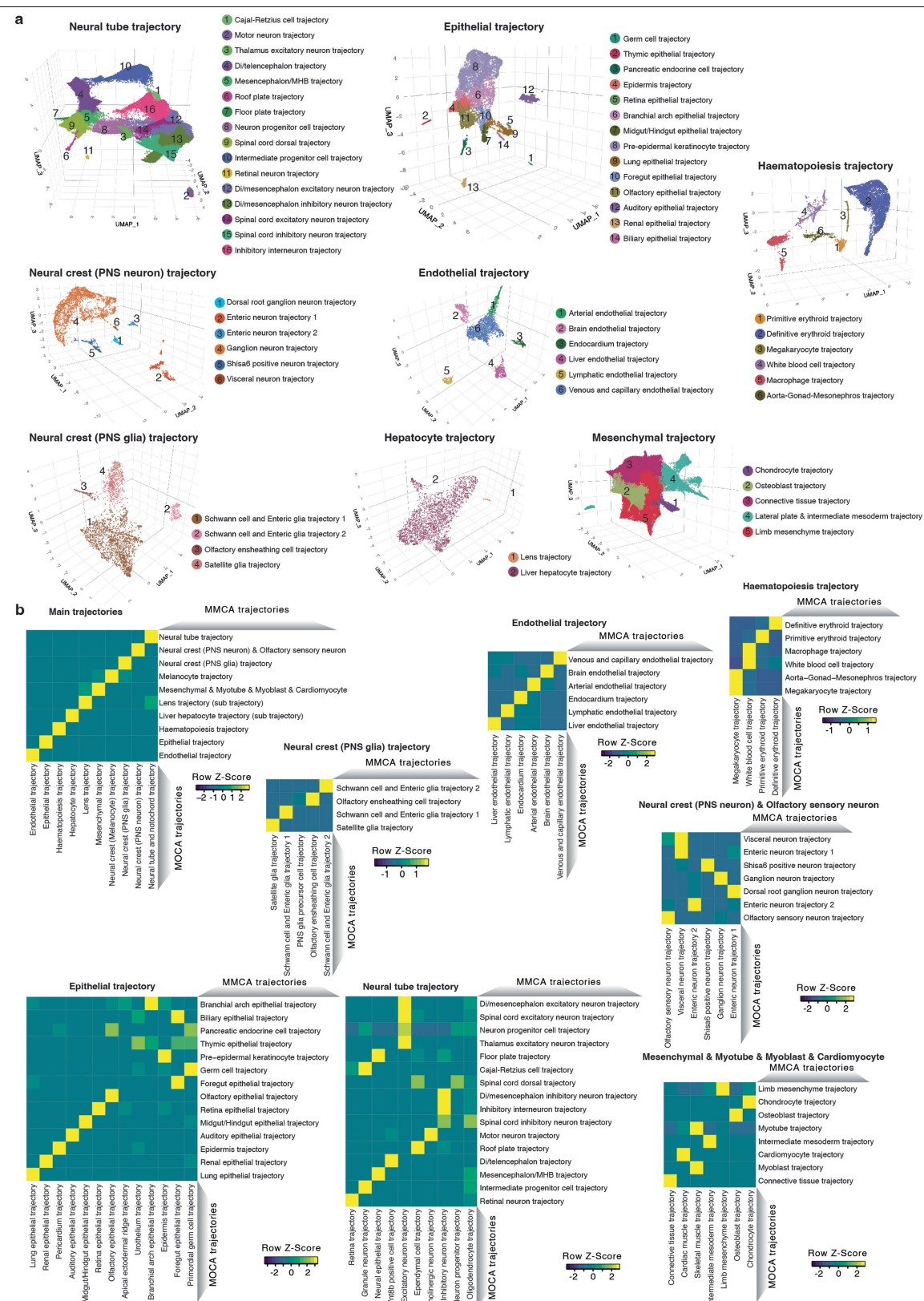

**Extended Data Fig. 2 | Annotation of sub-trajectories in data from wildtype E13.5 embryos and Correlated developmental major and sub-trajectories between MOCA (E9.5 - E13.5) and MMCA (E13.5 only) based on non-negative least-squares (*NNLS*) regression. a**, From 215,517 single cell profiles of wildtype E13.5 embryos of four strains in MMCA, we annotated 13 major trajectories. For 8 of these 13 major trajectories, iterative analysis identified the additional sub-trajectories shown here as 3D UMAP visualisations. Cells are coloured by sub-trajectory annotations. PNS: peripheral nervous system.

MHB: midbrain-hindbrain boundary. Di: Diencephalon. **b**, Shown in the top right is a heat map of the combined regression coefficients (row-scaled) between 10 developmental trajectories from MMCA (rows) and 10 corresponding developmental trajectories from the MOCA (columns). PNS: peripheral nervous system. The other heat maps show the combined regression coefficients (row-scaled) between developmental sub-trajectories from MMCA (rows) and developmental sub-trajectories from the MOCA (columns), within each major trajectory.

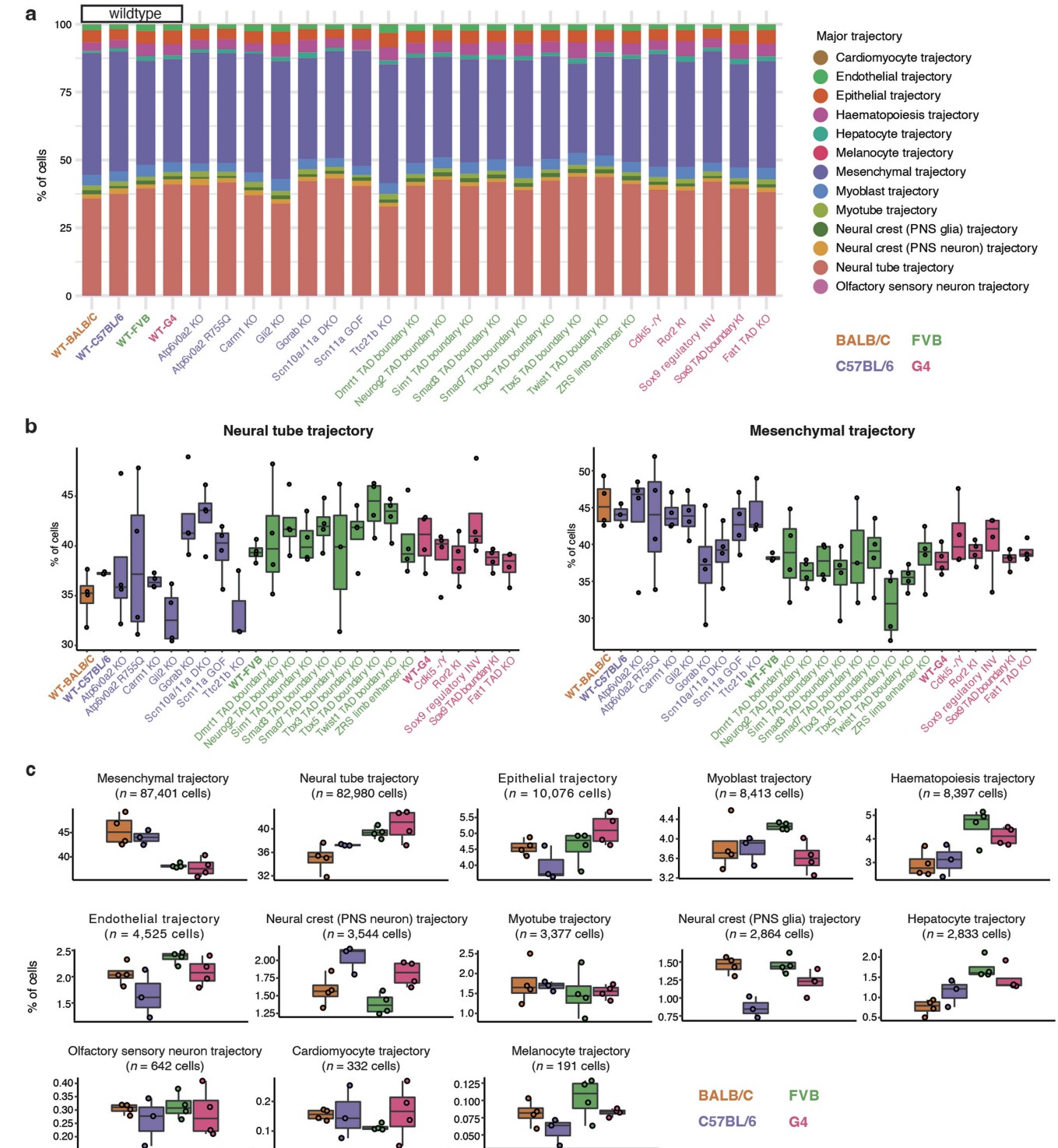

**Extended Data Fig. 3 | Cell composition for individual wildtype and mutant embryos across developmental trajectories. a**, Cell composition across 13 major trajectories of embryos from different wildtype or mutant strains. Cells from all replicates for each strain were pooled for this visualisation. The adjusted p-value by Chi-squared test on cell compositions for individual mutant type and its corresponding genetic background wildtype has been added above. **b**, Boxplots of cell proportions falling into neural tube (left) or mesenchymal (right) trajectories for different wildtype or mutant strains. Points correspond to individual embryos (n = 3 for WT-C57BL/6, n = 4 for all others). **c**, Boxplots of cell proportions falling into each of the 13 major trajectories for the four wildtype strains. Each point corresponds to an individual embryo. The total number of cells from each major trajectory profiled from wildtype embryos and the adjusted p-value by ANOVA (two-sided test) across different backgrounds are also listed. In the boxplots (panels **b** & **c**), the centre lines show the medians; the box limits indicate the 25th and 75th percentiles; the replicates are represented by the dots. PNS: peripheral nervous system.

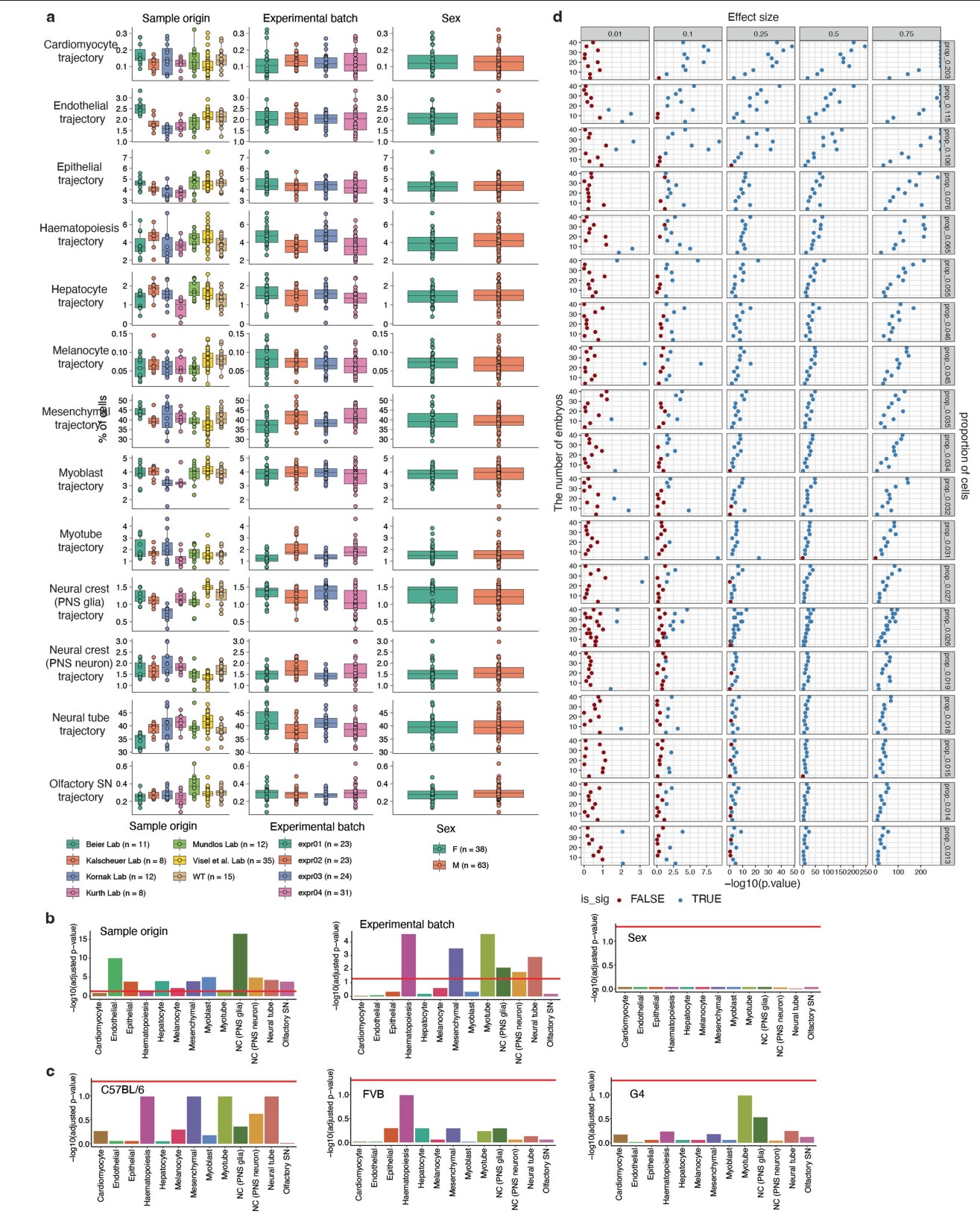

**Extended Data Fig. 4** | See next page for caption.

**Extended Data Fig. 4 | Cell composition for individual wildtype and mutant embryos across developmental trajectories, from different technical or biological groups and Simulation-based estimation of the number of replicates required to detect cell proportion changes. a**, Boxplots of cell proportions falling into each of the 13 major trajectories from different sample origins (left), experimental batch (middle), or sex (right). Each point corresponds to an individual embryo. In the boxplots, the centre lines show the medians; the box limits indicate the 25th and 75th percentiles; the replicates are represented by the dots. **b**, ANOVA (two-sided test) was performed on cell proportions falling into each of the 13 major trajectories from different sample origins (top), experimental batch (middle), or sex (bottom), and the minus log10-scaled adjusted p-values have been shown. The red horizontal line corresponds to significant cutoff (0.05). **c**, ANOVA (two-sided test) was performed on cell proportions falling into each of the 13 major trajectories from different experimental batches after subsetting samples from C57BL/6 (top), FVB (middle), or G4 (bottom), and the log10-scaled adjusted p-values have been shown. The red horizontal line corresponds to significant cutoff (0.05). NC: neural crest. PNS: peripheral nervous system. SN: sensory neuron. **d**, We simulated "wildtype" and "mutant" embryos with parameters drawn from our data (**Methods**), and then performed beta-binomial regression to ask whether cell-type proportions for a given cell type are different between genotypes while varying simulated effect sizes and varying numbers of replicates. In the global view, each column represents a given effect size (*e.g.* 0.01, highlighted on the top) and each row represents a given cell type, with its cell proportion in the whole embryo highlighted at the right. Each single plot represents the testing results of beta-binomial regression for different numbers of replicates of each genotype (y-axis, ranging from 4 to 40). The x-axis refers to -log10 scaled unadjusted p-values, and the dot is coloured either red (insignificant testing result with unadjusted p-value > 0.05) or blue (significant testing result with unadjusted p-value < 0.05).

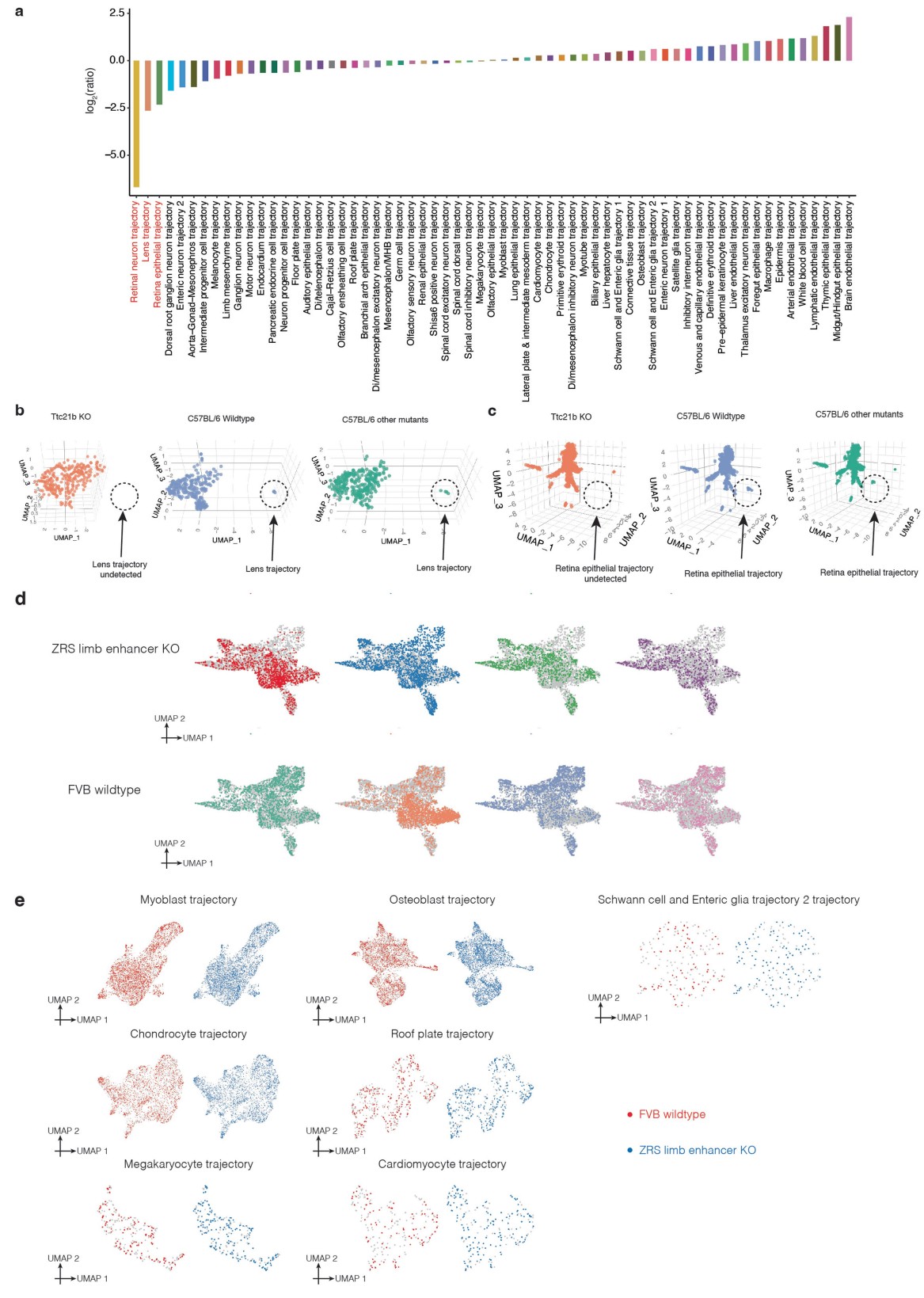

**Extended Data Fig. 5** | See next page for caption.

**Extended Data Fig. 5 | Multiple retinal trajectories are diminished in *Ttc21b* KO mice. a**, The log2 transformed ratio of the cell proportions of each sub-trajectory, comparing *Ttc21b* KO and C57BL/6 wildtype embryos, are shown. Although reductions in the retina epithelial and lens trajectories were excluded from the regression analysis due to their low numbers, they were, together with the retinal neuron trajectory, the most extreme in magnitude. **b**, 3D UMAP visualisation of the hepatocyte major trajectory, highlighting cells from either the *Ttc21b* KO (left), C57BL/6 wildtype (middle), or other mutants on the C57BL/6 background (right). The three plots were randomly downsampled to the same number of cells (*n* = 264 cells) **c**, 3D UMAP visualisation of the epithelial major trajectory, highlighting cells from either the *Ttc21b* KO (left), C57BL/6 wildtype (middle), or other mutants on the C57BL/6 background (right). The three plots were randomly downsampled to the same number of cells (*n* = 937 cells). **d**, UMAP visualisation of co-embedded cells of limb mesenchyme trajectory from the ZRS limb enhancer KO and FVB wildtype. The same UMAP is shown eight times, highlighting cells from either ZRS limb enhancer KO (top row) or FVB wildtype (bottom row), and breaking out the four individual replicates for each strain. **e**, UMAP visualisation of co-embedded cells of various sub-trajectories from the ZRS limb enhancer KO and FVB wildtype. The same UMAP is shown twice for each, highlighting cells from either FVB wildtype (left) or ZRS limb enhancer KO (right). These are the seven sub-trajectories in which, in addition to limb mesenchyme, we detected nominally significant differences in cell type proportions for the ZRS limb enhancer KO.

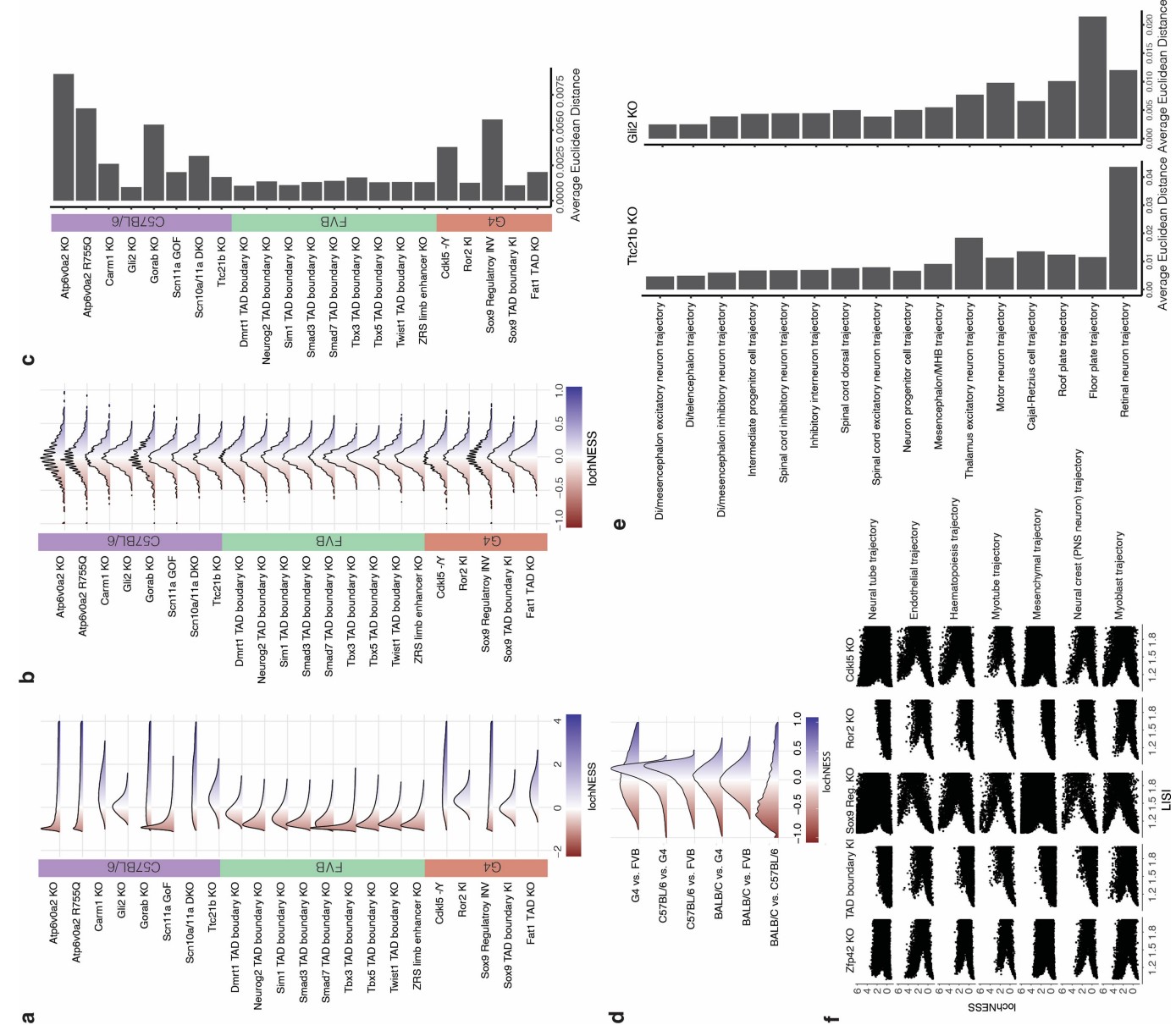

**Extended Data Fig. 6 | Quantitative analysis of lochNESS distributions.**
**a**, Distribution of lochNESS across all 64 sub-trajectories in each mutant.
**b**, Distribution of lochNESS in all cells of each mutant under random permutation of mutant labels. **c**, Barplot showing the average euclidean distance between lochNESS vs. lochNESS under permutation across all cells within a mutant.
**d**, Estimated density graphs of lochNESS shows distribution of lochNESS in wildtype comparisons. Each comparison is labelled by the strain treated as the 'mutant', followed by the strain treated as the reference (i.e. G4 vs. FVB indicates that G4 was treated as the 'mutant' in the comparison). **e**, Barplots showing the average euclidean distance between lochNESS and lochNESS under permutation, across all cells in neural tube sub-trajectories of the *Ttc21b* KO and *Gli2* KO mutants. **f**, Scaterplots showing the concordance of lochNESS and LISI of cells from the G4 mutants in various major trajectories. More extreme lochNESS (indicating separation between mutant and wildtype) is associated with LISI scores approaching one (indicating non-mixing).

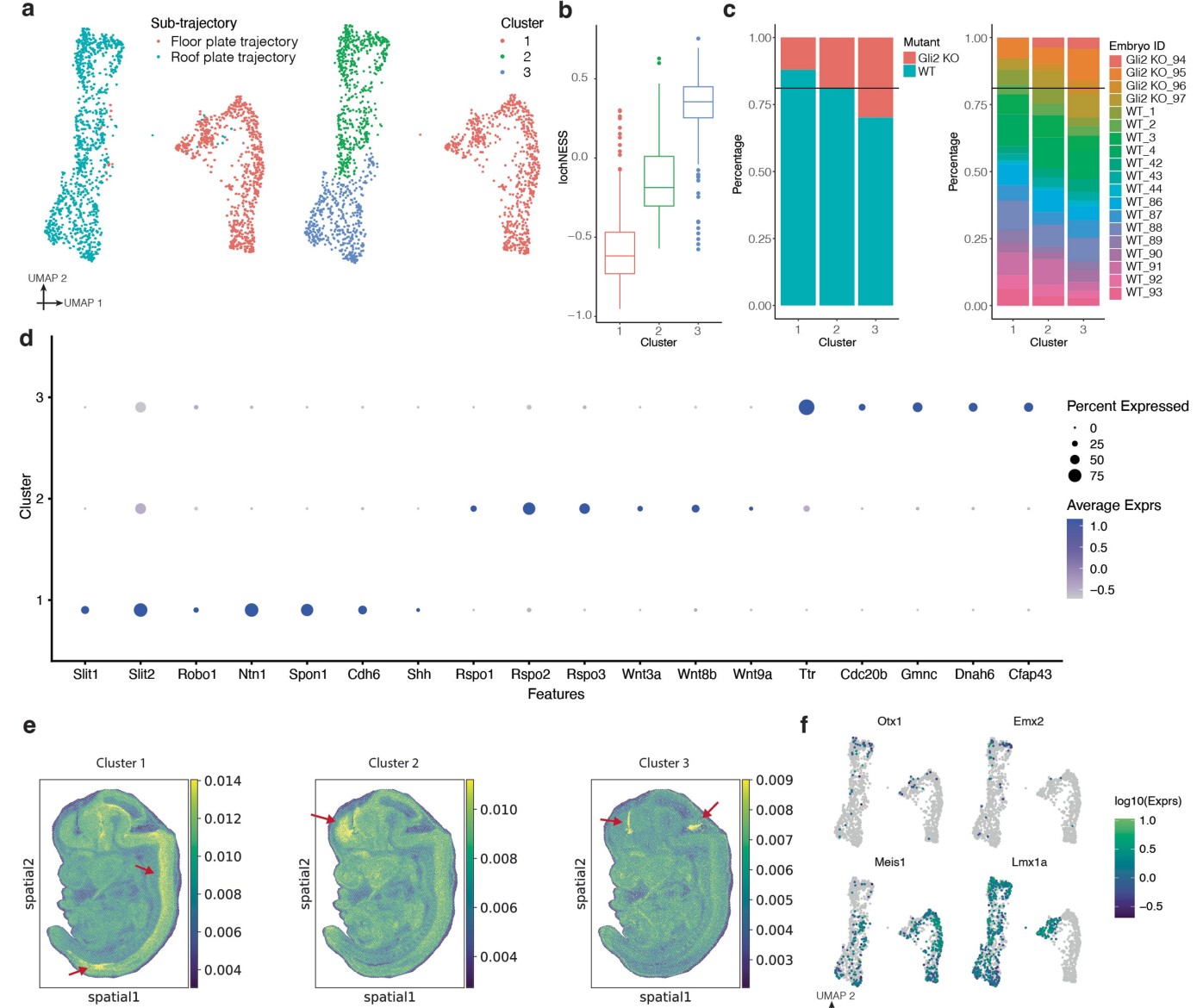

**Extended Data Fig. 7 | Analysis of *Gli2* KO in the roof plate and floor plate trajectories. a**, UMAP visualisation of co-embedded cells of the floor plate and roof plate sub-trajectories from the *Gli2* KO mutant and pooled wildtype, coloured by sub-trajectory (left) or cluster number (right). **b**, Boxplot showing the lochNESS distribution in each cluster shown on the right of panel **a** (n = 717 cells in cluster 1, n = 594 cells in cluster 2, n = 442 cells in cluster 3). Centre lines show medians; box limits indicate 25th and 75th percentiles; outlier individual cells are represented by dots. **c**, Barplots showing the cell composition of each cluster shown on the right of panel **a**, split by mutant vs. wildtype (left) or

individual embryo (right), with a reference line at the overall wildtype cell proportion. **d**, Dotplot summarising the expression of and percent of cells expressing selected marker genes in each cluster shown on the right of panel **a**. **e**, Tangram-inferred locations of each cluster shown on the right of panel **a**. Red arrows highlight the areas where cells map to with high probability. The colour scale is set from 1st percentile to 99th percentile. **f**, UMAP visualisation of co-embedded cells of the floor plate and roof plate sub-trajectories from the *Gli2* KO mutant and pooled wildtype, coloured by expression of marker genes.

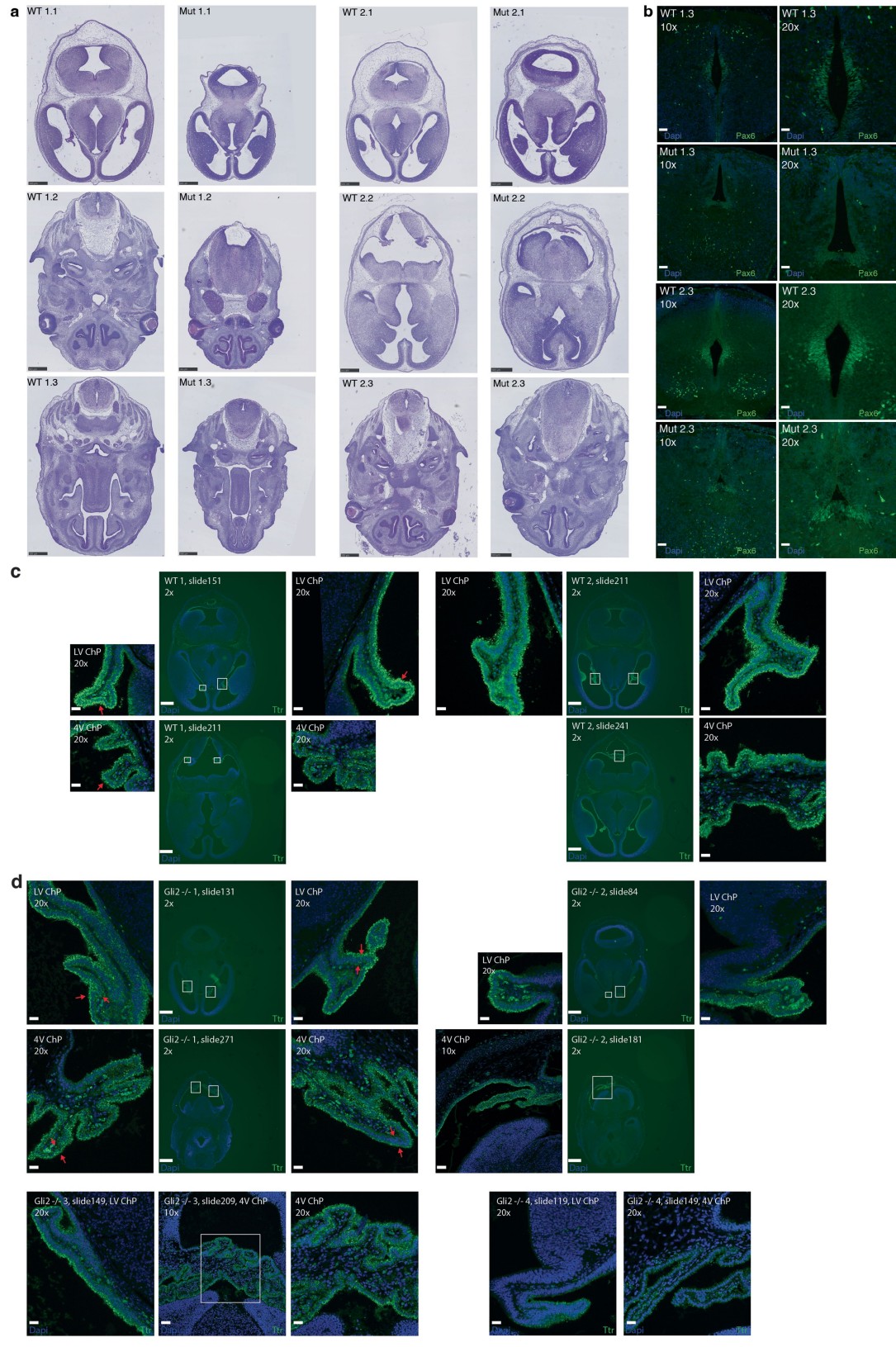

**Extended Data Fig. 8 |** See next page for caption.

**Extended Data Fig. 8 | Morphological phenotype of *Gli2* KO mutants and *Ttr* staining in wildtype mice and *Gli2* KO mutants. a**, H&E staining (**Methods**) of two mutant and two wild type E13.5 embryos in cranial-caudal (1–3) order within the head. In order to compare mutant and wildtype slides in neural tube development, the slides are matched based on hallmarks such as eyes, tongue muscle and nasal cavities (black scale bars correspond to 500 µm). **b**, Neural tube marker *Pax6* staining (**Methods**) of the developing neural tube in consecutive sections 1.3 and 2.3 to visualise the structure of the neural tube formation in wildtype and mutant in 10x and 20x magnification (white scale bars corresponds to 100 and 50 µm). *Ttr* staining (**Methods**) of the developing brain regions (LV = lateral ventricle, 4 V = 4th ventricle, ChP = choroid plexus) in sections of **c**, wildtype and **d**, *Gli2* KO mutants in 2x, 10x and 20x magnification. For each section (2x magnification, white scale bars correspond to 500 µm), the regions of interest are highlighted with white boxes and shown in higher magnification on the sides (10x or 20x magnification, white scale bars correspond to 100 or 50 µm respectively). Red arrows highlight areas with a normal single layer of *Ttr* expressing cells in wildtype, and two layers of cells in the mutant.

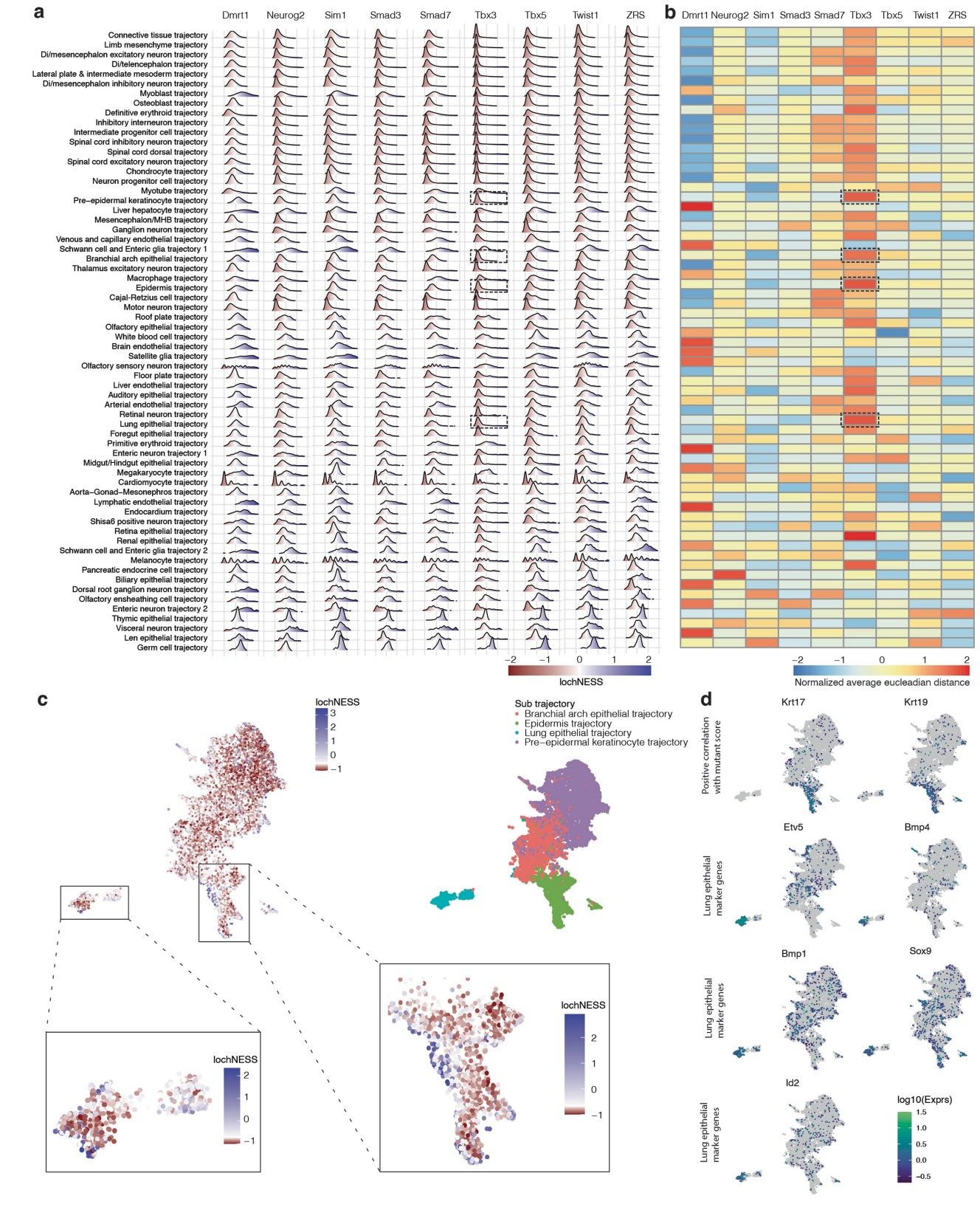

**Extended Data Fig. 9 |** See next page for caption.

**Extended Data Fig. 9 | Systematic screening of lochNESS distributions identifies altered epithelial sub-trajectories in the *Tbx3* TAD Boundary KO mutant. a**, Distribution of lochNESS in each sub-trajectory of the mutants in the FVB background strain, all of which are TAD boundary KOs. Dashed boxes in the sixth column highlight the most deviated epithelial sub-trajectories in the *Tbx3* TAD Boundary KO mutant. **b**, Row-normalised heatmap showing the average euclidean distance between lochNESS and lochNESS under permutation in each sub-trajectory for the same mutants shown in panel **a**, centred and scaled by row. Dashed boxes in the sixth column again highlight the most deviated epithelial sub-trajectories in the *Tbx3* TAD Boundary KO mutant. **c**, UMAP showing co-embedding of *Tbx3* TAD Boundary KO and pooled wildtype cells in the pre–epidermal keratinocyte, epidermis, branchial arch, and lung epithelial sub-trajectories, coloured by lochNESS (top left) [with blown up insets showing lochNESS in lung epithelial (bottom left) and epidermis (bottom right) sub-trajectories], or by sub-trajectory identity (right). LochNESS colour scale is centred at the median of lochNESS. **d**, same as in panel **c**, but coloured by expression of selected mutant related genes and marker genes.

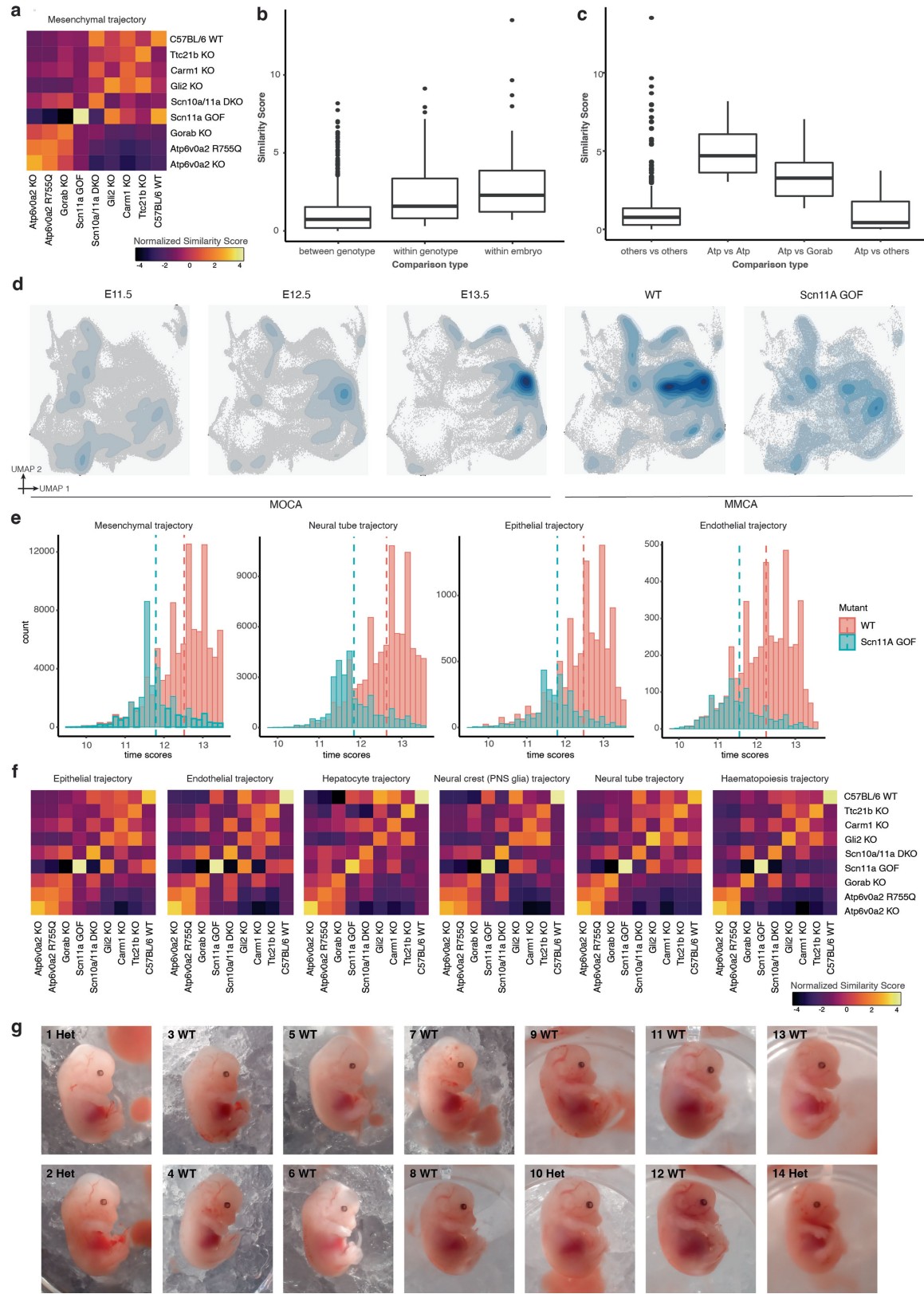

**Extended Data Fig. 10** | See next page for caption.

**Extended Data Fig. 10 | Similarity scores reveal mutant-shared and mutant-specific effects. a**, Heatmap showing similarity scores between C57BL/6 genotypes in the mesenchymal trajectory. **b**, Boxplot showing the similarity scores of comparisons between embryos of different genotypes (left), between embryos of the same genotype (middle), and within the same embryos (right) for C57BL/6 genotypes in the mesenchymal trajectory. **c**, Boxplot showing the similarity scores of comparisons between *Atp6v0a2* KO vs. *Atp6v0a2* R755Q (left), *Atp6v0a2* KO or *Atp6v0a2* R755Q vs. *Gorab* KO (middle), *Atp6v0a2* KO or *Atp6v0a2* R755Q vs. other C57BL/6 genotypes, in the mesenchymal trajectory. Genotype names are simplified in the x-axis legend ("Atp" = *Atp6v0a2* KO or *Atp6v0a2*, "Gorab" = *Gorab* KO, "others" = *Carm1* KO, *Gli2* KO, *Scn10a/11a* DKO, *Scn11a* GOF, *Ttc21b* KO or C57BL/6 wildtype). **d**, UMAPs showing co-embedding of *Scn11a* GOF cells with pooled wildtype cells and E11.5-E13.5 MOCA cells, in the neural tube trajectory, split by mutant (MMCA) and time point (MOCA), with cell density and distributions overlaid. **e**, Barplots showing the distribution of "time scores" for *Scn11a* GOF cells and pooled wildtype cells in the mesenchyme, neural tube, endothelial and epithelial major trajectories, with reference lines at the mean value of time scores. **f**, Heatmaps showing similarity scores between C57BL/6 genotypes in selected major trajectories. *Gorab* KO exhibits high similarity to the two *Atp6v0a2* genotypes in the epithelial, endothelial, hepatocyte and neural crest (PNS glia) trajectories, but not the neural tube and hematopoiesis trajectories. **g**, *Scn11a* mutant and wildtype morphology comparison. Images of 14 E13.5 staged embryos from two litters of wildtype and *Scn11a* heterozygous mutants. Accessible developmental features (limbs, eyes and body size) were compared between the mutants and the wildtype by eye.

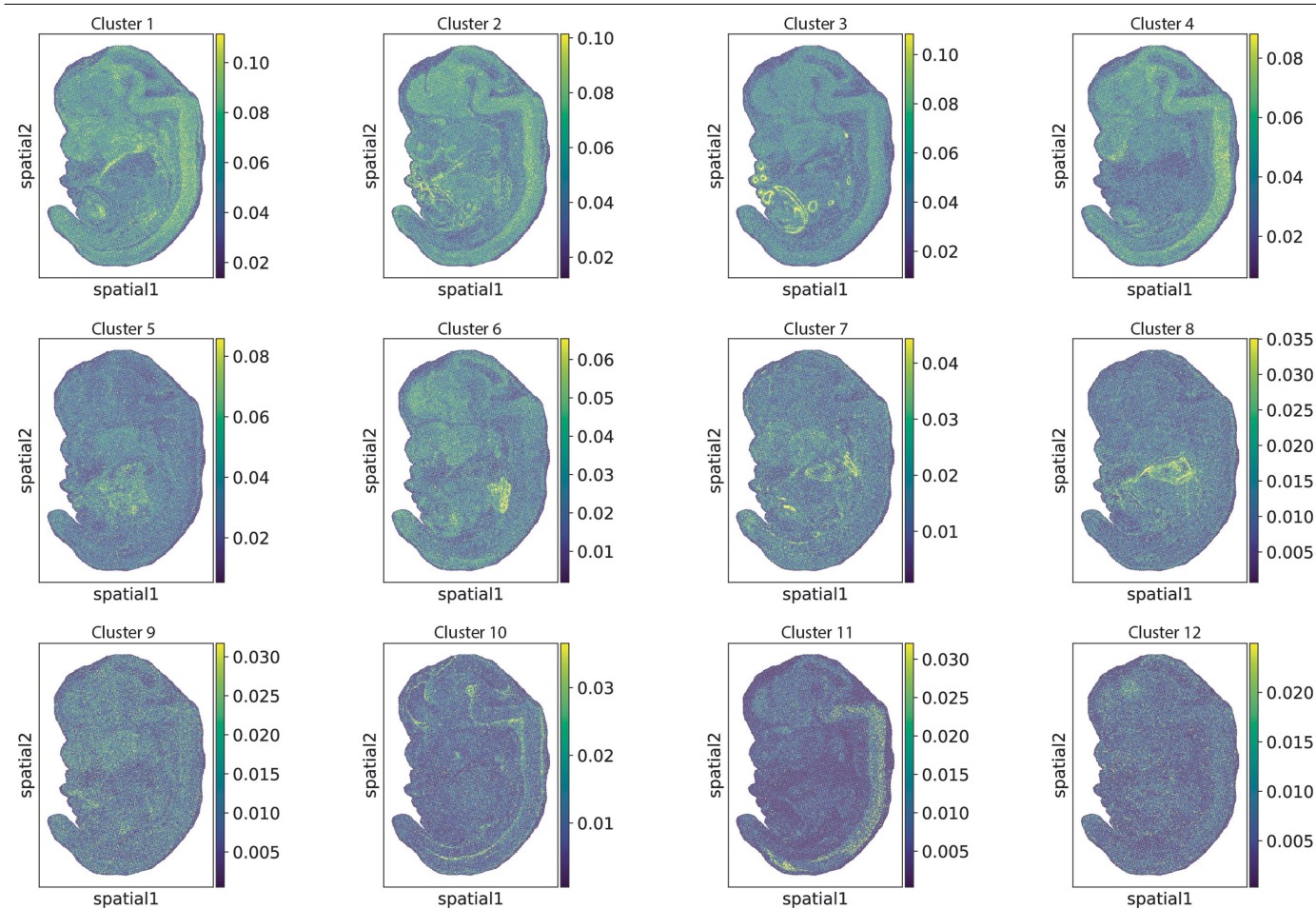

**Extended Data Fig. 11 | Spatial mapping of lateral plate & intermediate mesoderm sub-clusters.** Spatial mapping results by Tangram showing the most likely physical location of the cells from each cluster in the lateral plate & intermediate mesoderm sub-trajectory on a sagittal mouse section. Top 12 sub-clusters are shown. The colour scale is set from 1st percentile to 99th percentile.

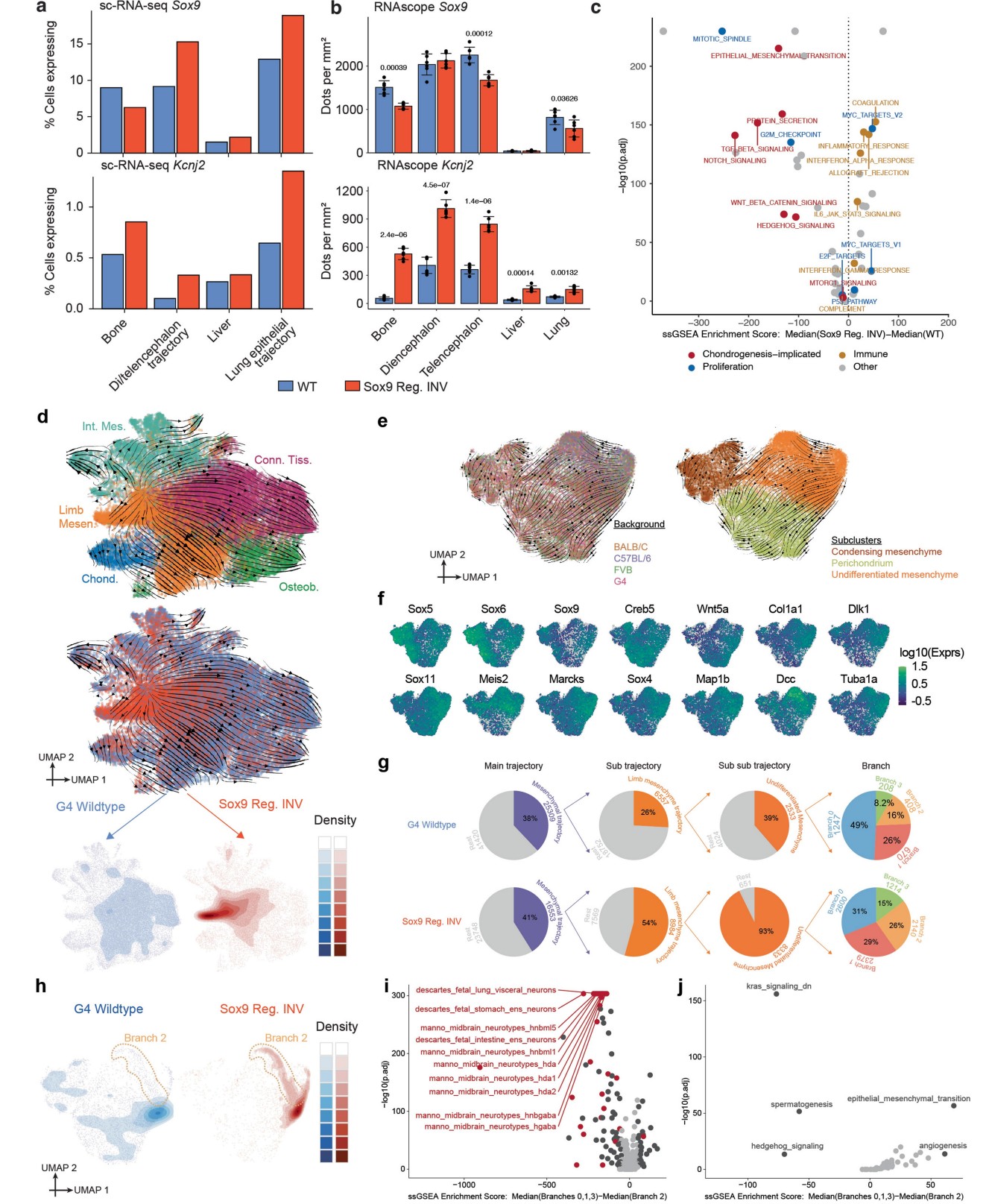

**Extended Data Fig. 12** | See next page for caption.

**Extended Data Fig. 12 | Misregulation of *Sox9* and *Kcnj2*, and stalling of cells in the undifferentiated mesenchyme in the *Sox9* regulatory INV mutant.**

**a**, Quantification of *Sox9* (top row) and *Kcnj2* (bottom row) expression in scRNA-seq data in the wildtype (blue) and *Sox9* regulatory INV (red) genotypes in selected trajectories. For "bone" and "liver", multiple sub-trajectories were pooled to match the tissue labels in the RNAscope data in panel **b**. Specifically, "bone" refers to cells from chondrocyte, osteoblast, and limb mesenchyme trajectories, whereas "liver" refers to cells from the liver endothelial and liver hepatocyte trajectories. The bars represent singular values represented as a fraction of 21128, 6375, 1728 and 229 cells in Bone, Di/telencephalon trajectory, Liver and Lung epithelial trajectories, respectively. **b**, Quantification of *Sox9* and *Kcnj2* expression based on RNAscope images of heterozygous E13.5 wildtype and Sox9 regulatory INV mutant embryos (n = 6 embryos for each condition). The mRNA signal was counted in a defined area ($1 \times 1 \, mm^2$). The numbers represent p-values of the differences between means calculated using a two-sided student t-test. Non-significant p-values (>0.05) not shown. Error bars represent standard deviation. **c**, Gene set enrichment analysis on bone cells. Comparison of median ssGSEA[65] scores between the *Sox9* regulatory INV and wild type for Hallmark gene sets[66]. Gene sets categorised as proliferation and immune-signalling[66] highlighted in blue and brown. Gene sets manually identified to be implicated in chondrogenesis highlighted in red. Note: Bone cells include cells from chondrocyte, osteoblast, and limb mesenchyme trajectories. Many of the hallmark pathways downregulated in the mutant are related to proliferation and chondrocyte differentiation (*e.g.* mitotic spindle, TGF-β signalling, notch signalling, wnt/β-catenin signalling, protein secretion, epithelial-to-mesenchymal transition), which are known to be mediated by *Sox9*. Additionally, six of the seven immune-related hallmark pathways were

upregulated in the mutant, possibly a secondary effect, as to our knowledge *Sox9* is not established to be involved in immune signalling. **d**, RNA velocity of mesenchymal G4 wildtype and *Sox9* regulatory INV cells labelled by sub-trajectories (top) or genotype (middle) and the corresponding 2D density plots split by genotype (bottom). **e**, Sub-clustering of the limb mesenchyme sub-trajectory based on cells from pooled wildtype. RNA velocity arrows generated using scVelo (Methods) indicate the transition of undifferentiated mesenchyme (marked by *Meis2, Marcks, Map1b*) into perichondrium (*Wnt5a, Creb5*) and condensing mesenchyme (*Sox5, Sox6, Sox9*) in all wildtype samples[67–72]. **f**, Marker gene expression used to annotate limb mesenchyme sub-clusters. All except *Dcc* and *Tuba1a* are literature-based markers of the three cell types. Note: Because the annotation of "limb mesenchyme" sub-trajectory was propagated forward from earlier stages of development during the creation of MOCA, it is possible that other, non-limb mesenchymal populations also contribute to this expanded, undifferentiated pool in the *Sox9* regulatory INV embryos. **g**, Proportion and the number of cells at different levels of clustering, leading up to the four branches of the undifferentiated mesenchyme. **h**, Density plots for UMAP embedding of G4 wildtype and *Sox9* regulatory INV cells in the limb mesenchymal trajectory (same embedding as Fig. 5e). Dotted lines highlight Branch 2 of the undifferentiated mesenchyme, based on the sub-clustering shown in Fig. 5f. Comparison of the ssGSEA[65] scores between the two branches of undifferentiated mesenchyme for *Sox9* regulatory INV cells for (**i**) cell type signature (C8) and (**j**) Hallmark gene sets. Gene sets that are both significantly different between the two branches and that have a difference in median ssGSEA scores greater than 50 are highlighted in dark grey, and the most significantly different gene sets are also labelled. In panel **i**, all significantly different gene sets with names containing "neuro" are highlighted in red.

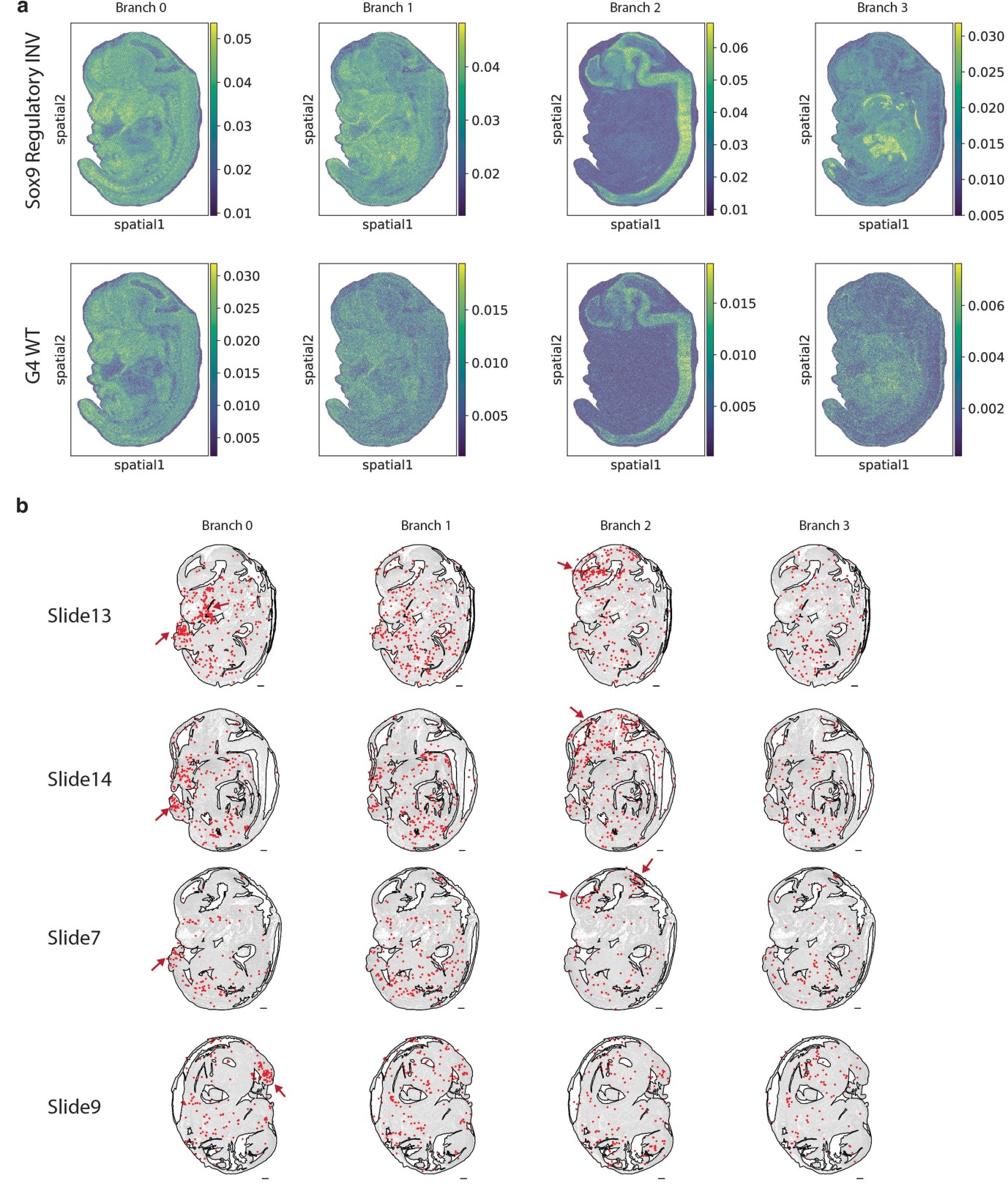

**Extended Data Fig. 13 | Spatial mapping of the cells of undifferentiated mesenchyme onto the Stereo-seq dataset and integration with the sci-space dataset. a**, Tangram inferred locations of cells from each branch shown in Fig. 5f, split by mutant (top) and wildtype (bottom) identity. The colour scale is set from 1st percentile to 99th percentile. **b**, Cells from the *Sox9* regulatory INV mutant assigned to the undifferentiated mesenchyme were integrated with a spatial transcriptomics dataset on mid-gestational mice (E14.5), generated via the sci-space method[47], in which a subset of transcriptionally profiled nuclei have known physical locations in sagittal sections within which they were mapped prior to scRNA-seq. We find the nearest neighbour of each *Sox9* regulatory INV mutant cell in sci-space data in the integrated co-embedding, and plot the location of the neighbouring sci-space cell where it is known (red dots). Red arrows highlight areas with aggregated cells (branch 0 matches with the limbs and branch 2 matches to the brain).

# Reporting Summary

## Statistics

For all statistical analyses, confirm that the following items are present in the figure legend, table legend, main text, or Methods section.

| n/a | Confirmed | |
|---|---|---|
| ☐ | ☒ | The exact sample size ($n$) for each experimental group/condition, given as a discrete number and unit of measurement |
| ☐ | ☒ | A statement on whether measurements were taken from distinct samples or whether the same sample was measured repeatedly |
| ☐ | ☒ | The statistical test(s) used AND whether they are one- or two-sided <br> *Only common tests should be described solely by name; describe more complex techniques in the Methods section.* |
| ☐ | ☒ | A description of all covariates tested |
| ☐ | ☒ | A description of any assumptions or corrections, such as tests of normality and adjustment for multiple comparisons |
| ☐ | ☒ | A full description of the statistical parameters including central tendency (e.g. means) or other basic estimates (e.g. regression coefficient) AND variation (e.g. standard deviation) or associated estimates of uncertainty (e.g. confidence intervals) |
| ☐ | ☒ | For null hypothesis testing, the test statistic (e.g. $F$, $t$, $r$) with confidence intervals, effect sizes, degrees of freedom and $P$ value noted <br> *Give P values as exact values whenever suitable.* |
| ☒ | ☐ | For Bayesian analysis, information on the choice of priors and Markov chain Monte Carlo settings |
| ☒ | ☐ | For hierarchical and complex designs, identification of the appropriate level for tests and full reporting of outcomes |
| ☒ | ☐ | Estimates of effect sizes (e.g. Cohen's $d$, Pearson's $r$), indicating how they were calculated |

*Our web collection on statistics for biologists contains articles on many of the points above.*

## Software and code

Policy information about availability of computer code

| Data collection | No software other than Illumina RTA basecalling was used in data collection. |
|---|---|
| Data analysis | The publicly available software used in the paper are described in the Methods section and are cited. These include: sci-RNA-seq3 processing pipeline (https://github.com/JunyueC/sci-RNA-seq3_pipeline), bcl2fastq/v2.20, Fiji/v2.13, deML/v1.1 (https://github.com/grenaud/deML), trim_galore/v0.6.5, STAR/v2.6.1d, python/v2.7.13, scrublet/v0.1, Seurat/v3, Monocle/v3, uwot/v0.1.8, Seurat/v4.0.6, ggplot2/v3.3.5, anndata/v0.7.5.2, escape/v1.6.0, scVelo/v0.2.4, Tangram/v1.0.4. <br> The code developed for the paper is made freely available through a public GitHub repository at https://github.com/shendurelab/MMCA. |

For manuscripts utilizing custom algorithms or software that are central to the research but not yet described in published literature, software must be made available to editors and reviewers. We strongly encourage code deposition in a community repository (e.g. GitHub). See the Nature Portfolio guidelines for submitting code & software for further information.

## Data

Policy information about availability of data

All manuscripts must include a data availability statement. This statement should provide the following information, where applicable:
- Accession codes, unique identifiers, or web links for publicly available datasets
- A description of any restrictions on data availability
- For clinical datasets or third party data, please ensure that the statement adheres to our policy

The data generated in this study can be downloaded in raw and processed forms from the NCBI Gene Expression Omnibus under accession number GSE199308. Other intermediate data files and an interactive app to explore our dataset is made freely available via https://atlas.gs.washington.edu/mmca_v2/.

# Field-specific reporting

Please select the one below that is the best fit for your research. If you are not sure, read the appropriate sections before making your selection.

☒ Life sciences          ☐ Behavioural & social sciences          ☐ Ecological, evolutionary & environmental sciences

For a reference copy of the document with all sections, see nature.com/documents/nr-reporting-summary-flat.pdf

# Life sciences study design

All studies must disclose on these points even when the disclosure is negative.

| | |
|---|---|
| Sample size | We did not perform prior explicit calculations for sample size. Sample size was determined by availability of embryos from the crossings. We sampled either 50/50 female and male embryos (C57BL/6, BALB/C, FVB) or male only (G4) from the respective genotypes. We included 4 replicates per genotype of mutants and corresponding wildtypes at the embryonic stage E13.5 in the study. For the validation studies, in H&E staining for the Ttc21b mutant we used  wildtype=2,  Ttc21b heterozygous=2, Ttc21b homozygous=4, for Gli2 (Pax and Ttr AB staining) we used Gli2-/- homozygous mutant=4 and wildtype=2 and for Sox9Inv mutant RNAscope homozygous=3 and wildtype=3 embryos. |
| Data exclusions | We excluded the embryos 104 and 41 after the quality control step of the analysis from downstream analysis for reasons of low cell number and/or quality of the sample. Sample Nr. 70 was lost in transport prior to the start of the experiment. |
| Replication | For the sci-RNA-seq3 experiment we isolated nuclei from 103 embryos staged E 13.5, 4 replicates each genotype including 22 mutant backgrounds and the corresponding 4 WT backgrounds. The attempts at replication were successful. For the validation studies, in H&E staining for the Ttc21b mutant we used  wildtype=2,  Ttc21b heterozygous=2, Ttc21b homozygous=4, for Gli2 (Pax and Ttr AB staining) we used Gli2-/- homozygous mutant=4 and wildtype=2 and for Sox9Inv mutant RNAscope homozygous=3 and wildtype=3 embryos. |
| Randomization | To minimize batch effects, the nuclei extraction from embryos was randomized. For the first round of indexing, nuclei from each embryo were deposited in seperate wells respectively, such that the first index could be linked to the individual embryos isolated from. After the first round of indexing, all samples were pooled and distributed randomly across four plates for the second indexing round. |
| Blinding | Investigators were blinded to group allocation during data collection and analysis: Embryo collection, nuclei isolation, library preparation and sci-RNA-seq3 analysis all were performed by different researchers, respectively. |

# Behavioural & social sciences study design

All studies must disclose on these points even when the disclosure is negative.

| | |
|---|---|
| Study description | *Briefly describe the study type including whether data are quantitative, qualitative, or mixed-methods (e.g. qualitative cross-sectional, quantitative experimental, mixed-methods case study).* |
| Research sample | *State the research sample (e.g. Harvard university undergraduates, villagers in rural India) and provide relevant demographic information (e.g. age, sex) and indicate whether the sample is representative. Provide a rationale for the study sample chosen. For studies involving existing datasets, please describe the dataset and source.* |
| Sampling strategy | *Describe the sampling procedure (e.g. random, snowball, stratified, convenience). Describe the statistical methods that were used to predetermine sample size OR if no sample-size calculation was performed, describe how sample sizes were chosen and provide a rationale for why these sample sizes are sufficient. For qualitative data, please indicate whether data saturation was considered, and what criteria were used to decide that no further sampling was needed.* |
| Data collection | *Provide details about the data collection procedure, including the instruments or devices used to record the data (e.g. pen and paper, computer, eye tracker, video or audio equipment) whether anyone was present besides the participant(s) and the researcher, and whether the researcher was blind to experimental condition and/or the study hypothesis during data collection.* |
| Timing | *Indicate the start and stop dates of data collection. If there is a gap between collection periods, state the dates for each sample cohort.* |
| Data exclusions | *If no data were excluded from the analyses, state so OR if data were excluded, provide the exact number of exclusions and the rationale behind them, indicating whether exclusion criteria were pre-established.* |
| Non-participation | *State how many participants dropped out/declined participation and the reason(s) given OR provide response rate OR state that no participants dropped out/declined participation.* |
| Randomization | *If participants were not allocated into experimental groups, state so OR describe how participants were allocated to groups, and if allocation was not random, describe how covariates were controlled.* |

# Ecological, evolutionary & environmental sciences study design

All studies must disclose on these points even when the disclosure is negative.

| | |
|---|---|
| Study description | *Briefly describe the study. For quantitative data include treatment factors and interactions, design structure (e.g. factorial, nested, hierarchical), nature and number of experimental units and replicates.* |
| Research sample | *Describe the research sample (e.g. a group of tagged Passer domesticus, all Stenocereus thurberi within Organ Pipe Cactus National Monument), and provide a rationale for the sample choice. When relevant, describe the organism taxa, source, sex, age range and any manipulations. State what population the sample is meant to represent when applicable. For studies involving existing datasets, describe the data and its source.* |
| Sampling strategy | *Note the sampling procedure. Describe the statistical methods that were used to predetermine sample size OR if no sample-size calculation was performed, describe how sample sizes were chosen and provide a rationale for why these sample sizes are sufficient.* |
| Data collection | *Describe the data collection procedure, including who recorded the data and how.* |
| Timing and spatial scale | *Indicate the start and stop dates of data collection, noting the frequency and periodicity of sampling and providing a rationale for these choices. If there is a gap between collection periods, state the dates for each sample cohort. Specify the spatial scale from which the data are taken* |
| Data exclusions | *If no data were excluded from the analyses, state so OR if data were excluded, describe the exclusions and the rationale behind them, indicating whether exclusion criteria were pre-established.* |
| Reproducibility | *Describe the measures taken to verify the reproducibility of experimental findings. For each experiment, note whether any attempts to repeat the experiment failed OR state that all attempts to repeat the experiment were successful.* |
| Randomization | *Describe how samples/organisms/participants were allocated into groups. If allocation was not random, describe how covariates were controlled. If this is not relevant to your study, explain why.* |
| Blinding | *Describe the extent of blinding used during data acquisition and analysis. If blinding was not possible, describe why OR explain why blinding was not relevant to your study.* |

Did the study involve field work?  ☐ Yes  ☐ No

## Field work, collection and transport

| | |
|---|---|
| Field conditions | *Describe the study conditions for field work, providing relevant parameters (e.g. temperature, rainfall).* |
| Location | *State the location of the sampling or experiment, providing relevant parameters (e.g. latitude and longitude, elevation, water depth).* |
| Access & import/export | *Describe the efforts you have made to access habitats and to collect and import/export your samples in a responsible manner and in compliance with local, national and international laws, noting any permits that were obtained (give the name of the issuing authority, the date of issue, and any identifying information).* |
| Disturbance | *Describe any disturbance caused by the study and how it was minimized.* |

# Reporting for specific materials, systems and methods

We require information from authors about some types of materials, experimental systems and methods used in many studies. Here, indicate whether each material, system or method listed is relevant to your study. If you are not sure if a list item applies to your research, read the appropriate section before selecting a response.

### Materials & experimental systems

| n/a | Involved in the study |
|---|---|
| ☐ | ☒ Antibodies |
| ☒ | ☐ Eukaryotic cell lines |
| ☒ | ☐ Palaeontology and archaeology |
| ☐ | ☒ Animals and other organisms |
| ☒ | ☐ Human research participants |
| ☒ | ☐ Clinical data |
| ☒ | ☐ Dual use research of concern |

### Methods

| n/a | Involved in the study |
|---|---|
| ☒ | ☐ ChIP-seq |
| ☒ | ☐ Flow cytometry |
| ☒ | ☐ MRI-based neuroimaging |

# Antibodies

| | |
|---|---|
| Antibodies used | Mm-Kcnj2 (Cat. No. 476261, Advanced Cell Diagnostics [ACD], Newark, CA, USA)<br>Mm-Sox9 (Cat. No. 401051-C2, Advanced Cell Diagnostics [ACD], Newark, CA, USA)<br>Prealbumin (Ttr) Antibody ( Cat. No. ab215202, [EPR20971], Abcam)<br>Pax6 Antibody ( Cat. No. AB2237, Merck-Sigma)<br>Goat Anti-Rabbit Alexa Fluor 488-conjugated secondary antibody (Leica, A-11008) |
| Validation | The RNA scope probes (Mm-Kcnj2, Mm-Sox9) were not further validated for this study.<br>The antibodies Pax6 and Prealbumin were tested in different concentrations on the wildtype embryos using DAB (3,3'-Diaminobenzidin) detection and compared it to literature to ensure specificity of the antibodies. Validation of Antibodies Pax and Prealbumin was proceeded with a standardized DAB (3,3'-Diaminobenzidine) validation on adult mouse brain tissues prior to test for specificity of the AB's.  DAB staining was followed up with validations using immunofluorescence on adult tissue until proper dilutions were found. |

# Eukaryotic cell lines

Policy information about cell lines

| | |
|---|---|
| Cell line source(s) | *State the source of each cell line used.* |
| Authentication | *Describe the authentication procedures for each cell line used OR declare that none of the cell lines used were authenticated.* |
| Mycoplasma contamination | *Confirm that all cell lines tested negative for mycoplasma contamination OR describe the results of the testing for mycoplasma contamination OR declare that the cell lines were not tested for mycoplasma contamination.* |
| Commonly misidentified lines<br>(See ICLAC register) | *Name any commonly misidentified cell lines used in the study and provide a rationale for their use.* |

# Palaeontology and Archaeology

| | |
|---|---|
| Specimen provenance | *Provide provenance information for specimens and describe permits that were obtained for the work (including the name of the issuing authority, the date of issue, and any identifying information). Permits should encompass collection and, where applicable, export.* |
| Specimen deposition | *Indicate where the specimens have been deposited to permit free access by other researchers.* |
| Dating methods | *If new dates are provided, describe how they were obtained (e.g. collection, storage, sample pretreatment and measurement), where they were obtained (i.e. lab name), the calibration program and the protocol for quality assurance OR state that no new dates are provided.* |

☐ Tick this box to confirm that the raw and calibrated dates are available in the paper or in Supplementary Information.

| | |
|---|---|
| Ethics oversight | *Identify the organization(s) that approved or provided guidance on the study protocol, OR state that no ethical approval or guidance was required and explain why not.* |

Note that full information on the approval of the study protocol must also be provided in the manuscript.

# Animals and other organisms

Policy information about studies involving animals; ARRIVE guidelines recommended for reporting animal research

| | |
|---|---|
| Laboratory animals | We included mouse mutant and wildtype embryos from the commonly used laboratory strains C57BL/6, BALBC, G4 and FVB at embryonic stage E13.5. We sampled either 50/50 female and male embryos (C57BL/6, BALB/C, FVB) or male only (G4) from the respective strains. |
| Wild animals | This study did not include wild animals. |
| Field-collected samples | This study did not include field-collected samples. |
| Ethics oversight | All animal procedures were conducted as approved by the local authorities (LAGeSo Berlin) under license numbers G0243/18 and G0176/19. All animal experiments followed relevant guidelines and regulations. |

Note that full information on the approval of the study protocol must also be provided in the manuscript.

# Human research participants

Policy information about studies involving human research participants

| | |
|---|---|
| Population characteristics | *Describe the covariate-relevant population characteristics of the human research participants (e.g. age, gender, genotypic* |

| Population characteristics | *information, past and current diagnosis and treatment categories). If you filled out the behavioural & social sciences study design questions and have nothing to add here, write "See above."* |
|---|---|
| Recruitment | *Describe how participants were recruited. Outline any potential self-selection bias or other biases that may be present and how these are likely to impact results.* |
| Ethics oversight | *Identify the organization(s) that approved the study protocol.* |

Note that full information on the approval of the study protocol must also be provided in the manuscript.

# Clinical data

Policy information about clinical studies

All manuscripts should comply with the ICMJE guidelines for publication of clinical research and a completed CONSORT checklist must be included with all submissions.

| Clinical trial registration | *Provide the trial registration number from ClinicalTrials.gov or an equivalent agency.* |
|---|---|
| Study protocol | *Note where the full trial protocol can be accessed OR if not available, explain why.* |
| Data collection | *Describe the settings and locales of data collection, noting the time periods of recruitment and data collection.* |
| Outcomes | *Describe how you pre-defined primary and secondary outcome measures and how you assessed these measures.* |

# Dual use research of concern

Policy information about dual use research of concern

## Hazards

Could the accidental, deliberate or reckless misuse of agents or technologies generated in the work, or the application of information presented in the manuscript, pose a threat to:

No | Yes
- ☒ ☐ Public health
- ☒ ☐ National security
- ☒ ☐ Crops and/or livestock
- ☒ ☐ Ecosystems
- ☒ ☐ Any other significant area

## Experiments of concern

Does the work involve any of these experiments of concern:

No | Yes
- ☒ ☐ Demonstrate how to render a vaccine ineffective
- ☒ ☐ Confer resistance to therapeutically useful antibiotics or antiviral agents
- ☒ ☐ Enhance the virulence of a pathogen or render a nonpathogen virulent
- ☒ ☐ Increase transmissibility of a pathogen
- ☒ ☐ Alter the host range of a pathogen
- ☒ ☐ Enable evasion of diagnostic/detection modalities
- ☒ ☐ Enable the weaponization of a biological agent or toxin
- ☒ ☐ Any other potentially harmful combination of experiments and agents

# ChIP-seq

## Data deposition

☐ Confirm that both raw and final processed data have been deposited in a public database such as GEO.

☐ Confirm that you have deposited or provided access to graph files (e.g. BED files) for the called peaks.

| Data access links *May remain private before publication.* | *For "Initial submission" or "Revised version" documents, provide reviewer access links.  For your "Final submission" document, provide a link to the deposited data.* |
|---|---|
| Files in database submission | *Provide a list of all files available in the database submission.* |

| Genome browser session (e.g. UCSC) | *Provide a link to an anonymized genome browser session for "Initial submission" and "Revised version" documents only, to enable peer review. Write "no longer applicable" for "Final submission" documents.* |

## Methodology

| Replicates | *Describe the experimental replicates, specifying number, type and replicate agreement.* |
| Sequencing depth | *Describe the sequencing depth for each experiment, providing the total number of reads, uniquely mapped reads, length of reads and whether they were paired- or single-end.* |
| Antibodies | *Describe the antibodies used for the ChIP-seq experiments; as applicable, provide supplier name, catalog number, clone name, and lot number.* |
| Peak calling parameters | *Specify the command line program and parameters used for read mapping and peak calling, including the ChIP, control and index files used.* |
| Data quality | *Describe the methods used to ensure data quality in full detail, including how many peaks are at FDR 5% and above 5-fold enrichment.* |
| Software | *Describe the software used to collect and analyze the ChIP-seq data. For custom code that has been deposited into a community repository, provide accession details.* |

# Flow Cytometry

## Plots

Confirm that:

☐ The axis labels state the marker and fluorochrome used (e.g. CD4-FITC).

☐ The axis scales are clearly visible. Include numbers along axes only for bottom left plot of group (a 'group' is an analysis of identical markers).

☐ All plots are contour plots with outliers or pseudocolor plots.

☐ A numerical value for number of cells or percentage (with statistics) is provided.

## Methodology

| Sample preparation | *Describe the sample preparation, detailing the biological source of the cells and any tissue processing steps used.* |
| Instrument | *Identify the instrument used for data collection, specifying make and model number.* |
| Software | *Describe the software used to collect and analyze the flow cytometry data. For custom code that has been deposited into a community repository, provide accession details.* |
| Cell population abundance | *Describe the abundance of the relevant cell populations within post-sort fractions, providing details on the purity of the samples and how it was determined.* |
| Gating strategy | *Describe the gating strategy used for all relevant experiments, specifying the preliminary FSC/SSC gates of the starting cell population, indicating where boundaries between "positive" and "negative" staining cell populations are defined.* |

☐ Tick this box to confirm that a figure exemplifying the gating strategy is provided in the Supplementary Information.

# Magnetic resonance imaging

## Experimental design

| Design type | *Indicate task or resting state; event-related or block design.* |
| Design specifications | *Specify the number of blocks, trials or experimental units per session and/or subject, and specify the length of each trial or block (if trials are blocked) and interval between trials.* |
| Behavioral performance measures | *State number and/or type of variables recorded (e.g. correct button press, response time) and what statistics were used to establish that the subjects were performing the task as expected (e.g. mean, range, and/or standard deviation across subjects).* |

## Acquisition

**Imaging type(s)**
*Specify: functional, structural, diffusion, perfusion.*

**Field strength**
*Specify in Tesla*

**Sequence & imaging parameters**
*Specify the pulse sequence type (gradient echo, spin echo, etc.), imaging type (EPI, spiral, etc.), field of view, matrix size, slice thickness, orientation and TE/TR/flip angle.*

**Area of acquisition**
*State whether a whole brain scan was used OR define the area of acquisition, describing how the region was determined.*

**Diffusion MRI**  ☐ Used   ☐ Not used

## Preprocessing

**Preprocessing software**
*Provide detail on software version and revision number and on specific parameters (model/functions, brain extraction, segmentation, smoothing kernel size, etc.).*

**Normalization**
*If data were normalized/standardized, describe the approach(es): specify linear or non-linear and define image types used for transformation OR indicate that data were not normalized and explain rationale for lack of normalization.*

**Normalization template**
*Describe the template used for normalization/transformation, specifying subject space or group standardized space (e.g. original Talairach, MNI305, ICBM152) OR indicate that the data were not normalized.*

**Noise and artifact removal**
*Describe your procedure(s) for artifact and structured noise removal, specifying motion parameters, tissue signals and physiological signals (heart rate, respiration).*

**Volume censoring**
*Define your software and/or method and criteria for volume censoring, and state the extent of such censoring.*

## Statistical modeling & inference

**Model type and settings**
*Specify type (mass univariate, multivariate, RSA, predictive, etc.) and describe essential details of the model at the first and second levels (e.g. fixed, random or mixed effects; drift or auto-correlation).*

**Effect(s) tested**
*Define precise effect in terms of the task or stimulus conditions instead of psychological concepts and indicate whether ANOVA or factorial designs were used.*

**Specify type of analysis:**  ☐ Whole brain   ☐ ROI-based   ☐ Both

**Statistic type for inference**
(See Eklund et al. 2016)
*Specify voxel-wise or cluster-wise and report all relevant parameters for cluster-wise methods.*

**Correction**
*Describe the type of correction and how it is obtained for multiple comparisons (e.g. FWE, FDR, permutation or Monte Carlo).*

## Models & analysis

| n/a | Involved in the study |
|---|---|
| ☐ | ☐ Functional and/or effective connectivity |
| ☐ | ☐ Graph analysis |
| ☐ | ☐ Multivariate modeling or predictive analysis |

**Functional and/or effective connectivity**
*Report the measures of dependence used and the model details (e.g. Pearson correlation, partial correlation, mutual information).*

**Graph analysis**
*Report the dependent variable and connectivity measure, specifying weighted graph or binarized graph, subject- or group-level, and the global and/or node summaries used (e.g. clustering coefficient, efficiency, etc.).*

**Multivariate modeling and predictive analysis**
*Specify independent variables, features extraction and dimension reduction, model, training and evaluation metrics.*

