## [Peer Review File · Nature]

Manuscript Title: Single cell, whole embryo phenotyping of mammalian developmental disorders

Reviewer Comments & Author Rebuttals

Reviewer Reports on the Initial Version:

Referees' comments:

Referee #1 (Remarks to the Author):

Manuscript 2022-04-06723 Single cell, whole embryo phenotyping of pleiotropic disorders of mammalian development

The authors report the application of scRNA-seq profiling to uncover phenotypes within the developing mouse embryo. The concept is quite straightforward, to obtain comprehensive whole embryo profiles at mid-organogenesis stages (e13.5), then apply clustering to identify cell diversity and then a variety of computational approaches to look for differences in the proportions or expression profiles amongst cell types.

General overview

The major criticism of the study is the all-important choice of “test cases” to examine. This represents a very mixed bag of genetic variants clearly of interest to different members of the investigative team but arguably too mixed, with two few solid predictions to really put the strategy to the test. One might have imagined a stronger list of well analyzed mutants demonstrating the approach detected known subtle phenotypes, and potentially undetected phenotypes that could be well-validated in secondary assays. This would have provided a stronger sense of the power of single cell expression profiling for phenotypic discovery. Generally, the paper lacks critical secondary validation of predictions made from analysis of the single cell data. This is clearly a general weakness throughout. Not a criticism as such, but if this is to become a tool of broader utility and not just one for the genomic center, computational and large budget privileged, the authors should be encouraged to explore how feasible the approach is now, or might reasonably be in the future, for the average scientist.

Specific comments in order within the manuscript

- Commenting on Figure 1b, the authors indicate that bulk-seq, that is simply mRNA-seq of an entire embryo is not a great strategy for phenotyping embryos. I think people realize this. I am not aware of anyone suggesting this as a reasonable approach?
- Given the pseudo-bulk data in Figure 1 emphasizes strain differences and mouse mutants may be on a variety of complex backgrounds, does this mean that the investigator must always generate a de novo reference embedded library?
- Can the authors comment on the absence of expected trajectories in Figure 1e – why no gut, kidney, lung, skin? Does this represent under-sequencing, technical issues of proportional cell representation in the data or operator selectivity which would be counter to the idea of whole

embryo phenotypic profiling? Understanding and providing a circumspect perspective on the limits as well as the successes would help balance the manuscript.

- In supplementary Figure 5 the authors note differences in cell numbers for individual tissue trajectories. Given substantial inter-embryo variability, is there a good sense of the number of replicates that would be required then to put these trends onto solid statistical grounds? The word “nominally” appears to be a qualifier to the significance of observed differences at a few points in the text.

- The observation that eye and lens contributions are reduced in *Ttc21b* correlating with an observed eye phenotype at 11.5 could be strengthened further by aligning the anatomical analysis with the e13.5 single cell RNA-seq data set. Some simple clear histology at e13.5 should be up to the task.

- What is the lesson to be learned from the ZRS mutant? In this case, this is a control with a well-documented limb phenotype not evident in first pass analysis requiring a tailored secondary analysis to reveal a limb phenotype. Does this mean in primary screening of an embryo of unknown phenotype, quite significant limb (and other) phenotypes would go unnoticed? The ZRS mutant is a macroscopic phenotype I believe (see line 309). The lochNESS approach looks promising in this example (Figure 3b). One assumes no false negative enrichments in other populations?

- The *Gli2* mutant floorplate gene expression data result is odd. Was *Shh* expression lost as expected in the *Gli2* mutant cells? This would be perhaps the most obvious gene to investigate. The roof plate results are unexpected given no obvious connection of *Gli2* with the roof-plate and no evidence of Hedgehog pathway regulation in this region. However, Hedgehog transcriptional regulation is complex and the loss of a *Gli2* repressor form could have some effect? Potentially, there is a strong proof-of-principle that novel phenotypes can be uncovered but this would require a significant analysis of the *Gli2* mutant to demonstrated there really are changes within the roof-plate population. It is not sufficient to report the unexpected without a rigorous validation when rolling out a new methodology. Ditto epidermal *Tbx3*. For *Scn11a*, the authors suggestion that *Scn11a* mutants are developmentally earlier makes sense. What does not make sense is why the authors believe there is a significance to this outcome and the mutation rather than the possibility that this represents expected variability in collection of litters. Again, if we are to accept a possibility that modifying pain reception has a general growth phenotype delaying development at e13.5, there should be some independent validation of additional *Scn11a* mutants (and littermates). No secondary validation of predictions for *Atp6v0a2*.

- As in the previous comments, no secondary validation.

- With respect to the interpretation of the *Sox9* phenotype, the proposed neural-like cell population is interesting and unexpected, again, a finding that could benefit from a simple follow up secondary validation directly on embryos.

- The sharing amongst a rather eclectic set of largely unrelated mutants in the final section (supplementary Figures 14-16) is indeed surprising as the authors note. It seems more likely this represents a technical issue than uncovering “sharing of molecular and cellular sub-phenotypes across mutants. Again, the authors can strengthen confidence in the approach and move tentative conclusions to more solid ground by investing more in validating predictions from the data.

-

Referee #2 (Remarks to the Author):

Huang and colleagues have undertaken the impressive challenge of comprehensively mapping embryonic phenotypes in whole embryos by leveraging tools from high throughput single-cell RNA sequencing (sci-RNA-seq), mouse genetics, and both established and newly developed single-cell analytical methods. The collaborative study combines expertise from several groups and aims to develop a standardized framework for unbiased interpretation of mammalian developmental defects, including pleiotropic defects. To accomplish this task, the authors collected and analyzed single-cell RNA seq profiles from 1.6M nuclei spanning 101 E13.5 mouse embryos, 26 genotypes, and 4 mouse strains. The authors have analyzed this massive dataset using several interrelated paradigms: (1) quantification of changes in cell type composition between genotypes, (2) quantification of wt/mutant ratio changes across cell state neighborhood tcat spanning the full range of detected embryonic cell types/tissues, and (3) systematic quantification of shared features between genotypes. Among the 26 genotypes analyzed, the authors highlight examples of phenotypes in which specific tissues fail to differentiate (Ttc21b, ZRS limb enhancer), and one case that involves the generation of a novel mutant-specific state (Sox9 INV). The study is both ambitious and well written. However, in its present form, the study has some serious weaknesses that would need to be fully addressed before publication could be considered. Most importantly, the authors themselves acknowledge that some of the most prominent trends in their MMCA analysis (Fig 2a) might be artifacts driven by a skewed wildtype baseline comparison. In addition, there are some important unresolved questions about the interpretability of the lochNESS scores. Given that both potential artifacts and ambiguity surround some of the central claims of this study, it will be imperative for authors to address these concerns by performing additional wildtype baseline comparisons and benchmarks of existing or new single-cell data (see below) and/or provide more extensive in vivo validation of claims that are not yet convincingly supported by their scRNA-seq data alone.

Major comments:

In Supplemental Figure 5, there appear to be wide ranges in the recovery efficiencies for each cell type across both genotype and strain(s). Given how heavily the authors' MMCA approach relies on the interpretation of cell type proportions, it would be helpful if the authors exhaustively examined any other possible sources of technical effects or biological effects that might confound cell type recovery rates. Similarly to Supplemental Figure 5c, can the authors also break down the recovery rates of each trajectory by lab of origin/ cryopreservation protocol, experimental batch, etc?

Main Fig 2a. Some cell types are responding in a stereotyped fashion regardless of perturbation, usually all going in the same direction with respect to the strain-matched wt control. For example, the Brain Endothelial Trajectory increases in abundance following any of the genetic perturbations performed in the C57BL/6 background, but in none of the perturbations from the other strains. A similar trend exists for Aorta-Gonad-Mesonephros and Endocardium but in the opposite direction, with these cell types nearly always decreasing in abundance. Could the authors rule out the stated possibility that these signatures are simply an artifact (e.g. driven by the cell type recovery rates in the C57BL/6 control to which all of these perturbations are compared), by (1) comparing to additional controls, and (2) providing in vivo validation of these phenotypes? These strain-specific effects are arguably the most pronounced trends presented by the authors in Fig 2a. If the reader

cannot be convinced that the most striking results from this analysis are not technical artifacts, the authors will not succeed in establishing MMCA as a reliable or interpretable method. If proper controls do ultimately suggest that these trends indeed reflect genuine biology, it would be interesting to investigate further how such stereotyped responses arise from such varied perturbations, e.g. do they reflect stress, developmental delay, or some other generic response to developmental mispatterning?

The authors' lochNESS analyses are a bit confusing and could benefit from further benchmarks / explanation. First, an internal check using wildtype samples (e.g. comparing between different wildtypes) would provide the reader with a better intuition on what a "remarkable" vs an "unremarkable" score distribution should look like. My expectation was that an unremarkable / null distribution should center on a score of zero, much like the random permuted data in Supplemental Fig 8a. The authors should demonstrate this explicitly by constructing histograms for wildtype vs wildtype comparisons and determine whether the resulting distributions of lochNESS scores center on zero, or some other value. According to the authors' interpretation, the TAD boundary knockouts are largely "unremarkable" but in Fig 3c (and Supplemental Fig 9a), many of these histograms appear to skew heavily towards negative lochNESS scores. Based on the how the lochNESS scores are calculated and the logic presented in Fig 3a, negative lochNESS scores should indicate that substantial numbers of wildtype-like cell states are not being populated in these mutants, much like the case for the ZRS limb enhancer KO in which a key *msx/hox* state is depleted (Fig 3b, Fig 2d-e). By this logic, the TAD KO samples would all be displaying similar lochNESS trends as the ZRS enhancer KO, suggesting that these perturbations in fact similarly eliminate some cell states. Can the authors explain this discrepancy? Are the TAD KO perturbations depleting specific cell types, or not? If not, where are these negative lochNESS scores coming from? If negative lochNESS scores are, in fact, unreliable or less reliable than positive scores, the authors should explain this clearly to the reader, ideally with data-driven benchmarks.

The authors' interpretation of the *Gli2* KO phenotype, particularly the differing effects on "two subpopulations of roofplate cells", should be followed up in further detail. The authors have not established that this second mutant-enriched cluster is indeed roofplate. Especially given the relatively shallow UMI-per cell sampling depth of the authors' sci-Seq method, this could simply be an unrelated cell type that shares superficial transcriptional similarity to the roofplate. The authors should perform *in vivo* validations with additional marker gene *in situ* hybridizations that distinguish the two candidate roofplate clusters in both wildtype and mutant embryos, to verify cell type identities, anatomical locations, and opposite responses to the *Gli2* KO.

Main Fig 3g - this figure plots similarity scores calculated between embryos of the same and different genotypes. The resulting scoring matrix appears to be asymmetric (e.g. compare the rows vs the columns associated with the *Scn11a* GOF samples), and could lead to different conclusions depending on how the matrix is evaluated. Can the authors explain how the similarity metric calculation can return different values between the same pairs of samples, and which direction is being used for their particular claims (e.g. rows or columns), and why? Same question for Supplemental Figs 10a and 10f. In all of these types of heatmaps, the authors should validate their assertions of genotype similarity scores (both the *Scn11a* and the *Atpv0a2/Gorab* claims) by performing hierarchical clustering of rows vs columns and confirming similarity trends in the

resulting dendrograms. In present form, it doesn't look like these heatmaps have been subjected to clustering (if they have, details and dendrograms should be provided).

The analysis of phenotypic similarities between different gene KOs (the topic of manuscript pages 16-17) does not mention the striking similarities at the level cell type abundances for Brain Endothelial, Aorta, Endocardium that were already noted in Fig 2a. If the possibility of artifacts in Fig 2a can be indeed ruled out, it might be worth emphasizing this point again, as it could reinforce the trends being asserted by the authors' kNN-based approach.

Referee #3 (Remarks to the Author):

In this manuscript, Huang, Henck, Qiu and colleagues present a mouse mutant cell atlas, MMCA, a collection of just over 1.6 million cell nuclei profiles from a combination of 101 mouse embryos at E13.5 across 22 mouse mutants and four wildtype genetic backgrounds. Combined with the previous work profiling wildtype mouse embryos across E9.5-E13.5, the authors perform a characterization of the relative staging and cell type trajectories present in the MMCA.

A large-scale enrichment analysis is performed to understand the relative enrichment and depletion of cells belonging to trajectories in accordance with the mutant or wildtype condition. Then, the authors extract a global differential abundance scoring, named LochNESS, for which they are able to characterise local differences in cell abundance according to the experimental condition (e.g. wildtype versus mutant). Using this scoring, the authors assess the similarity and differences of mutants in terms of their local cell distributions, identifying instances of distinct and importantly similar cell abundance phenotypes across different mutants. Some further examination and independent validation of abundance changes are made, particularly for the *Ttc21b* KO, and an elegant followup understanding the effect of the *Sox9* regulatory inversion mutant.

Overall, I think this manuscript is quite promising and represents a viable path towards larger scale cellular phenotyping of multiple genetic perturbations. However, I have some key concerns regarding the experiment and analysis, detailed in point form below.

- LochNESS is a score that reflects the relative mixing of cells coming from the mutant and wildtype conditions, and noted by the authors that is similar in nature to the Milo method presented by Dann et al. It appears that LochNESS does not use replication information in the calculation, thereby losing some ability to detect changes in abundance attributed to the experimental conditions rather than variation within conditions. LochNESS is a score returned per cell reflecting local mixing, and therefore is quite similar in nature to the LISI (Local Inverse Simpson Index) described and implemented by DOI 10.1038/s41592-019-0619-0. It would be worthwhile to assess the correlation of LochNESS scores with LISI, especially under varying neighbourhood size parameter choices.

- How is it confirmed that the embryo samples corresponded to the listed mutants? There is mention of the absence of the distal limb structures in the ZRS distal enhancer mutant, but is this extended to the other 21 mutant types? e.g. lower expression observed of the genes in certain cell types?

- What is the rationale for the mutations on the various genetic backgrounds? e.g. why is Fat1 TAD KO performed on a G4 background and not on the other three, some explanation or discussion of this would be helpful.
- Further to the above, it would be worthwhile to identify the background/perturbation interaction effect for at least some of the perturbations and background combinations, i.e. how consistent are the cell abundance and profile phenotypes when set against a different background.
- Observing "Block-like" structures along the genetic background in Figure 2a are concerning, especially since these are observed not only for B6 but also to some extent for FVB in the MHB and enteric neuron trajectory, and for G4 in auditory epithelial trajectory. Without performing additional perturbations against other backgrounds (e.g. a Ttc21b KO in FVB or G4 and not only in B6), it's difficult to disentangle this confounding. It may also be that strain effect could be masking some real biological enrichment or depletion of trajectories. I would suggest a follow-up check with a subset of perturbation combinations against different backgrounds as an experimental validation.
- Further to the above, it seems this manuscript is missing a comprehensive comparison between the wildtype genetic backgrounds. The authors could capitalise on the replicated experimental design to statistically assess if variation in cell distribution across trajectories could be explained by background, and if so this should be discussed in terms of implication for understanding the mutant conditions.
- Figure 3e-f: Examining differential gene expression among cells that are embedded within a low-dimensional space defined by gene expression can be fraught, given the notion of "double-dipping" (e.g. see DOI 10.1016/j.cels.2019.07.012). From the biological standpoint it may also be that the Gli2 KO reduces the capacity for cells fated towards the floor plate trajectory, therefore displaying no differential expression signal only differential abundance signal. While not something I see as required for this manuscript, an ideal exploration of this would be to incorporate a timecourse component to the assessment of the differential genes along the trajectory for the mutant, compared to the wildtype as captured already e.g. in Cao et al 2019.
- The similarity score between various genotypes to determine mutation-specific and mutant-shared is not clearly described and hard to follow. It would be good for authors to include clear description in the Methods for this score calculation, and what it represents.
- Overall how sure are authors that cells are accurately assigned to the experimental condition (mutant or wildtype)? What quality metrics are used in this case and how many "cells" are found to be ambiguous/misassigned and therefore removed?
- It would be highly desirable, in my opinion, to integrate the dissociated data generated in this study to the spatially-resolved wildtype mouse data captured by the sci-Space (10.1126/science.abb9536 from this manuscript's senior authors' labs) and stereo-seq (10.1016/j.cell.2022.04.003) and examine the relative enrichment or depletion of mutants in terms of the physical locations in the mouse embryo.

Minor

Figure 1d - more useful ordering boxplots by median cell number, typo Dmrt1 "boudary" which appears throughout manuscript and web app

Figure 1e and elsewhere - 3D UMAP is not particularly visual here, either only show 2D or include grids as in Figure 1b

Supp Figure 5a - add a statistical measure of departure in the cell type distribution, e.g. chi-square test of the perturbation condition and it's corresponding genetic background WT.

Figure 2a - barplot on top more useful if presented on a log-scale.

Figure 4c - axes needed

Author Rebuttals to Initial Comments:

We thank the three reviewers for their very constructive feedback, which we have sought to address as outlined below. Below, reviewer comments are replicated in full in dark blue text, with our responses provided inline in black text.

- Referee #1: pages 1 - 27
 - Referee #2: pages 28 - 44
 - Referee #3: pages 45 - 58
-

Referee #1 (Remarks to the Author):

Manuscript 2022-04-06723 Single cell, whole embryo phenotyping of pleiotropic disorders of mammalian development

The authors report the application of scRNA-seq profiling to uncover phenotypes within the developing mouse embryo. The concept is quite straightforward, to obtain comprehensive whole embryo profiles at mid-organogenesis stages (e13.5), then apply clustering to identify cell diversity and then a variety of computational approaches to look for differences in the proportions or expression profiles amongst cell types.

General overview

1-0: The major criticism of the study is the all-important choice of “test cases” to examine. This represents a very mixed bag of genetic variants clearly of interest to different members of the investigative team but arguably too mixed, with two few solid predictions to really put the strategy to the test. One might have imagined a stronger list of well analyzed mutants demonstrating the approach detected known subtle phenotypes, and potentially undetected phenotypes that could be well-validated in secondary assays. This would have provided a stronger sense of the power of single cell expression profiling for phenotypic discovery. Generally, the paper lacks critical secondary validation of predictions made from analysis of the single cell data. This is clearly a general weakness throughout. Not a criticism as such, but if this is to become a tool of broader utility and not just one for the genomic center, computational and large budget privileged, the authors should be encouraged to explore how feasible the approach is now, or might reasonably be in the future, for the average scientist.

Thank you for the insightful comments. There are three main points raised in this comment: one about the choice of mutants, another about utility outside of a genome centre, and a third about secondary validation. We address these in turn.

Choice of mutants: Our decision to go with a mixed bag was intentional. Specifically, we sought to test the technology under real life/lab conditions similar to a mouse clinic where a broad phenotypic spectrum can be expected. For a single centre, it is almost impossible to generate so many mutants in one location. Therefore, we collected mutants from three major locations: The Max Planck Institute (Berlin), Seattle Children’s Research Institute (Seattle), and the Lawrence Berkeley National Laboratory (Berkeley). The main objective was to understand whether whole embryo sc-RNA-seq is powerful enough to detect changes in mutant phenotypes ranging from severe (affecting many organ systems) to very mild (such as noncoding changes likely affecting only a subset of cells). Moreover, using mutants from different backgrounds allowed us to investigate whether there are strain-specific differences.

Taking these factors together, mutants were chosen as described in the original manuscript (lines 115-127):

"We grouped the 22 mutants, all homozygous but the *Scn11a*GOF mutant which is heterozygous, into four categories (**Supplementary Table 1**): 1) pleiotropic mutants, representing knockouts of developmental genes expressed in multiple organs (*Ttc21b* KO, *Carm1* KO, *Gli2* KO), as well as two mutations of the *Sox9* regulatory landscape suspected to have pleiotropic effects, both of which effectively result in the introduction of a boundary element between endogenous *Sox9* enhancers and the *Sox9* promoter (*Sox9* TAD boundary KI; *Sox9* regulatory INV)¹⁻⁴. 2) developmental disorder mutants, intended to model specific human diseases (*Scn11a* GOF, *Ror2* KI, *Gorab* KO, *Cdk15* -Y)⁵⁻⁷, 3) mutations of loci associated with human disease (*Scn10a/Scn11a* DKO, *Atp6v0a2* KO, *Atp6v0a2* R755Q, *Fat1*TAD KO)^{8,9}. 4) prospective deletions of cis-regulatory elements, including of TAD boundaries in the vicinity of developmental transcription factors including *Smad3*, *Twist1*, *Tbx5*, *Neurog2*, *Sim1*, *Smad7*, *Dmrt1*, *Tbx3*, and *Twist1*¹⁰, and, as a positive control, the ZRS distal enhancer (Zone of polarising activity Regulatory Sequences) which regulates sonic hedgehog (SHH) expression and results in absent distal limb structures¹¹."

Thus, although there is not a tightly-defined biological theme, this was by design, and there is a clear rationale that supports our choices. We argue that this choice allows the reader to draw more generalised conclusions about the contexts in which the strategy will be useful (e.g. as opposed to if we had only chosen very-large-effect mutations in a single signalling pathway, which would have been a very different study). Further, while we were reasonably confident that mutants with well-characterised developmental defects would show robust differences by sc-RNA-seq, we did not and could not know the sensitivity of the approach for more subtle perturbations; hence, this was worth testing.

In order to better visually communicate the four categories, we have extensively revised **Fig. 1a**:

Figure 1a. We applied sci-RNA-seq3 to profile 1.6M single cell transcriptomes from 103 individual E13.5 embryos, derived from 22 mutants and four wildtype strains, in one experiment.

Utility outside of a genome centre: We would point out that our sci- protocol is not only meant for big genomic centres but quite the opposite: the single cell experiments were performed by one grad student at dramatically lower costs than commercial single cell approaches. Furthermore, most bioinformatic analyses were performed by two grad students and we are making our computational tools publicly available to reduce the amount of "new work" for investigators adopting this strategy. It is true that our labs have substantial resources, but we are not a major genome centre. The primary dataset was generated in a few experiments, and sequenced across two NovaSeq runs. So yes, expensive, but even at the scale presented, still within the reach of many labs in a field where a rapidly growing number of trainees have the necessary analytical skills. Studies like ours spanning multiple mutants and different backgrounds will also help labs identify contexts in which this strategy is more vs.

less likely to work, and to more efficiently design their studies. Furthermore, with ongoing drops in sequencing costs, this approach stands to gain further traction for unbiased phenotyping. The major bottleneck is still animal breeding and handling (and not single cell data generation and analysis).

Secondary validation. In their comments below, Reviewer #1 requested secondary validations related to *Ttc21b* KO, *Gli2* KO, *Scn11a* GOF, *Tbx3* TAD boundary KO, *Atp6v0a2/Gorab*, and *Sox9* regulatory INV mutants. We were able to complete about half of the requested validations, specifically those related to the *Ttc21b* KO, *Gli2* KO, and *Scn11a* GOF mutants. The *Ttc21b* KO and *Gli2* KO validations in particular provide clear support of specific findings made through whole embryo sc-RNA-seq. We also indirectly validate the ‘neural-like branch’ of mesenchymal cells enriched in the *Sox9* regulatory INV mutant through the re-analysis of several publicly available spatial transcriptomic datasets.

We were not able to perform the other validations mainly due to limitations of traditional validation experiments in relation to the specific observations made (*Atp6v0a2/Gorab*), the subtlety of the changes (*Tbx3* TAD boundary KO) and/or access to mice (*Sox9* regulatory INV; although as noted above in this case, it was addressed through re-analysis of published spatial transcriptomic datasets). Some of these phenotypes have never been described before due to the lack of resolution of conventional phenotyping and/or a framework for secondary validation is currently missing. We therefore sought to further support some of these observations with additional computational analyses, while also toning down the extent of any associated claims.

Specific comments in order within the manuscript

1-1: Commenting on Figure 1b, the authors indicate that bulk-seq, that is simply mRNA-seq of an entire embryo is not a great strategy for phenotyping embryos. I think people realize this. I am not aware of anyone suggesting this as a reasonable approach?

We rewrote the passage as we agree that there is already consensus on this point:

Previous text (line 138-140): “However, embryos corresponding to individual mutants did not cluster separately, suggesting that none were affected with severe, global aberrations and highlighting the inadequacy of bulk RNA-seq for detecting mutant-specific effects.”

Now: “However, embryos corresponding to individual mutants did not cluster separately, suggesting that none were affected with severe, global aberrations.”

1-2: Given the pseudo-bulk data in Figure 1 emphasizes strain differences and mouse mutants may be on a variety of complex backgrounds, does this mean that the investigator must always generate a de novo reference embedded library?

Thank you for this comment. Batch effects abound in the single cell field. Although external data could be used in principle (e.g. reuse of our data for the wildtype strains used here), we think our data makes it clear that it’s critical to include an “in-line” wild-type control while profiling mutant embryos. Fortunately, the structure of sci-experiments (wherein cells from each embryo are processed within individual wells of a single experiment) makes this relatively straightforward.

The revised version of the manuscript includes the following changes:

Added sentence (in Discussion): “Finally we hope that the data and analytical approaches here can be used for future studies that pursue whole embryo sc-RNA-seq as a means of phenotyping. Of note, our results

emphasise the importance of a well-matched control; although data from our wildtype embryos could be re-used as control data for other studies, that risks batch effects, and a safer strategy would be to always include a well-matched, “in-line” wild-type control while profiling mutant embryos.”

We note this consideration is not unique to whole-embryo sc-RNA-seq. Particularly for investigating more subtle effects, the inclusion of a well-matched control is general to all approaches to phenotyping.

1-3: Can the authors comment on the absence of expected trajectories in Figure 1e – why no gut, kidney, lung, skin? Does this represent under-sequencing, technical issues of proportional cell representation in the data or operator selectivity which would be counter to the idea of whole embryo phenotypic profiling? Understanding and providing a circumspect perspective on the limits as well as the successes would help balance the manuscript.

We apologise for the confusion regarding the trajectory annotations. In the manuscript we focused on identifying cellular phenotypes changing in different types of mutant mice rather than building an atlas of single-cell transcriptomes for E13.5 mouse embryos. Therefore, we sought to minimise the amount of text spent on cell-type identifications in the main text.

When we annotated the dataset, we first identified 13 major trajectories based on the partitions in the dataset, as shown in **Fig. 1e**. We then performed sub-clustering analysis on each major trajectory, through which we identified 64 well-annotated sub trajectories. The gut, kidney, lung and skin are represented in these sub-trajectories. For example, we identified 14 sub-trajectories in the epithelial trajectory, including the epidermis trajectory (skin), the renal epithelial trajectory (kidney), the foregut/midgut/hindgut epithelial trajectory (gut), and the lung epithelial trajectory (lung). In some cases, the sub-trajectories of the same tissue are split between multiple major trajectories. For example, we found different types of cells related to eye development, including retinal epithelial cells, retinal neuron cells, and lens cells (**Supplementary Fig. 3a**, replicated below with the three eye cell types highlighted in red rectangles). Some of the sub-trajectories can be even further sub-clustered to annotate more detailed subtypes (e.g. see **Fig. 4e**, replicated below). Overall, we believe that the dataset has strong representation of cell types at the whole embryo scale.

Figure 4e. RNA velocity on UMAP embedding of G4 wildtype and Sox9 regulatory INV cells in the limb mesenchymal trajectory labelled by annotation (left) or sample (right).

The revised version of the manuscript includes the following changes:

Previous text (line 167-168): “Altogether, we identified 13 major trajectories, 8 of which could be further stratified into 59 sub-trajectories (**Fig. 1e**; **Supplementary Fig. 3**; **Supplementary Table 2**).”

Now: “Altogether, we identified 13 major trajectories, 8 of which could be further stratified into 59 sub-trajectories (Fig. 1e; Supplementary Fig. 3; Supplementary Table 2), generally covering the expected cell trajectories at this stage of development.”

Supplementary Figure 3. Annotation of sub-trajectories in data from wildtype E13.5 embryos. From 215,517 single cell profiles of wildtype E13.5 embryos of four strains in MMCA, we annotated 13 major trajectories. For 8 of these 13 major trajectories, iterative analysis identified the additional sub-trajectories shown here as 3D UMAP visualisations. Cells are coloured by sub-trajectory annotations. PNS: peripheral nervous system. MHB: midbrain-hindbrain boundary. Di: Diencephalon. Eye-related cell types are highlighted in red rectangles.

1-4: In supplementary Figure 5 the authors note differences in cell numbers for individual tissue trajectories. Given substantial inter-embryo variability, is there a good sense of the number of replicates that would be required then to put these trends onto solid statistical grounds? The word “nominally” appears to be a qualifier to the significance of observed differences at a few points in the text.

This is a great point, and we thank the reviewer for raising it. We hypothesise that the power of our strategy to detect the cell proportion changes between different genotypes is affected by three factors: a) the abundance of a given cell type; b) the number of replicates in each genotype group; and c) the effect size. To evaluate power, we performed a simulation analysis that varied these factors (script here), implemented as follows:

1. We selected the 20 most abundant cell types in wildtype embryos. Their abundances ranged from ~1% to ~20%. The proportions of these cell types served as the basis for our simulations.
2. We simulated ten groups of “wildtype” samples with 4, 8, 16, ..., 40 replicates in each group, wherein each sample consisted of cells drawn from the 20 cell types. For each replicate, the simulated number of cells of each cell type was calculated as the product of: a) the cell-type proportions, simulated by fitting a dirichlet model based on the real proportions from step 1; and b) the total number of cells recovered for that replicate, simulated based on the mean ($n = \sim 15,000$) and standard deviation of the cell numbers across replicates in the real dataset.
3. We simulated ten groups of “mutant” samples by repeating the above step except adding shifts to the numbers of cells within each cell type. The shifting scales were based on different effect sizes. For instance, effect size = 0.1 represents a 10% reduction in the number of cells.
4. We performed beta-binomial regression (the same test used in **Fig. 2a**) to test if the cell type proportions were significantly changed between simulated “wildtype” and “mutant” samples, further checking the results as stratified by cell type (with different abundances), the number of replicates, and the effect size.

The results are in line with our hypothesis that the detection power of our strategy varies among comparisons with different effect sizes, sample sizes, or cell-type abundances. We recognize that the new **Supplementary Fig. 7**, replicated below, is dense, but the main “take-home” messages are summarised below:

1. 25% changes are robustly detectable, even for rare cell types like <2%, with modest numbers of replicates.
2. 10% changes are possible to detect, but only for abundant cell types (e.g. >5%). More replicates can help in this zone.
3. 1% changes are almost impossible to detect with a cell proportions approach, even with very large numbers of replicates.

In general, at the level of single cell sampling performed in our study, four samples (corresponding to the number of samples used in the manuscript) would be sufficient to detect a 25% effect size for those cell types that is present at a 1% proportion in wildtype embryos. Qualitatively similar results were obtained in simulating increases in cell type proportions. We now describe this simulation in a new section in the **Methods**, and summarise the result in the Discussion and **Supplementary Fig. 7**:

“Based on a simulation analysis of the analytical approach that considers cell proportions only, four replicates of each mutant is likely sufficient to detect modest changes in abundant cell types (e.g. a 10% change for cell types at 10% abundance) but only large changes in rarer cell types (e.g. a 25% change in cell types at 1% abundance; **Supplementary Fig. 7**). As such, to detect more subtle changes in model organisms like the mouse where very large numbers of replicates are not feasible, more sophisticated strategies such as lochNESS, which is not based on counts of cell types but rather directly considers the distribution of cells derived from different genotypes in a complex embedding, may be essential.”

Supplementary Figure 7. Simulation-based estimation of the number of replicates required to detect cell proportion changes. We simulated “wildtype” and “mutant” embryos with parameters drawn from our data (**Methods**), and then performed beta-binomial regression to ask whether cell-type proportions for a given cell type are different between genotypes while varying simulated effect sizes and varying numbers of replicates. In the global view, each column represents a given effect size (e.g. 0.01, highlighted on the top) and each row represents a given cell type, with its cell proportion in the whole embryo highlighted at the right. Each single plot represents the testing results of beta-binomial regression for different numbers of replicates of each genotype (y-axis, ranging from 4 to 40). The x-axis refers to $-\log_{10}$ scaled unadjusted p-values, and the dot is coloured either red (insignificant testing result with unadjusted p-value > 0.05) or blue (significant testing result with unadjusted p-value < 0.05).

1-5: The observation that eye and lens contributions are reduced in *Ttc21b* correlating with an observed eye phenotype at 11.5 could be strengthened further by aligning the anatomical analysis with the e13.5 single cell RNA-seq data set. Some simple clear histology at e13.5 should be up to the task.

As suggested, we further investigated eye development in the homozygous and heterozygous *Ttc21b* mutants and wildtype embryos at E13.5 using H&E staining. These results strongly support the phenotype identified via cell type composition analysis of this mutant.

As described in lines 257-264 of the original manuscript, our initial analysis revealed a strong depletion of retinal neurons ($\log_2(\text{ratio}) = -6.69$; unadjusted p-value = 0.028; beta-binomial regression) also seen in the heatmap (**Fig. 2b**). Further investigating the composition of the eye developmental trajectories, we saw a significant reduction in cell numbers of the lens ($\log_2(\text{ratio}) = -2.64$) and retina epithelium ($\log_2(\text{ratio}) = -2.32$) trajectories (note that these numbers have slightly changed in the updated manuscript because we are now using all genotypes of the same background as controls, rather than wildtypes only; see response to comment 2-2 for more details). In a brief validation, we described the collapse of the eye structure at E11.5 in the homozygous mutant (**former Fig. 2c**).

Former Figure 2c. Homozygous *Ttc21b* KO mice embryo (E11.5) showed abnormal eye development.

For the revision, as suggested by the reviewer, we sought to go deeper on this validation, investigating the morphological phenotype in depth using H&E stainings. In brief, these experiments confirm that the homozygous mutant has a depletion in various cell types composing the developing eye. In the H&E staining, the retinal neurons (3, **Fig. 2c**), lens (4, **Fig. 2c**) and the optic nerve (5, **Fig. 2c**) are missing completely in the homozygous mutant. This observation correlates to the lens and retinal neurons showing strong reductions in our single cell data. We furthermore detect delocalization and strong reduction of the retinal epithelium (6, **Fig. 2c**) compared to the wildtype and heterozygous mutant, matching with the loss of cells in the retinal epithelium trajectory in our single cell data (**Supplementary Fig. 8c**). Of note, the abnormal eye phenotype in *Ttc21b* mutants has been previously known but was considered likely secondary to the more general craniofacial defect in this line. The sc-RNA-seq analysis and histology suggest this is not the case, and rather that overactive SHH signalling in this mutant has a primary effect on retinal development.

Figure 2c. H&E staining of 2μm sections in the developing eye of the homozygous (Hom), heterozygous (Het) and wildtype *Ttc21b* litter-derived, E13.5 staged embryos. Images taken with 20x magnification show the loss of structures such as the retinal neurons (3), lens (4) and the optic nerve (5) and a strong reduction of the retinal epithelium (6) exclusively in the homozygous mutant.

The revised manuscript includes the following changes:

Previous text (lines 262-264): “Validating this finding, the developing eye appears diminished in the homozygous *Ttc21b* mutant at E11.5 embryos compared to the wildtype or heterozygous mutant (**Fig. 2c**).”

Now: “Proceeding to H&E staining to validate these findings, we investigated the morphology of the eye at E13.5 in wildtype, heterozygous and homozygous embryos from the same litter. The morphological phenotype detected supports the changes in cell type composition of the mutant analysis. The homozygous mutant displays a visible collapse in structures that are detectable within the WT and heterozygous mutants eye at E13.5. The retinal neurons (3, **Fig. 2c**), lens (4, **Fig. 2c**) and the optic nerve (5, **Fig. 2c**) are missing in the homozygous mutant. These morphological alterations coincide with the changes in cell type composition of the mutant, as the lens and retinal neurons show strong reduction in numbers. The retinal epithelium (6, **Fig. 2c**) was delocalized and reduced in numbers as well, matching to the loss of cells detected in the retinal epithelium trajectory (**Supplementary Fig. 8c**).”

New: “The diverse set of mutants we analysed yielded a variety of results that speak to the utility of whole embryo sc-RNA-seq as a phenotyping tool. For example, an abnormal eye phenotype in *Ttc21b* mutants has been previously described, but was considered likely to be secondary to the more general craniofacial defect in this line^{2,12}. The sc-RNA-seq analysis of E13.5 *Ttc21b* mutants demonstrated that multiple retinal cell trajectories were essentially absent (**Fig. 2a,b, Supplementary Fig. 8**). Detailed histological analysis (**Fig. 2c**) confirmed this, suggesting that the eye abnormality is likely not a secondary effect, but rather that the overactive SHH signalling in this mutant has a primary effect on retinal development. This is consistent with the observation that reducing *Shh* signalling in *Ttc21b* mutants partially rescues the eye abnormality². Of note, these results are consistent with a recent case study of a 9-yr old patient carrying pathologic variants in *Ttc21b* who had, in addition to the well-characterised cystic kidney phenotype associated with this mutation, a retinal dystrophy consistent with a failure of normal eye development¹³.”

1-6: What is the lesson to be learned from the ZRS mutant? In this case, this is a control with a well-documented limb phenotype not evident in first pass analysis requiring a tailored secondary analysis to reveal a limb

phenotype. Does this mean in primary screening of an embryo of unknown phenotype, quite significant limb (and other) phenotypes would go unnoticed? The ZRS mutant is a macroscopic phenotype I believe (see line 309). The lochNESS approach looks promising in this example (Figure 3b). One assumes no false negative enrichments in other populations?

The reviewer is correct that we included the ZRS mutant as a control, as this enhancer is very specific for the limb development and, when knocked out, leads to a limb-specific ‘truncated’ phenotype¹¹. Since no other significant phenotype has been documented for this mutant, we are assuming that we should not *a priori* expect any enrichments or depletions in other cell populations in this mutant.

We indeed were able to detect the limb-specific truncation phenotype in the first pass analysis (this was only briefly mentioned in the submitted manuscript). The cell type composition heatmap revealed that the limb mesenchyme exhibits a ~24% reduction in cell numbers (line 270, **Fig. 2a**), and even more strikingly, the UMAP of the wildtype compared to the ZRS mutant shows the cells are lost in a distinct area (**Fig. 2d**). This subpopulation of cells is expressing markers of the distal mesenchyme of the early embryonic limb bud (**Fig. 2e**). The first pass cell type composition analysis also identified seven other sub-trajectories whose proportions were nominally altered in the ZRS limb enhancer KO (fold-change between ZRS mutant and controls ranging from 0.91 to 1.86), but we did not observe “regional” losses or gains in the co-embeddings as in the limb mesenchyme (**Supplementary Fig. 9b**, formerly **Supplementary Fig. 7b**, replicated below).

Supplementary Figure 9b. UMAP visualisation of co-embedded cells of various sub-trajectories from the ZRS limb enhancer KO and FVB wildtype. The same UMAP is shown twice for each, highlighting cells from either FVB wildtype (left) or ZRS limb enhancer KO (right). These are the seven sub-trajectories in which, in addition to limb mesenchyme, we detected nominally significant differences in cell type proportions for the ZRS limb enhancer KO.

As such, although it's challenging to know with certainty whether there are false negatives in less characterised mutants, the expected ZRS effect was definitely observed in our first pass analysis, which is encouraging. We

then used the lochNESS approach to further analyse the ZRS mutant, in order to introduce this new method and test its potential to detect subpopulations of cells altered in a cluster-annotation-independent manner.

To further elaborate on this point, we made some changes to the manuscript:

Previous text (line 123-127): “4) prospective deletions of cis-regulatory elements, including of TAD boundaries in the vicinity of developmental transcription factors including Smad3, Twist1, Tbx5, Neurog2, Sim1, Smad7, Dmrt1, Tbx3, and Twist1, and, as a positive control, the ZRS distal enhancer (Zone of polarizing activity Regulatory Sequence) which regulates sonic hedgehog (SHH) expression and results in absent distal limb structures.”

Now: “4) prospective deletions of cis-regulatory elements, including of TAD boundaries in the vicinity of developmental transcription factors including Smad3, Tbx5, Neurog2, Sim1, Smad7, Dmrt1, Tbx3, and Twist1¹⁰. All mutants were homozygous except for *Scn11a* GOF. As a positive control, this category includes a ZRS distal enhancer (Zone of polarising activity Regulatory Sequence) KO mutant. This enhancer, which regulates sonic hedgehog (Shh) expression, results specifically in absent distal limb structures when deleted and therefore serves as a test case for our analytical strategies¹⁴.”

Also:

Previous text (line 266-271): “However, most changes were relatively subtle. For example, the ZRS limb enhancer KO is a well-studied mutant that shows a loss of the distal limb structure at birth. This analytical framework highlighted eight sub-trajectories whose proportions were nominally altered in the ZRS limb enhancer KO, most of which were mesenchymal. However, although the most extreme, the reduction in limb mesenchymal cells was only about 30% ($\log_2(\text{ratio}) = -0.49$; unadjusted p-value = $6.32e-3$; beta-binomial regression).”

Now: “Further testing our approach we examined the positive control, the ZRS limb enhancer KO, which is a well-studied mutant which shows a loss of the distal limb structure at birth. This analytical framework highlighted eight sub-trajectories whose proportions were nominally altered in the ZRS limb enhancer KO, most of which were mesenchymal. The reduction in limb mesenchymal cells was about 24% ($\log_2(\text{ratio}) = -0.39$; unadjusted p-value = $6.32e-3$; beta-binomial regression).”

1-7: The *Gli2* mutant floorplate gene expression data result is odd. Was *Shh* expression lost as expected in the *Gli2* mutant cells? This would be perhaps the most obvious gene to investigate. The roof plate results are unexpected given no obvious connection of *Gli2* with the roof-plate and no evidence of Hedgehog pathway regulation in this region. However, Hedgehog transcriptional regulation is complex and the loss of a *Gli2* repressor form could have some effect? Potentially, there is a strong proof-of-principle that novel phenotypes can be uncovered but this would require a significant analysis of the *Gli2* mutant to demonstrate there really are changes within the roof-plate population. It is not sufficient to report the unexpected without a rigorous validation when rolling out a new methodology.

Thank you for this important comment. Our response to this comment is notably identical to the related comment 2-4 below (*i.e.* comment 4 from Reviewer 2).

Previous publications on *Gli2* knockouts show the lack of floor plate development and further a malformation of the neural tube, due to disrupted *Shh* signalling pathways^{15,16,17,18}. However, several studies in which *Gli2* knockouts are analysed, effects on the roof plate or roof plate derivatives, were either not described or excluded^{4,12,15}.

To investigate the changes we describe in the floor and roof plates of the *Gli2* KO mutant in greater detail, we performed both additional computational analyses as well as new experimental validations. First, we mapped

the three clusters we identify in **Supplementary Fig. 11a** (formerly **Supplementary Fig. 8d**) to recently published spatially-resolved wildtype mouse embryo transcriptomics data (E13.5) using the ML-based Tangram algorithm (**Supplementary Fig. 11e**). Cells from cluster 1, which we previously labelled as floor plate, were spatially mapped to the floor of the neural tube, supporting our annotation. Clusters 2 and 3, which we previously labelled as roof plate, did not map to the traditional roof plate location in the roof of the neural tube, but instead mapped to distinct areas within the developing brain. Cluster 3 mapped to the choroid plexus (ChP), a structure that is derived from the roof plate and develops in both the lateral ventricle (anterior) and 4th ventricle (posterior) of the developing brain, while cells from cluster 2 mapped next to the lateral ventricle ChP. This is consistent with the strong expression of *Ttr* in cluster 3, which is the classic marker for the ChP, and expression of *Wnt8b/9a/3a* in cluster 2, which are genes expressed in the cortical hem, a roof plate derivative attached to the lateral ventricle ChP (**Fig. 3e, Supplementary Fig. 11d**)^{19,20,21}. We then investigated gene expression markers that further separate lateral ventricle and 4th ventricle of ChP²² and found that in addition to roof plate marker *Lmx1a*, cluster 3 expresses 4th ventricle marker *Meis1* and cluster 2 expresses lateral ventricle markers *Otx1* and *Emx2* (**Supplementary Fig. 11f**).

In summary, cluster 1 remains annotated as the floor plate, cluster 2 is now annotated as the cortical hem of the lateral ChP, and cluster 3 is now annotated as the lateral and 4th ventricle ChP. Our analyses of these whole embryo sc-RNA-seq-derived clusters with lochNESS suggest that both the floor plate (cluster 1) and cortical hem (cluster 2) are (relatively) reduced in cell numbers in the *Gli2* KO, while the ChP (cluster 3) is increased.

Towards secondary validation of the floor plate (expected) and roof plate (unexpected) observations, we investigated these structures in *Gli2* KO mutants and wildtype mice. In coronal sections through the developing brain and neural tube, the *Gli2* KO mutant showed severe developmental defects such as deformed forebrain lobes and delayed neural tube development (**Supplementary Fig. 12a**). Focusing on the neural tube, the expected location of the floor and roof plates, the *Gli2* KO mutant revealed a severely disturbed shape upon immunofluorescence imaging of *Pax6*, a neural tube developmental marker (**Supplementary Fig. 12b**).

In particular, the *Gli2* KO mutants exhibit the well-described “dorsalization” of the neural tube, which is complemented by our independent determination (by whole embryo sc-RNA-seq) that the floor plate is markedly reduced in the *Gli2* KO mutant. This effect has been previously described in the *Gli2* KO mutant, caused by the disruption of the Shh signalling pathway¹⁵. Consistent with this, we observe downregulated Shh in the floor plate of the *Gli2* KO mutant, relative to WT, as well as of target genes of this pathway including *Nkx2.2*, *Foxa2*, *Olig2*, *Ptch1* and *Smo* (**Revision Fig. 1**)^{23,24}. Additionally, *Gli3*, known to function as a repressor for *Gli2* activity in the neural tube²⁵, is downregulated in the *Gli2* KO mutant. To summarise, direct Shh pathway targets, as well as pathway-relevant receptors and repressors, are downregulated in the *Gli2* KO floor plate, emphasising the deregulation of the pathway upon mutation of the *Gli2* gene, leading to the described ‘dorsalization’ phenotype.

Revision Figure 1. Dotplot showing normalised expression of and percent of cells expressing *Shh* and genes in the *Shh* pathway of the neural tube, focusing on cells in cluster 1 (floor plate) of *Gli2* KO and wildtype cells.

Examining the ChP in more detail, *Ttr* staining in mutant and wildtype sections of the lateral ventricle as well as the 4th ventricle shows that the shape of the tissue ('branching') as well as the development of the single cell layer is disturbed within the mutant, revealing a 'double dapi' layer in the mutant compared to the wildtype, which displayed *Ttr* signal consistently in a single layer (Fig. 3f; Supplementary Fig. 13, red arrows). Because the *Gli2* KO mutants are overall smaller in size at this stage, we normalised the *Ttr* positive cells of the lateral and 4th ventricle to the size of the embryo section area (in μm) and found the mutants to have an increased proportion of *Ttr* positive cells (Supplementary Fig 13c; Supplementary Table 7). These imaging-based observations indicate a relative enrichment of cell counts in the ChP in *Gli2* KO mutants, consistent with the lochNESS results.

Taken together, these secondary studies validate our findings in the *Gli2* mutant of differences in both the floor plate and the ChP, a derivative of the roof plate. Unfortunately due to multiple dysfunctioning antibodies, we have not yet been able to validate the changes within cluster 2, the cortical hem, directly. However, given the extensive deformations in the embryo in this region and that the other two related observations (cluster 1, cluster 3) were successfully experimentally validated, we are reasonably confident that the changes in the juxtaposed cortical hem (cluster 2) are valid as well. The relevant figure panels of the revised manuscript are reproduced below:

Figure 3d-f. **d**, UMAP visualisation of co-embedded cells of the floor plate and roof plate sub-trajectories from the *Gli2* KO mutant and pooled wildtype, coloured by lochNESS. ChP = choroid plexus. **e**, same as in panel d, but coloured by expression of selected marker genes. **f**, Choroid plexus marker *Ttr* staining of the developing brain regions (LV = lateral ventricle, 4V = 4th ventricle, ChP = choroid plexus) in H&E sections of wildtype and *Gli2* $-/-$ mutants in 20x magnification.

Supplementary Figure 11. Analysis of *Gli2* KO in the roof plate and floor plate trajectories. **a**, UMAP visualisation of co-embedded cells of the floor plate and roof plate sub-trajectories from the *Gli2* KO mutant and pooled wildtype, coloured by sub-trajectory (left) or cluster number (right). **b**, Boxplot showing the lochNESS distribution in each cluster shown on the right of panel **a**. **c**, Barplots showing the cell composition of each cluster shown on the right of panel **a**, split by mutant vs. wildtype (left) or individual embryo (right), with a reference line at the overall wildtype cell proportion. **d**, Dotplot summarising the expression of and percent of cells expressing selected marker genes in each cluster shown on the right of panel **a**. **e**, Tangram-inferred locations of each cluster shown on the right of panel **a**. Red arrows highlight the areas where cells map to with high probability. The colour scale is set from 1st percentile to 99th percentile. **f**, UMAP visualisation of co-embedded cells of the floor plate and roof plate sub-trajectories from the *Gli2* KO mutant and pooled wildtype, coloured by expression of marker genes.

Supplementary Figure 12. Morphological phenotype of *Gli2*^{-/-} mutants. **a**, H&E staining of two mutant and two wild type E13.5 embryos in cranial-caudal (1-3) order within the head. In order to compare mutant and wildtype slides in neural tube development, the slides are matched based on hallmarks such as eyes, tongue muscle and nasal cavities. **b**, Neural tube marker *Pax6* staining of the developing neural tube in consecutive sections of H&E sections 1.3 and 2.3 to visualise the structure of the neural tube formation in wildtype and mutant in 10x and 20x magnification.

Supplementary Figure 13. *Ttr* staining in wildtype mice and *Gli2* ^{-/-} mutants. Choroid plexus marker *Ttr* staining of the developing brain regions (LV = lateral ventricle, 4V = 4th ventricle, ChP = choroid plexus) in H&E sections of **a**, wildtype and **b**, *Gli2* ^{-/-} mutants in 2x, 10x and 20x magnification. For each section (2x magnification), the regions of interest are highlighted with white boxes and shown in higher magnification on the sides (10x or 20x magnification). Red arrows highlight areas with a normal single layer of *Ttr* expressing cells in wildtype, and two layers of cells in the mutant.

Relevant sections of the text have also been revised as follows:

Previous text (lines 355-370): “Overall, these observations are consistent with the established role of *Gli2* in floor plate induction and the previous demonstration that *Gli2* knockouts fail to induce a floor plate^{15,17}. Less expectedly, this focused analysis also revealed two subpopulations of roof plate cells, one depleted and the other enriched for *Gli2* KO cells (**Fig. 3e**; **Supplementary Fig. 8d-f**). To annotate these subpopulations, we examined genes whose expression was predictive of lochNESS via regression (**Methods**). The mutant-enriched group of roof plate cells was marked by *Ttr*, a marker for choroid plexus¹⁹ and dorsal roof plate development²⁶, as well as genes associated with the development of cilia (*e.g.* *Cdc20b*, *Gmnc*, *Dnah6* and *Cfap43*), while the mutant-depleted group was marked

by Wnt signalling-related genes including *Rspo1/2/3* and *Wnt3a/8b/9a* (**Fig. 3f; Supplementary Fig. 8g; Supplementary Table 3**)²⁷. It has been shown that ventrally-expressed Gli2 plays a central role in dorsal-ventral patterning of the neural tube by antagonising Wnt/Bmp signalling from the dorsally-located roof plate²⁸. Our results are consistent with this, and also define two subpopulations of roof plate cells on which Gli2 KO appears to have differential effects.”

Now: “Overall, these observations are consistent with the established role of *Gli2* in floor plate induction, its role as activator of *Shh* in dorso-ventral patterning of the neural tube and the previous demonstration that *Gli2* knockouts fail to properly induce a floor plate^{15,17} .

Less expectedly, this focused analysis also revealed two subpopulations of roof plate derivative cell types, one relatively depleted and the other relatively enriched in *Gli2* KO embryos (**Fig. 3d; Supplementary Fig. 11a-c**). To annotate these subpopulations, we examined genes whose expression could be predicted by lochNESS via regression (**Methods**). The mutant-enriched group of roof plate cells was marked by *Ttr*, a marker for choroid plexus¹⁹, as well as genes associated with the development of cilia (e.g. *Cdc20b*, *Gmnc*, *Dnah6* and *Cfap43*), while the mutant-depleted group was marked by Wnt signalling-related genes including *Rspo1/2/3* and *Wnt3a/8b/9a*, indicating it to be a region close to the choroid plexus of the lateral ventricle, namely the cortical hem (**Fig. 3f; Supplementary Fig. 11d; Supplementary Table 4**)²⁷. To confirm these annotations, we mapped the three clusters shown in **Supplementary Fig. 11a** to spatial transcriptomic data from the mouse embryo (E13.5) using the Tangram algorithm²⁹ (**Supplementary Fig. 11e**). Cells from cluster 1 mapped to the floor of the neural tube, consistent with their annotation as floor plate. Cells from cluster 2 mapped next to the lateral ventricle choroid plexus (ChP), while cells from cluster 3 mapped to the ChP, both in the lateral (anterior) and 4th (posterior) ventricles, supporting those annotations as well. We then investigated gene expression markers that further separate lateral ventricle and 4th ventricle of ChP²² and found that in addition to roof plate marker *Lmx1a*, cluster 3 expresses 4th ventricle marker *Meis1* and cluster 2 expresses lateral ventricle markers *Otx1* and *Emx2* (**Supplementary Fig. 11f**).

To experimentally validate these findings, we investigated the developmental progress of the neural tube and brain in stage E13.5 *Gli2* KO mutant and WT embryos. In coronal sections of the mutant, we observed severe developmental defects including deformed forebrain lobes and delayed neural tube development compared to the wildtype (**Supplementary Fig. 12a**). More detailed immunofluorescence imaging of *Pax6*, a neural tube developmental marker, revealed a severely disturbed shape of the neural tube, confirming the well-described “dorsalization” phenotype of the neural tube in *Gli2* KO mutants (**Supplementary Fig. 12b**), and consistent with the marked reductions in the relative number of floor plate cells in this mutant as determined by whole embryo sc-RNA-seq (**Fig. 3d**). Turning to the less expected observation of increased ChP, we found the lateral ventricle as well as the 4th ventricle displayed a disturbed *Ttr* staining pattern. While the wildtype shows inner and outer *Ttr* signal within the single cell layer, the mutant displayed a ‘double dapi’ layer, indicating a disordered tissue organisation (**Fig. 3f; Supplementary Fig. 13a,b**). Adjusting for the overall smaller size of the *Gli2* KO mutants at E13.5, we quantified *Ttr*-positive cells in the lateral and 4th ventricle relative, finding a proportional increase in the mutant relative to WT (**Supplementary Fig. 13c, Supplementary Table 5**), again consistent with the marked increase in the relative number of ChP cells in this mutant as determined by whole embryo sc-RNA-seq. In summary, we could confirm the expected reduction in floor plate in the mutant, as well as the unexpected increase in roof plate-derived ChP. ”

New: “The utility of pursuing whole embryo sc-RNA-seq was also demonstrated by an unexpected finding of both a depleted and an enriched cell population of roof plate cells in the *Gli2* KO mutant. The “dorsalization” of the neural tube in the absence of SHH signalling is well-described^{2,15,17} and was confirmed in our histological analysis of this line (**Supplementary Fig. 12**). However, the roof plate in *Gli2* KO mice has been described as normal, based on traditional analyses of dorso-ventral patterning by examination of transverse sections of the developing neural tube¹⁵. In contrast, whole embryo sc-RNA-seq uncovered that derivatives of the roof plate also depict changes in composition (a primary finding) and tissue development (based on secondary validation) in the mutant, illustrating how this approach can potentially yield new insight into even well-studied developmental pathways.”

1-8: Ditto epidermal Tbx3.

We thank the reviewer for this comment and agree that a validation of this finding would be beneficial. In another study, bulk-RNA seq of the face and the heart as well as qPCR of several tissues of this mutant did not reveal any significant changes in gene expression of the two neighbouring genes Tbx3 and Tbx5¹⁰. This is most likely due to the fact that bulk-RNA seq is much less sensitive than the lochNESS method we applied on single cell data. However, since the changes we observe in our analysis are very subtle, we were not able to establish a new validation method that could detect such changes in a timely manner without delaying the manuscript much further. We have revised the relevant text to emphasise the preliminary nature of the *Tbx3* finding:

New: “Of note, the shifts we observed in *Tbx3* TAD boundary KO cells remain preliminary and would need to be confirmed by further validation experiments.”

We also try to make a broader point in the Discussion, which is that there are likely to be subtle findings through the application of whole embryo sc-RNA-seq for which conventional validation methods may not be sufficiently sensitive. How to address this remains an open question. We now say:

New: “Additionally, some of the phenotypes identified here have probably not been described before due to the lack of resolution of conventional phenotyping approaches. New secondary validation strategies need to be developed to confirm subtle defects in molecular programs or subtle changes in the relative proportions of specific cell types.”

1-9: For *Scn11a*, the authors suggestion that *Scn11a* mutants are developmentally earlier makes sense. What does not make sense is why the authors believe there is a significance to this outcome and the mutation rather than the possibility that this represents expected variability in collection of litters. Again, if we are to accept a possibility that modifying pain reception has a general growth phenotype delaying development at e13.5, there should be some independent validation of additional *Scn11a* mutants (and littermates).

Thank you for the helpful comment. To follow up on the difference in developmental stage between the *Scn11a* mutants and the rest of the embryos, we looked at *Scn11a* mutants and wildtypes from two heterozygous x WT crossed litters, comparing the morphology of the wildtype vs. heterozygous embryos at E13.5 by visual inspection (**Supplementary Fig. 16**). As depicted in the embryo pictures below, there is no significant difference in size or observable developmental features such as eye and limb development which would explain the hypothesised overall developmental delay based on two litters including 10 wildtypes and 4 heterozygous mutants in total. If, as hypothesised in our original manuscript, the delay was due to developmental stagnation within the mutant comparable to embryonic day E12.5 (**Fig. 1c**), the heterozygous mutants would be expected to show distinct differences such as smaller body size and delay in external features. Considering these new results, wrong staging of the mutants used for the initial single cell experiment is the most parsimonious explanation.

Supplementary Figure 16. *Scn11a* mutant and wildtype morphology comparison. Images of 14 E13.5 staged embryos from two litters of WT and *Scn11a* heterozygous mutants. Accessible developmental features (limbs, eyes and body size) were compared between the mutants and the WT by eye.

The relevant section in the manuscript has been revised accordingly:

Previous text (lines 450-454): “As such, the apparently unique signature of *Scn11a* GOF cells might be attributable to these embryos simply being earlier in development, suggesting a more global role for sodium ion channels not only for neuronal function but also early development and cell fate determination³⁰. Incorrect staging is formally possible, but unlikely because the embryos derived from three independent litters.”

New: “To further follow up on the possibility of earlier time points being inadvertently harvested for this mutant, we examined 2 mixed litters of wildtype and heterozygous mutants at the stage E13.5 (**Supplementary Fig. 16**). As the heterozygous mutants did not show any signs of developmental delay such as smaller size, difference in eye formation or limb development, the theory of general developmental delay seems unlikely, as a more parsimonious explanation is incorrect staging of these litters.”

1-10: No secondary validation of predictions for *Atp6v0a2*.

- As in the previous comments, no secondary validation.

Thank you for this comment. As noted above (comment 1-8 related to *Tbx3*), we elected to pursue secondary validations for other novel observations where it was clearer how to go about doing so. In this case, we note that the greater similarity of the *Atp6v0a2* KO, *Atp6v0a2* R755Q and *Gorab* KO mutants relative to one another, as compared to other mutants, is very clear in the data, and to some degree expected (partly in retrospect). As stated in the manuscript (lines 515-526):

“In sharp contrast with the relative uniqueness of the *Scn11a* GOF mutant, we also observed that the similarity scores between three mutants -- *Atp6v0a2* KO, *Atp6v0a2* R755Q and *Gorab* KO -- was consistent with shared effects, in the mesenchymal, epithelial, endothelial, hepatocyte and neural crest (PNS glia) trajectories in particular; in other main trajectories, such as neural tube and hematopoiesis, *Atp6v0a2* KO and *Atp6v0a2* R755Q exhibited high similarity scores with one another, but not with *Gorab* KO (**Fig. 3g; Supplementary Fig. 15a,c,f**). Such sharing is perhaps expected between the *Atp6v0a2* KO and *Atp6v0a2* R755Q mutants, as they involve the same gene. In human patients, mutations in *ATP6V0A2* and *GORAB* cause overlapping connective tissue disorders, which is reflected in the misregulation of the mesenchymal trajectory of *Atp6v0a2* and *Gorab* mutants⁸⁻⁷. However, only the *ATP6V0A2*-related disorder displays a prominent CNS phenotype, consistent with the changes in the neural tube trajectory seen only in both *Atp6v0a2* models (**Supplementary Fig. 15a,c,f**).”

Our ability to experimentally validate this sharing at the level of individual trajectories, e.g. the mesenchymal trajectory (**Fig. 3g**), is challenged by factors including: (1) mesenchymal subtypes are very poorly understood, relative to other cell types; (2) conventional phenotyping approaches may lack the necessary resolution to quantify subtle shifts that collectively underlie the sharing that we see in **Fig. 3g**; and (3) more generally, a framework for secondary experimental validation of patterns that are driven by relatively subtle shifts in across the entire transcriptome, as opposed to individual genes, is lacking.

Nonetheless, towards beginning to address such issues and as an indirect means of shoring up the observations, we performed an additional computational analyses to understand in greater detail the underlying reasons for why these three mutants – *Atp6v0a2* KO, *Atp6v0a2* R755Q and *Gorab* KO – are exhibiting much higher similarity with one another than with other mutants.

For this, we focused on the intermediate mesoderm trajectory (a subset of mesenchyme), and performed sub-clustering followed by an attempt at more detailed annotation. We were able to resolve 10 out of 13 sub-clusters using known marker genes (**Fig. 3h**), which include not only intermediate mesoderm (*Pax2+*, *Pax8+*), but also multiple different subtypes of lateral plate mesoderm-derivatives (e.g. lung mesenchyme (*Tbx5+*, *Tbx4*)), vascular smooth muscle cells (*Acta2+*, *Mylk+*), proepicardium (*Upk3b+*), and adrenocortical (*Cyp21a1+*, *Cyp11b1+*) & leydig cells (*Cyp11a1+*, *Cyp17a1+*); the full list of marker genes are provided in **Supplementary Table 6**). In light of this extensive heterogeneity and the nature of these subtypes, we renamed the previous “intermediate mesoderm” to “lateral plate & intermediate mesoderm” trajectory.

The similarity scores are based on the distribution of cells in a complex embedding. We sought to identify the subsets of this embedding (i.e. annotated subclusters) where *Atp6v0a2* KO, *Atp6v0a2* R755Q and *Gorab* KO are similarly altered (former **Fig. 3h**). In particular, we compared the cell type proportion of each cell subcluster between individual *Atp6v0a2* KO, *Atp6v0a2* R755Q and *Gorab* KO mutants and other individuals within the same background strain (**Fig. 3i**). From this analysis, we find that clusters 2, 5, 6 are enriched for cells from *Atp6v0a2* KO, *Atp6v0a2* R755Q and *Gorab* KO mice (2: proepicardium; 5: hepatic mesenchyme; 6: lung mesenchyme), while clusters 3 and 4 are depleted in all three mutants (3: gastrointestinal smooth muscle; 4: unknown) (**Fig. 3j**).

The mutants exhibited similar trends in cell number changes; for example, in sub-cluster 2 (pro-epicardium), the *Atp6v0a2* KO, *Atp6v0a2* R755Q and *Gorab* KO mutant all display an increase (35%, 53% and 18%) of cells compared to the other mutants and wild-type of this background. In contrast, in sub-cluster 3 (gastrointestinal smooth muscles), the *Atp6v0a2* KO, *Atp6v0a2* R755Q and *Gorab* KO mutants are relatively decreased (-33%, -16% and -8%).

Figure 3h-j. **h**, UMAP showing the co-embedding of the lateral plate & intermediate mesoderm sub-trajectory for mutants from the C57BL/6 background strain, coloured and labelled by cluster, detailed cell type label (marker genes and references are provided in **Supplementary Table 6**). **i**, Boxplots showing the composition of each cell subcluster for individual *Atp6v0a2* KO, *Atp6v0a2* R755Q and *Gorab* KO mutants (blue) and other individuals within the same background strain (red). Top 6 subclusters are included in the visualisation. **j**, same as in **h**, but coloured by log-transformed expression of marker genes.

The following text was changed to reflect these updated annotations.

Previous Text (Lines 471-478): “We identified three subclusters of intermediate mesoderm where *Atp6v0a2* KO, *Atp6v0a2* R755Q and *Gorab* KO mice are similarly distributed compared to other C57BL/6 genotypes (**Fig.3h,i**). In particular, cluster 1 is enriched for cells from *Atp6v0a2* KO, *Atp6v0a2* R755Q and *Gorab* KO mice and is marked by genes related to epithelial-to-mesenchymal transition, cell-cell adhesion and migration, such as *Podxl*, *Frem2* and *Muc16*³¹⁻³³. Clusters 2 and 3 are depleted in cells from *Atp6v0a2* KO, *Atp6v0a2* R755Q and *Gorab* KO mice and are marked by muscular development related genes like *Synpo2*, *Myh11* and *Myocd* (cluster 2), and cell-cell adhesion related genes like *Itga1* and *Ctnna3* (cluster 3)³⁴⁻³⁸.”

New: “We resolved the identity of most sub-clusters of lateral plate & intermediate mesoderm using marker genes and ML based spatial mapping, and identified multiple subsets where *Atp6v0a2* KO, *Atp6v0a2* R755Q and *Gorab* KO mice are similarly distributed compared to other C57BL/6 genotypes (**Fig. 3h; Supplementary Fig. 17**). Clusters 2, 5, 6 are enriched for cells from *Atp6v0a2* KO, *Atp6v0a2* R755Q and *Gorab* KO mice and are marked by proepicardium marker *Upk3b* (cluster 2), hepatic mesenchymal markers *Reln*, *Lrat*, *Nr1h5*, and *Gata4* (cluster 5) and lung mesenchyme marker *Tbx5*, *Tbx4*, *Wnt2*, *Fgf10* and *Fmo2* (cluster 6)³¹⁻³⁵. Clusters 3 and 4 are depleted in cells from *Atp6v0a2* KO, *Atp6v0a2* R755Q and *Gorab* KO mice and cluster 3 is marked by gastrointestinal smooth muscle markers *Acta2*, *Myh11*, *Mylk* and *Nkx2-3* (**Fig. 3i,j**)³⁶⁻⁴⁰. Although individually subtle, the consistent shifts in cell type proportions between the two *Atp6v0a2* mutants and the *Gorab* KO mutant across these many subsets of

lateral plate mesoderm-derived mesenchyme presumably underlie the high mesenchymal similarity scores between these three mutants (**Fig. 3i**).”

Because these mesenchymal subsets are quite poorly described in the literature, we sought to further support our annotations by mapping them to a spatially resolved dataset using Tangram. The spatial reference dataset was obtained from the recent Stereo-seq publication²⁹, and we generally observed the subcluster annotations to be supported. For example, cluster 3 was annotated as the gastrointestinal cells and the spatial mapping corresponds to the GI tract in the spatial dataset; cluster 6 was annotated as the lung mesenchyme and the spatial mapping corresponds to the lung in the spatial dataset (**Supplementary Fig. 17**).

Supplementary Figure 17. Spatial mapping results by Tangram showing the most likely physical location of the cells from each cluster in the lateral plate & intermediate mesoderm sub-trajectory on a sagittal mouse section. Top 12 sub-clusters are shown. The colour scale is set from 1st percentile to 99th percentile.

These additional computational analyses provide more granular insight into subtle shifts in cellular composition that are driving the greater similarity scores between these three mutants. As noted above, these changes are not easily experimentally validated by conventional methods, particularly given how relatively understudied derivatives of the lateral plate mesenchyme are at this timepoint in mouse development, compared to other systems. A broader point is that the whole embryo, whole transcriptome scale of this method arguably challenges the field to develop methods which are better matched in terms of scale and resolution to detect subtle effects. For example, we can imagine the very recently described “wild-DISCO” method for whole mouse histology serving this purpose⁴¹.

We have added the following text to the Discussion reflecting on both the challenges and opportunities in this space:

New: “Additionally, some of the phenotypes identified here have probably not been described before due to the lack of resolution of conventional phenotyping approaches. New secondary validations strategies need to be developed to confirm subtle defects in molecular programs or subtle changes in the relative proportions of specific cell types. A very promising approach would be to complement whole embryo sc-RNA-seq with rapidly advancing methods for whole mouse body antibody labelling and 3D imaging⁴¹ .

1-11: With respect to the interpretation of the Sox9 phenotype, the proposed neural-like cell population is interesting and unexpected, again, a finding that could benefit from a simple follow up secondary validation directly on embryos.

Thank you for this comment. To get a better handle on this result, we first performed some additional computational analyses to better understand the finding. We then took advantage of several publicly available spatial transcriptomic datasets to validate the finding, which also provided some greater insight into the spatial distribution of the neural-like population.

First, we asked whether there are other major transcriptome-level gene-expression changes in addition to the misexpression of Sox9 and Kcnj2 (former **Fig. 4d** and **Supplementary Fig. 11**). More specifically, we performed a gene-set enrichment analysis, comparing the “bone” cells (cells from the Chondrocyte, Osteoblast, and the Limb Mesenchyme trajectories) between the mutant and the background-matched wild type, which shows a clear downregulation of Hallmark gene sets implicated in chondrogenesis, including protein_secretion and hedgehog_signalling. We also identified upregulation of six out of seven immune-related signalling pathways in the mutant. Major changes to the pathways related to proliferation are also apparent. These results have been added as a new **Supplementary Fig. 19**, as well as some new corresponding text:

New: “Since Sox9 encodes for a transcription factor, which among other roles induces the differentiation and proliferation of the chondrogenic lineage, we sought to explore other effects of this regulatory mutant by performing a single-sample gene set enrichment analysis (ssGSEA)⁴² of bone cells, comparing the mutant to the WT (**Supplementary Fig. 19**). This revealed that many of the hallmark pathways⁴³ downregulated in the mutant are related to proliferation and chondrocyte differentiation, including mitotic spindle, TGF-β signalling, notch signalling, wnt/β-catenin signalling, protein secretion, epithelial-to-mesenchymal transition, etc. Sox9 is known to mediate many of these pathways^{44–47}. On the other hand, 6/7 immune-related hallmark pathways were upregulated in the mutant, possibly a secondary effect, as to our knowledge Sox9 is not established to be involved in immune signalling.”

Supplementary Figure 19. Gene set enrichment analysis on bone cells. Comparison of median ssGSEA⁴² scores between the Sox9 regulatory INV and wild type for Hallmark gene sets. Gene sets categorised as proliferation and immune-signalling⁸⁵ highlighted in blue and brown. Gene sets manually identified to be implicated in chondrogenesis highlighted in red.

Next, we further investigated the neural-like cell population we had identified in the mutant, in particular focusing on quantifying if and by how much this sub-trajectory was enriched, in terms of absolute cell numbers or proportionally, in the mutants. This required re-clustering of the undifferentiated mesenchyme, this time also including the G4 wild type cells. As a result, we obtained four branches of cells (instead of two branches as in former **Fig. 4f**), of which “Branch 2” exhibited the neural-like transcriptome (**Fig. 4g**). The new panel **a** in **Supplementary Fig. 20** shows that the mutant embryos contain 8.7 times (accounting for the total number of cells in the datasets) as many “neural-like” cells in the mesenchyme. However, there is an overall increase in undifferentiated mesenchyme in the Sox9 mutant, such that the proportion of neural-like Branch 2 is only modestly higher (26% vs 16%). So, while cells corresponding to all four branches are observed in the wildtype embryo, this particular branch is increased in the Sox9 mutant by the combination of increased undifferentiated mesenchyme together with proportional changes within undifferentiated mesenchyme.

Figure 4e-g. **e**, RNA velocity on UMAP embedding of G4 wildtype and Sox9 regulatory INV cells in the limb mesenchymal trajectory labelled by annotation (left) or sample (right). **f**, UMAP embedding of Sox9 regulatory INV and wild type cells in the undifferentiated mesenchyme, visualised in the same embedding as in panel **e**. **g**, Dot plot of the top six (where available) significantly differentially expressed genes in the four branches.

Supplementary Figure 20a. Proportion and the number of cells at different levels of clustering, leading up to the four branches of the undifferentiated mesenchyme.

Can we verify and locate these ‘neural like’ mesenchyme cells? Although we do not have whole embryo spatial transcriptomic data on Sox9 mutants, we were able to leverage the aforementioned Stereo-seq data (E13.5) to map this “neural like” mesenchyme, given that this branch is present (albeit at lower numbers) in the wildtype as well. The results from this analysis are shown in the new **Supplementary Fig. 22**. Specifically, we performed spatial mapping to infer the physical location of each ‘branch’ split by mutant and wildtype identity. In contrast with the other branches, branch 2 clearly maps to the neural tube and brain regions.

Supplementary Figure 22. Spatial mapping of the cells of the undifferentiated mesenchyme onto Stereo-seq dataset. Tangram inferred locations of cells from each branch shown in **Fig. 4f**, split by mutant (top) and wildtype (bottom) identity. The colour scale is set from 1st percentile to 99th percentile.

However, although the Stereo-seq data is very high resolution spatial transcriptomics, it is not ‘formally’ single cell resolution. As a consequence, there is a risk that the strong mapping of Branch 2 (neural-like mesenchyme) to the neural tube and brain regions shown in **Supplementary Fig. 22** is an artefact of shared gene expression between this mesenchymal branch and neural derivatives. To address this, we integrated our dataset with another spatial transcriptomics dataset on mid-gestational mice (E14.5), based on the sci-space method⁴⁸, in which a subset of transcriptionally profiled nuclei have known physical locations in sagittal sections within which they were mapped prior to sc-RNA-seq. For each mutant cell in the 4 branches, we find the nearest neighbour in sci-space data in the integrated co-embedding, and plot the location of the neighbouring sci-space cell where it is known (new **Supplementary Fig. 23**). Across multiple sections, neighbours of “branch 0” cells are aggregated in the limb regions, and neighbours of neural-like “branch 2” cells are aggregated in the brain region. Branch 1 and branch 3 cells do not exhibit a clear pattern, other than notably being largely excluded from the brain regions.

Supplementary Figure 23. Integration of the cells of the undifferentiated mesenchyme with sci-space dataset. Sox9 Regulatory INV cells in the undifferentiated mesenchyme were integrated with a spatial transcriptomics dataset on mid-gestational mice (E14.5), generated via the sci-space method⁴⁸, in which a subset of transcriptionally profiled nuclei have known physical locations in sagittal sections within which they were mapped prior to sc-RNA-seq. We find the nearest neighbour of each Sox9 Regulatory INV cell in sci-space data in the integrated co-embedding, and plot the location of the neighbouring sci-space cell where it is known (red dots). Red arrows highlight areas with aggregated cells (branch 0 matches with the limbs and branch 2 matches to the brain).

These data provide secondary support, based on two spatial transcriptomics datasets obtained via two different techniques, that “branch 2” is not an artefact, and furthermore are enriched in the brain region, even in wildtype mice. The following text has been changed in the main text to reference these additional findings:

Previous text (Lines 560-569): “To investigate these two branches further, we performed sub-clustering of Sox9 regulatory INV undifferentiated mesenchyme cells, followed by differential expression analysis (**Fig. 4f,g**). Interestingly, the most differentially expressed genes in “branch 1” were neuronal, e.g. several neurexins and neuregulin 3, an observation that was supported by single-sample gene set enrichment analysis (ssGSEA)⁴², which further highlighted KRAS and other signalling pathways (**Fig. 4g; Supplementary Fig. 13b,c**). Of note, mesenchymal stem cells can be differentiated to neuronal states *in vitro*⁴⁹. Although further investigation is necessary, we note that cells contributing to “branch 0” as well as the neuronal-trending “branch 1” are present in wildtype embryos, albeit at much reduced frequencies compared to the Sox9 regulatory INV mutant (**Supplementary Fig. 13a, left**).”

New: “To investigate these branch-like structures further, we performed sub-clustering of undifferentiated mesenchyme cells from the mutant and wild type, followed by differential expression analysis (**Fig. 4f,g**). Interestingly, the most differentially expressed genes in “branch 2” were largely neuronal, e.g. several neurexins and neuregulin 3, an observation that was supported by gene set enrichment analysis (**Fig. 4g; Supplementary Fig. 21c,d**). A cellular composition analysis revealed that these neuronal-like cells were not restricted to the mutant,

but were also found in the wildtype embryos as well, albeit at much reduced numbers compared to the *Sox9* regulatory INV mutant (**Supplementary Fig. 21a,b**). As a means to verify this unexpected ‘neural-like’ branch of mesenchymal cells as well as to assess their anatomical distribution, we first we mapped these mesenchymal cells to spatial transcriptomic data from the mouse embryo (E13.5) obtained via Stereo-seq²⁹ using the Tangram algorithm²⁹. Strikingly, this analysis placed the cells of the branch 2 along the neural tube and the brain regions (**Supplementary Fig. 22**). To address concerns that artefacts might arise from mapping single-cell data onto Stereo-seq maps (which are high resolution but not formally single cell resolution), we also integrated our data with sci-space⁴⁸ spatial transcriptomic data obtained from E14.5 embryos, as these spatial data are sparser but retain single nucleus resolution. The results are consistent, in that branch 2 mesenchymal cells are enriched in brain regions (**Supplementary Fig. 23**). In contrast, branch 0 mesenchymal cells are enriched in limb bud regions, while branch 1 & 3 cells are diffusely distributed but largely excluded from brain regions (**Supplementary Fig. 23**). Taken together, the differential expression analysis, gene set enrichment analysis and spatial mapping based on two independent technologies/datasets, support the validity of this neural-like subset of mesenchyme (present in wildtype mice and increased in the *Sox9* regulatory INV mutant), which to our knowledge has not been previously reported. The observation is consistent with the reports that mesenchymal stem cells can be differentiated to neuronal states *in vitro*⁴⁹.”

We are in the process of pursuing ‘whole embryo’ spatial transcriptomics on *Sox9* to further characterise this cell population but argue that this is beyond the scope of this manuscript.

1-12: The sharing amongst a rather eclectic set of largely unrelated mutants in the final section (supplementary Figures 14-16) is indeed surprising as the authors note. It seems more likely this represents a technical issue than uncovering “sharing of molecular and cellular sub-phenotypes across mutants. Again, the authors can strengthen confidence in the approach and move tentative conclusions to more solid ground by investing more in validating predictions from the data.

As noted above (see response to comment 1-10), aspects of sharing between the *Gorab* KO, *Atp6v0a2* R755Q, and *Atp6v0a2* KO mutants make sense in light of the literature and our further computational analyses. The eclectic/surprising aspect highlighted by the reviewer in this comment (and covered in the cited supplementary figures of the original manuscript) relate to an apparently similar accumulation of undifferentiated mesenchyme in these mutants as well as the *Sox9* regulatory mutant. Overall, we feel that the former aspects of inter-mutant sharing (*i.e.* *Gorab* KO, *Atp6v0a2* R755Q, and *Atp6v0a2* KO mutants) are on more solid ground than this finding, which we have not yet been able to validate due to our bandwidth being focused on the other validations and associated analyses. For that reason, we have removed this section and the corresponding supplementary figures from the manuscript.

Referee #2 (Remarks to the Author):

2-0: Huang and colleagues have undertaken the impressive challenge of comprehensively mapping embryonic phenotypes in whole embryos by leveraging tools from high throughput single-cell RNA sequencing (sci-RNA-seq), mouse genetics, and both established and newly developed single-cell analytical methods. The collaborative study combines expertise from several groups and aims to develop a standardized framework for unbiased interpretation of mammalian developmental defects, including pleiotropic defects. To accomplish this task, the authors collected and analyzed single-cell RNA seq profiles from 1.6M nuclei spanning 101 E13.5 mouse embryos, 26 genotypes, and 4 mouse strains. The authors have analyzed this massive dataset using several interrelated paradigms: (1) quantification of changes in cell type composition between genotypes, (2) quantification of wt/mutant ratio changes across cell state neighborhood that spanning the full range of detected embryonic cell types/tissues, and (3) systematic quantification of shared features between genotypes. Among the 26 genotypes analyzed, the authors highlight examples of phenotypes in which specific tissues fail to differentiate (Ttc21b, ZRS limb enhancer), and one case that involves the generation of a novel mutant-specific state (Sox9 INV). The study is both ambitious and well written. However, in its present form, the study has some serious weaknesses that would need to be fully addressed before publication could be considered. Most importantly, the authors themselves acknowledge that some of the most prominent trends in their MMCA analysis (Fig 2a) might be artifacts driven by a skewed wildtype baseline comparison. In addition, there are some important unresolved questions about the interpretability of the lochNESS scores. Given that both potential artifacts and ambiguity surround some of the central claims of this study, it will be imperative for authors to address these concerns by performing additional wildtype baseline comparisons and benchmarks of existing or new single-cell data (see below) and/or provide more extensive *in vivo* validation of claims that are not yet convincingly supported by their scRNA-seq data alone.

Thanks for your positive comments on our manuscript and for providing constructive feedback regarding adding baseline comparisons. Overall, to address the main concerns raised by this reviewer (some of which overlap with other reviewers), we have: 1) performed additional computational analysis towards ruling out artefacts and establishing a better wildtype baseline; 2) performed additional computational analysis to further characterise the phenotypes we observed in several mutants; and 3) performed validation experiments to support a subset of phenotypes identified through computational analysis. Details on the additional analyses and validations are listed in response to the specific comments below.

Major comments:

2-1: In Supplemental Figure 5, there appear to be wide ranges in the recovery efficiencies for each cell type across both genotype and strain(s). Given how heavily the authors' MMCA approach relies on the interpretation of cell type proportions, it would be helpful if the authors exhaustively examined any other possible sources of technical effects or biological effects that might confound cell type recovery rates. Similarly to Supplemental Figure 5c, can the authors also break down the recovery rates of each trajectory by lab of origin/ cryopreservation protocol, experimental batch, etc?

These are reasonable concerns. Towards addressing them, we first performed ANOVA for each main trajectory, to test if the % of cells are different across backgrounds (*i.e.* mouse strains). The adjusted p-values are listed in the updated **Supplementary Fig. 5c**. We identified that the cell type proportions from many of the 13 trajectories are significantly different (albeit modestly) across the four backgrounds (adjusted p-value < 0.05, by ANOVA), indicating that background is a factor that might potentially confound cell type recovery rates. Of note, that's also why we sought to consistently compare samples between different genotypes within the same background throughout the whole manuscript (*e.g.* **Fig. 2a**).

Supplementary Figure 5c. Boxplots of cell proportions falling into each of the 13 major trajectories for the four wildtype strains. Each point corresponds to an individual embryo. The total number of cells from each major trajectory profiled from wildtype embryos and the adjusted p-value by ANOVA across different backgrounds are also listed.

As the reviewer suggested, in addition to the background, we checked three additional potentially confounding variables, including: a) the lab of origin; b) the experimental batch; c) the sex, which might potentially affect the cell type proportions for the individual embryo samples. The corresponding boxplots are shown in the new **Supplementary Fig. 6a**. Then, we performed a similar ANOVA tests to quantitatively check if cell type proportions are significantly affected by those three factors (new **Supplementary Fig. 6b**):

1. **Lab of origin:** Our mice samples were collected from 6+ different places (**Supplementary Table 1**). For many trajectories, the cell proportions are significantly different across different sources (top panel in new **Supplementary Figure 6b**). However, as samples with the same perturbation genotype were collected from the same origin, resulting in the two variables, “genotype” and “sample origin”, being highly overlapped. Unfortunately, it’s impossible to definitively disentangle these, although it’s challenging to imagine how the sample origin would shift these proportions. We remind the reviewer that all embryos were processed together in terms of the sc-RNA-seq in the context of a few sci- experiments. Furthermore, the extraction of nuclei from embryos was randomised across mutant and strain backgrounds and performed in a standardised fashion by one person in a single lab (in Seattle).
2. **Experimental batch:** We performed four sci-RNA-seq experiments, during which samples with the same genotype & background were profiled across different experiments. Thus, the experimental batch could be a potential factor that affects our results, and we saw the cell type proportions are different across experimental batches for many trajectories (middle panel in new **Supplementary Figure 6b**). We suspected that such a difference could be due to another important variable - background (*i.e.* mouse strain), so we split the samples to different strains, and repeat the ANOVA tests for subset of samples within the same background. We didn’t identify any significant differences across experimental batches (new **Supplementary Figure 6c**), suggesting that the experimental batch doesn’t affect our comparison

of samples within the same background. Of note, all the four experiments were performed by the same person and in a standardised fashion in a single lab.

3. Sex: We find there is no significant effect driven by different sex (bottom panel in new **Supplementary Figure 6b**).

In summary, we cannot exclude the potentially technical effects due to mice being collected from different resources, but we didn't identify any concerning batch effects driven by different experimental batches or different sex of samples within the same background.

Supplementary Figure 6. Cell composition for individual wildtype and mutant embryos across developmental trajectories, from different technical or biological groups. a, Boxplots of cell proportions falling into each of the 13 major trajectories from different sample origins (left), experimental batch (middle), or sex (right). Each point corresponds to an individual embryo. In the boxplots, the centre lines show the medians; the box limits indicate the 25th and 75th percentiles;

the replicates are represented by the dots. **b**, ANOVA was performed on cell proportions falling into each of the 13 major trajectories from different sample origins (top), experimental batch (middle), or sex (bottom), and the minus log₁₀-scaled adjusted p-values have been shown. The red horizontal line corresponds to significant cutoff (0.05). **c**, ANOVA was performed on cell proportions falling into each of the 13 major trajectories from different experimental batches after subsetting samples from C57BL/6 (top), FVB (middle), or G4 (bottom), and the log₁₀-scaled adjusted p-values have been shown. The red horizontal line corresponds to significant cutoff (0.05). NC: neural crest. PNS: peripheral nervous system. SN: sensory neuron.

2-2: Main Fig 2a. Some cell types are responding in a stereotyped fashion regardless of perturbation, usually all going in the same direction with respect to the strain-matched wt control. For example, the Brain Endothelial Trajectory increases in abundance following any of the genetic perturbations performed in the C57BL/6 background, but in none of the perturbations from the other strains. A similar trend exists for Aorta-Gonad-Mesonephros and Endocardium but in the opposite direction, with these cell types nearly always decreasing in abundance. Could the authors rule out the stated possibility that these signatures are simply an artifact (e.g. driven by the cell type recovery rates in the C57BL/6 control to which all of these perturbations are compared), by (1) comparing to additional controls, and (2) providing in vivo validation of these phenotypes? These strain-specific effects are arguably the most pronounced trends presented by the authors in Fig 2a. If the reader cannot be convinced that the most striking results from this analysis are not technical artifacts, the authors will not succeed in establishing MMCA as a reliable or interpretable method. If proper controls do ultimately suggest that these trends indeed reflect genuine biology, it would be interesting to investigate further how such stereotyped responses arise from such varied perturbations, e.g. do they reflect stress, developmental delay, or some other generic response to developmental mispatterning?

Thank you for pointing this out, particularly as this comment led to a major update to **Fig. 2a**. We believe that the issue was in part due to the limited number of control replicates ($n = \sim 4$) used in each comparison (*i.e.* calculating log₂ transformed ratios of the cell proportions between each mutant type and its corresponding wildtype background), thus any potential outlier of the control samples will change the result abruptly. To this end, for the revision, we expanded our control samples through a technical adjustment, by including not only wildtype but also other mutant types from the same strain ($n = \sim 30$ for C57BL/6, $n = \sim 35$ for FVB, $n = \sim 20$ for G4). To do this analysis, we assume compositions of cells across cell-types for samples: (1) *from different strains* are following different distributions (**Supplementary Fig. 5**); this is also supported by the transcriptional stratification among different backgrounds that we observed through the pseudobulk analysis (**Fig. 1b**); 2) *from the same strain*, regardless of its genotype, are roughly following the same distribution. This strategy is also consistent with how we performed the beta-binomial regression analysis, in which we included all the samples within the same strain to gain larger statistical power.

In the updated **Fig. 2a**, we find that our new strategy of increasing the sample size in the background group makes the results much more robust. First, most of the key results are repeated with the previous findings (**Revision Fig. 2a**). Second, for the sub-trajectories highlighted by the reviewer (Brain endothelial, Aorta-Gonad-Mesonephros, Endocardium), the potential artefacts have disappeared. Finally, compared to the previous result, we noticed that the effect sizes for most of the FVB mice are decreasing, which is expected (**Revision Fig. 2b**).

Of note, based on our power analysis (see details in our response to Reviewer 1-4 and **Supplementary Fig. 7**), our current sample size (four samples in each group) would be sufficient to detect a 25% effect size for those cell types that present at a 1% proportion in wildtype embryos. We realised that it might be challenging to detect the minor changes in each mutant. As we have highlighted in the updated **Fig. 2a**, those trajectories with proportions >1% might be much more reliable than those trajectories with proportions <0.1%.

Figure 2a. Heatmap shows \log_2 transformed ratios of the cell proportions between each mutant type (y -axis) and a pooled reference (including both wildtype and other mutants from the strain), across individual sub-trajectories (x -axis). Only those combinations of mutant and sub-trajectory which were nominally significant in the regression analysis are shown (see text and **Methods**; uncorrected p -value < 0.05 ; beta-binomial regression). The number of cells assigned to each developmental trajectory in the overall dataset is shown above the heatmap. The thresholds of sub-trajectories with proportions $<10\%$, $<1\%$, and $<0.1\%$ are highlighted by the red vertical lines.

Revision Figure 2. Comparing the different strategies of calculating the \log_2 transformed ratios of the cell proportions between each mutant type and its corresponding background. **a**, For those 300 significant items (uncorrected p -value < 0.05 ; beta-binomial regression) shown in the heatmap of **Fig. 2a**, we plot the \log_2 ratios calculated by the new strategy (x -axis, using the wildtype from the same strain as the background) vs. the old way (y -axis, using the wildtype plus the other mutants from the same strain as the background). **b**, For a subset of the 68 significant items (uncorrected p -value < 0.05 ; beta-binomial regression) from the FVB strain, we compared the absolute value of \log_2 ratios calculated by the new strategy vs. the old strategy.

2-3: The authors' lochNESS analyses are a bit confusing and could benefit from further benchmarks / explanation. First, an internal check using wildtype samples (e.g. comparing between different wildtypes) would provide the reader with a better intuition on what a "remarkable" vs an "unremarkable" score distribution should look like. My expectation was that an unremarkable / null distribution should center on a score of zero, much like the random permuted data in Supplemental Fig 8a. The authors should demonstrate this explicitly by constructing histograms for wildtype vs wildtype comparisons and determine whether the resulting distributions of lochNESS scores center on zero, or some other value. According to the authors' interpretation, the TAD boundary knockouts are largely "unremarkable" but in Fig 3c (and Supplemental Fig 9a), many of these histograms appear to skew heavily towards negative lochNESS scores. Based on the how the lochNESS scores are calculated and the logic presented in Fig 3a, negative lochNESS scores should indicate that substantial numbers of wildtype-like cell states are not being populated in these mutants, much like the case for the ZRS limb enhancer KO in which a key *msx/hox* state is depleted (Fig 3b, Fig 2d-e). By this logic, the TAD KO samples would all be displaying similar lochNESS trends as the ZRS enhancer KO, suggesting that these perturbations in fact similarly eliminate some cell states. Can the authors explain this discrepancy? Are the TAD KO perturbations depleting specific cell types, or not? If not, where are these negative lochNESS scores coming from? If negative lochNESS scores are, in fact, unreliable or less reliable than positive scores, the authors should explain this clearly to the reader, ideally with data-driven benchmarks.

Thank you for your detailed comment and suggestions.

To conduct an internal check with wildtype samples, we calculated lochNESS and similarity scores to assess the baseline variation as suggested (**Supplementary Fig. 10d**). Indeed, for three of the background strains, G4, FVB and BALB/C, the lochNESS distribution is centred around zero similar to random permuted data. Comparisons that include the C57BL/6 wildtype mice show slightly skewed distributions, indicating potential strain-specific distributions in C57BL/6 wildtype mice. However, compared to the skews observed in mutant vs. wildtype comparisons (**Supplementary Fig. 10a**, lochNESS plotted in range [-2,4]), the between wildtype variability is relatively small (**Supplementary Fig. 10d**, lochNESS plotted in range [-1,1]).

Supplementary Figure 10a,d. **a**, Distribution of lochNESS across all 64 sub-trajectories in each mutant. **d**, Estimated density graphs of lochNESS shows distribution of lochNESS in wildtype comparisons. Each comparison is labelled by the strain treated as the 'mutant', followed by the strain treated as the reference (i.e. G4 vs. FVB indicates that G4 was treated as the 'mutant' in the comparison).

We now reference this new figure panel in the following sentence:

“In addition, we computed lochNESS between wildtypes from different background strains and observed minimal variation in cell distribution between wildtype from G4, FVB and BALB/C strains and potential strain-specific distributions in C57BL/6 wildtype mice (**Supplementary Fig. 10d**).”

Another type of "wildtype baseline" which we have already constructed (without restricting to only wildtype samples) is based on the similarity scores between different embryos (**Supplementary Fig. 15b**), where the "between genotype" scores would correspond to a "remarkable" distribution (low similarities) and the "within embryo" scores would correspond to an "unremarkable" score distribution (high similarities).

Supplementary Figure 15b. Boxplot showing the similarity scores of comparisons between embryos of different genotypes (left), between embryos of the same genotype (middle), and within the same embryos (right) for C57BL/6 genotypes in the mesenchymal trajectory.

Regarding the latter part of the comment concerning the interpretation of TAD boundary knockouts, we indeed found several potential factors that would impact our interpretation. In former **Fig. 3c** (now **Supplementary Fig. 10a**), the background-specific patterns are potentially due to our choice of using a pooled wildtype as control instead of a background-specific reference. When updating to use background-specific controls, the resulting distributions are as follows (**Revision Figure 3**). In this view, we no longer observe the negative skew in TAD boundary KO mutants.

Revision Figure 3. Distribution of lochNESS with strain-specific references. Estimated density graphs of lochNESS shows distribution of lochNESS across all 64 sub-trajectories in each mutant. A strain specific wildtype was used as control in lochNESS calculation.

Additionally, the interpretation of such a global view is confounded by the fact that larger trajectories contribute many more cells to the estimated distributions shown. Therefore, the trajectory-specific distributions focusing on specific sub-trajectories in specific mutants (formerly **Fig. 3d**, now **Fig. 3c**) is more informative. Overall, we believe the negative skew in the TAD KO indicates a background specific bias, instead of true depletion. What remains informative is the distance of lochNESS to the empirical null distribution of lochNESS we generated through perturbation (**Supplementary Fig. 10a-c,e**), which shows that the changes in TAD KO are generally insignificant.

Figure 3c (formerly Fig. 3d). Distribution of lochNESS in the neural tube sub-trajectories of the *Ttc21b* KO and *Gli2* KO mutants. Dashed boxes highlight the shifted distributions of the retinal neuron sub-trajectory of the *Ttc21b* KO mutant and the floor plate and roof plate sub-trajectories of the *Gli2* KO mutant.

Since we agree that the interpretation based on such a global graph alone may be misleading, we have moved the former **Fig. 3c** to **Supplementary Fig. 10a** to avoid confusion, such that former **Fig. 3d** has become **Fig. 3c** (reproduced above).

2-4: The authors' interpretation of the Gli2 KO phenotype, particularly the differing effects on "two subpopulations of roofplate cells", should be followed up in further detail. The authors have not established that this second mutant-enriched cluster is indeed roofplate. Especially given the relatively shallow UMI-per cell sampling depth of the authors' sci-Seq method, this could simply be an unrelated cell type that shares superficial transcriptional similarity to the roofplate. The authors should perform *in vivo* validations with additional marker gene *in situ* hybridizations that distinguish the two candidate roofplate clusters in both wildtype and mutant embryos, to verify cell type identities, anatomical locations, and opposite responses to the Gli2 KO.

Thank you for this important comment. Our response to this comment is notably identical to the related comment 1-7 above (*i.e.* comment 7 from Reviewer 1).

Previous publications on Gli2 knockouts show the lack of floor plate development and further a malformation of the neural tube, due to disrupted Shh signalling pathways^{15,16,17,18}. However, several studies in which Gli2 knockouts are analysed, effects on the roof plate or roof plate derivatives, were either not described or excluded^{4,12,15}.

To investigate the changes we describe in the floor and roof plates of the Gli2 KO mutant in greater detail, we performed both additional computational analyses as well as new experimental validations. First, we mapped the three clusters we identify in **Supplementary Fig. 11a** (formerly **Supplementary Fig. 8d**) to recently published spatially-resolved wildtype mouse embryo transcriptomics data (E13.5) using the ML-based Tangram algorithm (**Supplementary Fig. 11e**). Cells from cluster 1, which we previously labelled as floor plate, were spatially mapped to the floor of the neural tube, supporting our annotation. Clusters 2 and 3, which we previously labelled as roof plate, did not map to the traditional roof plate location in the roof of the neural tube, but instead mapped to distinct areas within the developing brain. Cluster 3 mapped to the choroid plexus (ChP), a structure that is derived from the roof plate and develops in both the lateral ventricle (anterior) and 4th ventricle (posterior) of the developing brain, while cells from cluster 2 mapped next to the lateral ventricle ChP. This is consistent with the strong expression of *Ttr* in cluster 3, which is the classic marker for the ChP, and expression of *Wnt8b/9a/3a* in cluster 2, which are genes expressed in the cortical hem, a roof plate derivative attached to the lateral ventricle ChP (**Fig. 3e, Supplementary Fig. 11d**)^{19,20,21}. We then investigated gene expression markers that further separate lateral ventricle and 4th ventricle of ChP²² and found that in addition to roof plate marker *Lmx1a*, cluster 3 expresses 4th ventricle marker *Meis1* and cluster 2 expresses lateral ventricle markers *Otx1* and *Emx2* (**Supplementary Fig. 11f**).

In summary, cluster 1 remains annotated as the floor plate, cluster 2 is now annotated as the cortical hem of the lateral ChP, and cluster 3 is now annotated as the lateral and 4th ventricle ChP. Our analyses of these whole embryo sc-RNA-seq-derived clusters with lochNESS suggest that both the floor plate (cluster 1) and cortical hem (cluster 2) are (relatively) reduced in cell numbers in the Gli2 KO, while the ChP (cluster 3) is increased.

Towards secondary validation of the floor plate (expected) and roof plate (unexpected) observations, we investigated these structures in Gli2 KO mutants and wildtype mice. In coronal sections through the developing brain and neural tube, the *Gli2* KO mutant showed severe developmental defects such as deformed forebrain lobes and delayed neural tube development (**Supplementary Fig. 12a**). Focusing on the neural tube, the

expected location of the floor and roof plates, the *Gli2* KO mutant revealed a severely disturbed shape upon immunofluorescence imaging of *Pax6*, a neural tube developmental marker (**Supplementary Fig. 12b**).

In particular, the *Gli2* KO mutants exhibit the well-described “dorsalization” of the neural tube, which is complemented by our independent determination (by whole embryo sc-RNA-seq) that the floor plate is markedly reduced in the *Gli2* KO mutant. This effect has been previously described in the *Gli2* KO mutant, caused by the disruption of the Shh signalling pathway¹⁵. Consistent with this, we observe downregulated Shh in the floor plate of the *Gli2* KO mutant, relative to WT, as well as of target genes of this pathway including *Nkx2.2*, *Foxa2*, *Olig2*, *Ptch1* and *Smo* (**Revision Fig. 1**)^{23,24}. Additionally, *Gli3*, known to function as a repressor for *Gli2* activity in the neural tube²⁵, is downregulated in the *Gli2* KO mutant. To summarise, direct Shh pathway targets, as well as pathway-relevant receptors and repressors, are downregulated in the *Gli2* KO floor plate, emphasising the deregulation of the pathway upon mutation of the *Gli2* gene, leading to the described ‘dorsalization’ phenotype.

Revision Figure 1. Dotplot showing normalised expression of and percent of cells expressing Shh and genes in the Shh pathway of the neural tube, focusing on cells in cluster 1 (floor plate) of *Gli2* KO and wildtype cells.

Examining the ChP in more detail, *Ttr* staining in mutant and wildtype sections of the lateral ventricle as well as the 4th ventricle shows that the shape of the tissue (‘branching’) as well as the development of the single cell layer is disturbed within the mutant, revealing a ‘double dapi’ layer in the mutant compared to the wildtype, which displayed *Ttr* signal consistently in a single layer (**Fig. 3f; Supplementary Fig. 13, red arrows**). Because the *Gli2* KO mutants are overall smaller in size at this stage, we normalised the *Ttr* positive cells of the lateral and 4th ventricle to the size of the embryo section area (in μm) and found the mutants to have an increased proportion of *Ttr* positive cells (**Supplementary Fig 13c; Supplementary Table 7**). These imaging-based observations indicate a relative enrichment of cell counts in the ChP in *Gli2* KO mutants, consistent with the lochNESS results.

Taken together, these secondary studies validate our findings in the *Gli2* mutant of differences in both the floor plate and the ChP, a derivative of the roof plate. Unfortunately due to multiple dysfunctioning antibodies, we have not yet been able to validate the changes within cluster 2, the cortical hem, directly. However, given the extensive deformations in the embryo in this region and that the other two related observations (cluster 1, cluster 3) were successfully experimentally validated, we are reasonably confident that the changes in the juxtaposed cortical hem (cluster 2) are valid as well. The relevant figure panels of the revised manuscript are reproduced below:

Figure 3d-f. **d**, UMAP visualisation of co-embedded cells of the floor plate and roof plate sub-trajectories from the *Gli2* KO mutant and pooled wildtype, coloured by lochNESS. ChP = choroid plexus. **e**, same as in panel d, but coloured by expression of selected marker genes. **f**, Choroid plexus marker *Ttr* staining of the developing brain regions (LV = lateral ventricle, 4V = 4th ventricle, ChP = choroid plexus) in H&E sections of wildtype and *Gli2* ^{-/-} mutants in 20x magnification.

Supplementary Figure 11. Analysis of *Gli2* KO in the roof plate and floor plate trajectories. **a**, UMAP visualisation of co-embedded cells of the floor plate and roof plate sub-trajectories from the *Gli2* KO mutant and pooled wildtype, coloured by sub-trajectory (left) or cluster number (right). **b**, Boxplot showing the lochNESS distribution in each cluster shown on the right of panel **a**. **c**, Barplots showing the cell composition of each cluster shown on the right of panel **a**, split by mutant vs. wildtype (left) or individual embryo (right), with a reference line at the overall wildtype cell proportion. **d**, Dotplot summarising the expression of and percent of cells expressing selected marker genes in each cluster shown on the right of panel **a**. **e**, Tangram-inferred locations of each cluster shown on the right of panel **a**. Red arrows highlight the areas where cells map to with high probability. The colour scale is set from 1st percentile to 99th percentile. **f**, UMAP visualisation of co-embedded cells of the floor plate and roof plate sub-trajectories from the *Gli2* KO mutant and pooled wildtype, coloured by expression of marker genes.

Supplementary Figure 12. Morphological phenotype of *Gli2*^{-/-} mutants. **a, H&E staining of two mutant and two wild type E13.5 embryos in cranial-caudal (1-3) order within the head. In order to compare mutant and wildtype slides in neural tube development, the slides are matched based on hallmarks such as eyes, tongue muscle and nasal cavities. **b**, Neural tube marker *Pax6* staining of the developing neural tube in consecutive sections of H&E sections 1.3 and 2.3 to visualise the structure of the neural tube formation in wildtype and mutant in 10x and 20x magnification.**

Supplementary Figure 13. *Ttr* staining in wildtype mice and *Gli2*^{-/-} mutants. Choroid plexus marker *Ttr* staining of the developing brain regions (LV = lateral ventricle, 4V = 4th ventricle, ChP = choroid plexus) in H&E sections of **a**, wildtype and **b**, *Gli2*^{-/-} mutants in 2x, 10x and 20x magnification. For each section (2x magnification), the regions of interest are highlighted with white boxes and shown in higher magnification on the sides (10x or 20x magnification). Red arrows highlight areas with normal single layer of *Ttr* expressing cells in wildtype, and two layers of cells in the mutant

Relevant sections of the text have also been revised as follows:

Previous text (lines 355-370): “Overall, these observations are consistent with the established role of *Gli2* in floor plate induction and the previous demonstration that *Gli2* knockouts fail to induce a floor plate^{15,17}. Less expectedly, this focused analysis also revealed two subpopulations of roof plate cells, one depleted and the other enriched for *Gli2* KO cells (**Fig. 3e**; **Supplementary Fig. 8d-f**). To annotate these subpopulations, we examined genes whose expression was predictive of lochNESS via regression (**Methods**). The mutant-enriched group of roof plate cells was marked by *Ttr*, a marker for choroid plexus¹⁹ and dorsal roof plate development²⁶, as well as genes associated with the development of cilia (e.g. *Cdc20b*, *Gmnc*, *Dnah6* and *Cfap43*), while the mutant-depleted group was marked

by Wnt signalling-related genes including *Rspo1/2/3* and *Wnt3a/8b/9a* (**Fig. 3f; Supplementary Fig. 8g; Supplementary Table 3**)²⁷. It has been shown that ventrally-expressed *Gli2* plays a central role in dorsal-ventral patterning of the neural tube by antagonising Wnt/Bmp signalling from the dorsally-located roof plate²⁸. Our results are consistent with this, and also define two subpopulations of roof plate cells on which *Gli2* KO appears to have differential effects.”

Now: “Overall, these observations are consistent with the established role of *Gli2* in floor plate induction, its role as activator of *Shh* in dorso-ventral patterning of the neural tube and the previous demonstration that *Gli2* knockouts fail to properly induce a floor plate^{15,17} .

Less expectedly, this focused analysis also revealed two subpopulations of roof plate derivative cell types, one relatively depleted and the other relatively enriched in *Gli2* KO embryos (**Fig. 3d; Supplementary Fig. 11a-c**). To annotate these subpopulations, we examined genes whose expression could be predicted by lochNESS via regression (**Methods**). The mutant-enriched group of roof plate cells was marked by *Ttr*, a marker for choroid plexus¹⁹, as well as genes associated with the development of cilia (e.g. *Cdc20b*, *Gmnc*, *Dnah6* and *Cfap43*), while the mutant-depleted group was marked by Wnt signalling-related genes including *Rspo1/2/3* and *Wnt3a/8b/9a*, indicating it to be a region close to the choroid plexus of the lateral ventricle, namely the cortical hem (**Fig. 3f; Supplementary Fig. 11d; Supplementary Table 4**)²⁷. To confirm these annotations, we mapped the three clusters shown in **Supplementary Fig. 11a** to spatial transcriptomic data from the mouse embryo (E13.5) using the Tangram algorithm²⁹ (**Supplementary Fig. 11e**). Cells from cluster 1 mapped to the floor of the neural tube, consistent with their annotation as floor plate. Cells from cluster 2 mapped next to the lateral ventricle choroid plexus (ChP), while cells from cluster 3 mapped to the ChP, both in the lateral (anterior) and 4th (posterior) ventricles, supporting those annotations as well. We then investigated gene expression markers that further separate lateral ventricle and 4th ventricle of ChP²² and found that in addition to roof plate marker *Lmx1a*, cluster 3 expresses 4th ventricle marker *Meis1* and cluster 2 expresses lateral ventricle markers *Otx1* and *Emx2* (**Supplementary Fig. 11f**).

To experimentally validate these findings, we investigated the developmental progress of the neural tube and brain in stage E13.5 *Gli2* KO mutant and WT embryos. In coronal sections of the mutant, we observed severe developmental defects including deformed forebrain lobes and delayed neural tube development compared to the wildtype (**Supplementary Fig. 12a**). More detailed immunofluorescence imaging of *Pax6*, a neural tube developmental marker, revealed a severely disturbed shape of the neural tube, confirming the well-described “dorsalization” phenotype of the neural tube in *Gli2* KO mutants (**Supplementary Fig. 12b**), and consistent with the marked reductions in the relative number of floor plate cells in this mutant as determined by whole embryo sc-RNA-seq (**Fig. 3d**). Turning to the less expected observation of increased ChP, we found the lateral ventricle as well as the 4th ventricle displayed a disturbed *Ttr* staining pattern. While the wildtype shows inner and outer *Ttr* signal within the single cell layer, the mutant displayed a ‘double dapi’ layer, indicating a disordered tissue organisation (**Fig. 3f; Supplementary Fig. 13a,b**). Adjusting for the overall smaller size of the *Gli2* KO mutants at E13.5, we quantified *Ttr*-positive cells in the lateral and 4th ventricle relative, finding a proportional increase in the mutant relative to WT (**Supplementary Fig. 13c, Supplementary Table 5**), again consistent with the marked increase in the relative number of ChP cells in this mutant as determined by whole embryo sc-RNA-seq. In summary, we could confirm the expected reduction in floor plate in the mutant, as well as the unexpected increase in roof plate-derived ChP. ”

New: “The utility of pursuing whole embryo sc-RNA-seq was also demonstrated by an unexpected finding of both a depleted and an enriched cell population of roof plate cells in the *Gli2* KO mutant. The “dorsalization” of the neural tube in the absence of SHH signalling is well-described^{2,15,17} and was confirmed in our histological analysis of this line (**Supplementary Fig. 12**). However, the roof plate in *Gli2* KO mice has been described as normal, based on traditional analyses of dorso-ventral patterning by examination of transverse sections of the developing neural tube¹⁵. In contrast, whole embryo sc-RNA-seq uncovered that derivatives of the roof plate also depict changes in composition (a primary finding) and tissue development (based on secondary validation) in the mutant, illustrating how this approach can potentially yield new insight into even well-studied developmental pathways.”

2-5: Main Fig 3g - this figure plots similarity scores calculated between embryos of the same and different genotypes. The resulting scoring matrix appears to be asymmetric (e.g. compare the rows vs the columns associated with the *Scn11a* GOF samples), and could lead to different conclusions depending on how the matrix is evaluated. Can the authors explain how the similarity metric calculation can return different values between the same pairs of samples, and which direction is being used for their particular claims (e.g. rows or columns), and why? Same question for Supplemental Figs 10a and 10f. In all of these types of heatmaps, the authors should validate their assertions of genotype similarity scores (both the *Scn11a* and the *Atpv0a2/Gorab* claims) by performing hierarchical clustering of rows vs columns and confirming similarity trends in the resulting dendrograms. In present form, it doesn't look like these heatmaps have been subjected to clustering (if they have, details and dendrograms should be provided).

The similarity score matrix is calculated based on the k-NN graph similar to the lochNESS analysis, but instead of focusing on just one mutant and wild type, the similarity score is calculated for all mutants and wild types within a background strain. A further description of the calculation was included in the **Methods** section (lines 918-930 of original manuscript), repeated below:

We can extend the lochNESS analysis, which is computed on each mutant and its corresponding wildtype mice, to compute “similarity scores” between all pairs of individual embryos from the same background strain. We consider all embryos in the same background in a main trajectory as a dataset. For each dataset, we take define a “similarity score” between $cell_n$ and $embryo_j$ as:

$$similarity\ score_{cell_n, embryo_j} = \frac{\# of\ cells\ from\ embryo_i\ in\ kNNs\ of\ cell_n}{k} / \frac{\# of\ cells\ from\ embryo_j\ in\ dataset}{N}$$

Where N is the total number of cells in the dataset and $k = \frac{\sqrt{N}}{2}$. We take the mean of the similarity scores across all cells in the same embryo, resulting in an embryo similarity score matrix where entries are:

$$similarity\ score_{embryo_i, embryo_j} = \frac{1}{n_i} \sum_{n=1}^{n_i} similarity\ score_{cell_n, embryo_j}$$

Where n_i is the number of cells in $embryo_i$.

To summarise, for each cell, we calculated a score for each genotype, resulting in a cell-by-genotype matrix. We then calculated the average of these scores in each genotype, resulting in a genotype-by-genotype matrix which is visualised as in the form of heat maps as in **Fig. 3** and **Supplementary Fig. 15** (former **Supplementary Fig. 10**).

The asymmetry of the matrix is expected because the nearest neighbour search is not symmetrical (A being a nearest neighbour of B does not guarantee that B is a neighbour of A). In the similarity score matrix M , the entry M_{ij} is the average enrichment of $genotype_j$ in cells from $genotype_i$; the entry M_{ji} is the average enrichment of $genotype_i$ in cells from $genotype_j$.

The rows and columns of the matrices shown in **Fig. 3** and former **Supplementary Fig. 10** were not clustered. Instead, the rows and clusters in the matrices are organised by the embryo id, so that embryos from the same genotype are grouped together. Due to the asymmetry of the matrix, when the rows and columns of the matrices are clustered, the row and column orders do not match and make interpretation difficult. Here we include another version of the heatmaps in **Fig. 3** and **Supplementary Fig. 10** where the rows are hierarchically clustered and columns are reordered to match the row orders (**Revision Fig. 4**). We can arrive at similar conclusions as our previous version, that the *Scn11a* GOF mutant is the most distinct in nearly all trajectories and the *Atp6v0a2* KO, *Atp6v0a2* R755Q and *Gorab* KO mutants are similar in some but not all trajectories.

Revision Figure 4. Similarity score heatmaps of C57BL/6 mutants with hierarchical clustering on rows. **a**, Heatmap showing similarity scores between individual C57BL/6 embryos in the mesenchymal trajectory. Rows are labelled by embryo id and genotype and ordered by hierarchical clustering results. Columns are reordered following the order of rows. **b**, Heatmap showing similarity scores between C57BL/6 genotypes in the mesenchymal trajectory. Rows are labelled by genotype and ordered by hierarchical clustering results. Columns are reordered following the order of rows. **c**, Heatmaps showing similarity scores between C57BL/6 genotypes in selected main trajectories. Rows are labelled by genotype and ordered by hierarchical clustering results. Columns are reordered following the order of rows.

2-6: The analysis of phenotypic similarities between different gene KOs (the topic of manuscript pages 16-17) does not mention the striking similarities at the level cell type abundances for Brain Endothelial, Aorta, Endocardium that were already noted in Fig 2a. If the possibility of artifacts in Fig 2a can be indeed ruled out, it might be worth emphasizing this point again, as it could reinforce the trends being asserted by the authors' kNN-based approach.

Thank you for the suggestion. As mentioned in response to an earlier point 2-1, we explicitly checked for possible confounding factors. More importantly, in response to an earlier point 2-2, we have updated the analysis related to **Fig. 2a** by increasing the number of control samples in the comparisons (i.e. using all wildtype samples instead of strain specific wildtype samples as control). The cell type abundance change similarities in the three sub-trajectories highlighted by the reviewer, Brain endothelial trajectory, Aorta-Gonad-Mesonephros trajectory, and Endocardium trajectory have disappeared in this updated version.

Referee #3 (Remarks to the Author):

3-0: In this manuscript, Huang, Henck, Qiu and colleagues present a mouse mutant cell atlas, MMCA, a collection of just over 1.6 million cell nuclei profiles from a combination of 101 mouse embryos at E13.5 across 22 mouse mutants and four wildtype genetic backgrounds. Combined with the previous work profiling wildtype mouse embryos across E9.5-E13.5, the authors perform a characterization of the relative staging and cell type trajectories present in the MMCA.

A large-scale enrichment analysis is performed to understand the relative enrichment and depletion of cells belonging to trajectories in accordance with the mutant or wildtype condition. Then, the authors extract a global differential abundance scoring, named LochNESS, for which they are able to characterise local differences in cell abundance according to the experimental condition (e.g. wildtype versus mutant). Using this scoring, the authors assess the similarity and differences of mutants in terms of their local cell distributions, identifying instances of distinct and importantly similar cell abundance phenotypes across different mutants. Some further examination and independent validation of abundance changes are made, particularly for the *Ttc21b* KO, and an elegant followup understanding the effect of the *Sox9* regulatory inversion mutant.

Overall, I think this manuscript is quite promising and represents a viable path towards larger scale cellular phenotyping of multiple genetic perturbations. However, I have some key concerns regarding the experiment and analysis, detailed in point form below.

We thank Referee #3 for these comments and constructive feedback. In response to the concerns regarding our experimental and computational methods, we have made clarifications in text, added additional analyses comparing our strategies with related methods and integrated external datasets to strengthen our previous analyses. Additionally, we have performed several experimental validation experiments, validating the several of the phenotypes we identified with our computational strategies.

3-1: LochNESS is a score that reflects the relative mixing of cells coming from the mutant and wildtype conditions, and noted by the authors that is similar in nature to the Milo method presented by Dann et al. It appears that LochNESS does not use replication information in the calculation, thereby losing some ability to detect changes in abundance attributed to the experimental conditions rather than variation within conditions. LochNESS is a score returned per cell reflecting local mixing, and therefore is quite similar in nature to the LISI (Local Inverse Simpson Index) described and implemented by DOI 10.1038/s41592-019-0619-0. It would be worthwhile to assess the correlation of LochNESS scores with LISI, especially under varying neighbourhood size parameter choices.

Thank you for this comment. LochNESS uses replicate information inherently. During the k-NN search for each cell, the cells from the same embryo/replicate are excluded from the search, excluding changes that are embryo/replicate specific. Here we show the lochNESS score distributions if we include the cells from the same embryo/replicate from the k-NN search (**Revision Fig. 5**):

Revision Figure 5. Distribution of lochNESS without excluding the cells from the same embryo in the k-NN search. Estimated density graphs of lochNESS shows distribution of lochNESS across all 64 sub-trajectories in each mutant. In the k-NN search for each cell, all other cells, including those from the same embryo as the cell itself, were included.

We agree that there are conceptual similarities between lochNESS and batch mixing quantification metrics like LISI. LISI quantifies the amount of mixing in a cell's neighbourhood by counting the number of batches represented in the neighbourhood. Compared to LISI, lochNESS has several conceptual advantages: 1) lochNESS can easily determine whether the mutant sample is enriched or depleted in an area that is not well mixed using the sign of the value (positive = enrichment, negative = depletion), while LISI can only separate mixed vs. separated (mixed \rightarrow 2, separated \rightarrow 1); 2) lochNESS can be easily extended to comparisons between multiple samples (as in the similarity score analysis), while LISI is relatively restricted to pairwise comparisons; 3) lochNESS considers a dataset specific neighbourhood size and baseline proportions, while LISI ignores such information.

As a direct comparison, we calculated LISI on each mutant with a pooled wildtype reference in PCA space. We calculated LISI with a dynamic perplexity based on the dataset size ($\text{perplexity} = \text{floor}(0.5 * \sqrt{N/3})$, $K = 3 * \text{perplexity}$), similar to our strategy for determining the neighbourhood size for lochNESS. Focusing on the G4 mutants as an example, the results show a correlation between LISI and lochNESS, where LISI values close to 1 correspond to the more extreme positive or negative values of lochNESS as expected (**Revision Fig. 6**).

Revision Figure 6. Correlation between LISI and lochNESS. Scatterplots of LISI and lochNESS shows correlation of the two scores across G4 mutants in major trajectories. The upper limit of lochNESS was capped at 6. The 9 major trajectories with more cells were included.

3-2: How is it confirmed that the embryo samples corresponded to the listed mutants? There is mention of the absence of the distal limb structures in the ZRS distal enhancer mutant, but is this extended to the other 21 mutant types? e.g. lower expression observed of the genes in certain cell types?

Indeed, we sought to ensure that no mix-ups happened throughout the experiment, by using an embryo ID (a specific adapter) in the first round of indexing. Additionally, the remaining nuclei from all G4 mutants that were left over in the experiment were genotyped again, to ensure that the embryos were assigned correctly to the corresponding IDs. Moreover, if we would have had a mix-up between embryos from different backgrounds, this would have probably been shown in our pseudobulk RNA-seq profiles analysis of the individual aggregated transcriptomes (**Fig. 1b**).

In our pseudobulk RNA-seq profiles analysis, we only observe separation due to different backgrounds as we discussed in the original text (lines 135-142), reproduced below:

“Applying principal components analysis (PCA) to ‘pseudobulk’ profiles of the 103 embryos resulted in two roughly clustered groups corresponding to genetic background (**Fig. 1b**). In particular, wildtype and mutant FVB embryos clustered separately from C57BL/6J, G4, and BALB/C embryos. However, embryos corresponding to individual mutants did not cluster separately, suggesting that none were affected with severe, global aberrations and highlighting the inadequacy of bulk RNA-seq for detecting mutant-specific effects. A single outlier embryo (#104) was aberrant with respect to cell recovery (n = 1,047) as well as appearance (**Supplementary Fig. 1**).”

3-3: What is the rationale for the mutations on the various genetic backgrounds? e.g. why is Fat1 TAD KO performed on a G4 background and not on the other three, some explanation or discussion of this would be helpful.

Thank you for the question. This is due to the different institutes and labs by which the embryos were contributed (**Supplementary Table 1**). The mutant lines are established on a specific background (C57BL/6, G4, FVB or BALB/C), the respective institute has used for many years. For example, all G4 derived mutants come from Berlin, whereas all FVB derived mutants come from Berkeley. In order to account for potential mouse strain derived differences, we added 4 wildtype replicates for each strain which enabled us to account for strain-specific differences in cell composition and/or gene expression, which are known to exist but not well-quantified prior to the current study.

3-4: Further to the above, it would be worthwhile to identify the background/perturbation interaction effect for at least some of the perturbations and background combinations, i.e. how consistent are the cell abundance and profile phenotypes when set against a different background.

Thank you for pointing this out. To be clear, first, we observed a significant difference of cell-type proportions across backgrounds (*i.e.* mouse strains) (**Supplementary Fig. 5c**). Thus, we performed all the comparisons on samples only from the same background. This decision is also supported by the transcriptional stratification among different backgrounds that we observed through the pseudobulk analysis (**Fig. 1b**). Moreover, as the reviewer suggested, we repeated the cell-composition analysis as shown in the previous **Fig. 2a**, except comparing each mutant type to the wildtype mice from different backgrounds (after randomly downsampling the wildtype to 4 samples in order for matching the sample size). The updated heatmap is provided below (**Revision Fig. 7**). As predicted, it shows obvious “block-like” structures, which correspond to the effect driven by different backgrounds rather than genotypes.

Revision Figure 7. We repeat the cell-composition analysis as shown in the previous **Fig. 2a**, except comparing each mutant type to the wildtype mice from different backgrounds (after randomly downsampling the wildtype to 4 samples in order for matching the sample size). Of note, in the revision, we have optimised our strategy by extending the size of control

samples, not only including wildtype but also other mutant types from the same strain ($n = \sim 30$ for C57BL/6, $n = \sim 35$ for FVB, $n = \sim 20$ for G4).

Of note, to improve statistical power, we have optimised our strategy by extending the size of control samples, not only including wildtype but also other mutant types from the same strain ($n = \sim 30$ for C57BL/6, $n = \sim 35$ for FVB, $n = \sim 20$ for G4). Please see more details in our response to point 3-5 below.

3-5: Observing "Block-like" structures along the genetic background in Figure 2a are concerning, especially since these are observed not only for B6 but also to some extent for FVB in the MHB and enteric neuron trajectory, and for G4 in auditory epithelial trajectory. Without performing additional perturbations against other backgrounds (e.g. a *Ttc21b* KO in FVB or G4 and not only in B6), it's difficult to disentangle this confounding. It may also be that strain effect could be masking some real biological enrichment or depletion of trajectories. I would suggest a follow-up check with a subset of perturbation combinations against different backgrounds as an experimental validation.

Thank you for pointing this out. First, we would like to clarify that we observed a significant difference of cell-type proportions across backgrounds (*i.e.* mouse strains) (**Supplementary Fig. 5c**). Thus, we performed comparisons between different genotypes on samples only within the same background. After ruling out the "background" factor, we suspect the "block-like" structure was due to the limited number of control samples ($n = \sim 4$) used in each comparison (*i.e.* calculating log2 transformed ratios of the cell proportions between each mutant type and its corresponding wildtype background); as a consequence, any potential outlier of the control samples will change the result abruptly.

Towards addressing this (and in response to related comment 2-2; see above), we adjusted our approach to control samples, now including in the control set for each comparison, not only the four wildtype embryos of the same strain, but also all other mutant types from the same strain ($n = \sim 30$ for C57BL/6, $n = \sim 35$ for FVB, $n = \sim 20$ for G4). To do this analysis, we assume compositions of cells across cell-types for samples: 1) *from different strains* are following different distributions (**Supplementary Fig. 5**); this is also supported by the transcriptional stratification among different backgrounds that we observed through the pseudobulk analysis (**Fig. 1b**); and 2) *from the same strain*, regardless of its genotype, are roughly following the same distribution. This strategy is also consistent with how we performed the beta-binomial regression analysis, in which we included all the samples within the same strain to gain larger statistical power. In the updated **Fig. 2a**, we show that our new strategy of increasing the sample size in the background group makes the results much more robust. First, most of the key results are retained (**Revision Fig. 2a**). Second, "block-like" structures (as well as several likely artifactual findings highlighted by Reviewer 2 in comment 2-2) are dramatically reduced. Finally, compared to the previous result, we noticed that the effect sizes for most of the FVB mice are decreasing, which is expected (**Revision Fig. 2b**).

Of note, based on our power analysis (see details in our response to Reviewer 1-4 and **Supplementary Fig. 7**), our current sample size of four samples in each group would be sufficient to detect a 25% effect size for those cell types that present at a 1% proportion in wildtype embryos. We realised that the current sample size in each group might be challenging to detect the minor changes in each mutant. As we have highlighted in the updated **Fig. 2a**, those trajectories with proportions $>1\%$ might be much more reliable than those trajectories with proportions $<0.1\%$.

Figure 2a. Heatmap shows \log_2 transformed ratios of the cell proportions between each mutant type (y -axis) and a pooled reference (including both wildtype and other mutants from the strain), across individual sub-trajectories (x -axis). Only those combinations of mutant and sub-trajectory which were nominally significant in the regression analysis are shown (see text and **Methods**; uncorrected p -value < 0.05 ; beta-binomial regression). The number of cells assigned to each developmental trajectory in the overall dataset is shown above the heatmap. The thresholds of sub-trajectories with proportions $<10\%$, $<1\%$, and $<0.1\%$ are highlighted by the red vertical lines.

Revision Figure 2. Comparing the different strategies of calculating the \log_2 transformed ratios of the cell proportions between each mutant type and its corresponding background. **a**, For those 300 significant items (uncorrected p -value < 0.05 ; beta-binomial regression) shown in the heatmap of **Fig. 2a**, we plot the \log_2 ratios calculated by the new strategy (x -axis, using the wildtype from the same strain as the background) vs. the old way (y -axis, using the wildtype plus the other mutants from the same strain as the background). **b**, For a subset of the 68 significant items (uncorrected p -value < 0.05 ; beta-binomial regression) from the FVB strain, we compared the absolute value of \log_2 ratios calculated by the new strategy vs. the old strategy.

3-6: Further to the above, it seems this manuscript is missing a comprehensive comparison between the wildtype genetic backgrounds. The authors could capitalise on the replicated experimental design to statistically assess if variation in cell distribution across trajectories could be explained by background, and if so this should be discussed in terms of implication for understanding the mutant conditions.

We apologise that we were not clearer on this point. As we have explained in our previous responses (mainly for points 3-4 & 3-5), we did in fact detect significant heterogeneity on cell-type proportions across different backgrounds (**Fig. 1b; Supplementary Fig. 5c**), such that it would have been problematic to conduct these analyses between a mutant and an unmatched background wildtype. Instead, we only performed cell composition comparisons (**Fig. 2a**) within the same background (both in the original manuscript as well as in the revision).

However, for lochNESS analysis (**Fig. 3**), we did use a pooled wildtype as a reference. To assess the variability across wildtype cells from different backgrounds in PC space, as discussed in our response to comment 2-3, we calculated lochNESS between all possible strain specific wildtypes to assess the baseline variation (see new **Supplementary Fig. 10d**, reproduced below). For three of the background strains, G4, FVB and BALB/C, the lochNESS distribution is centred around zero as expected, indicating minimal variation in cell distribution between these wild-type strains. Comparisons that include the C57BL/6 wildtype mice show slightly skewed distributions, indicating potential strain specific distributions in C57BL/6 wildtype mice. However, compared to the skews observed in mutant vs wildtype comparisons (**Supplementary Fig. 10a**, lochNESS plotted in range [-2,4]), the between wildtype variability is relatively small (**Supplementary Fig. 10d**, lochNESS plotted in range [-1,1]).

Supplementary Figure 10a,d. **a**, Distribution of lochNESS across all 64 sub-trajectories in each mutant. **d**, Estimated density graphs of lochNESS shows distribution of lochNESS in wildtype comparisons. Each comparison is labelled by the strain treated as the 'mutant', followed by the strain treated as the reference (i.e. G4 vs. FVB indicates that G4 was treated as the 'mutant' in the comparison).

3-7: Figure 3e-f: Examining differential gene expression among cells that are embedded within a low-dimensional space defined by gene expression can be fraught, given the notion of "double-dipping" (e.g. see DOI

10.1016/j.cels.2019.07.012). From the biological standpoint it may also be that the Gli2 KO reduces the capacity for cells fated towards the floor plate trajectory, therefore displaying no differential expression signal only differential abundance signal. While not something I see as required for this manuscript, an ideal exploration of this would be to incorporate a timecourse component to the assessment of the differential genes along the trajectory for the mutant, compared to the wildtype as captured already e.g. in Cao et al 2019.

Thank you for pointing out the potential flaws in the DE gene identification process. We agree that “double-dipping” could be an issue if we over-cluster. On the main and sub-trajectory levels, we were largely conservative and the DE genes identified have matches with markers of known cell types in literature. As we go into more detailed clustering as in the focused analysis of Gli2 KO in the roof plate and floor plate, we could potentially be “double-dipping”. Therefore, we applied the suggested TN-test framework to check if we get consistent p-values under a traditional t-test and the TN-test analysis. In the TN-test analysis, we randomly split our dataset into two partitions (**Revision Fig. 8a**). We processed the dataset with the standard *scanpy* pipeline and clustered the dataset into 3 clusters (leiden clustering resolution=0.3, **Revision Fig. 8b**). We learned the separating hyperplanes from the first partition and applied the model to the second partition to transfer the cluster labels, with visually similar boundaries as shown on the UMAP plots (**Revision Fig. 8c**). The t-test and TN-test results were fairly similar, identifying genes with consistently low p-values across both methods (**Revision Fig. 8d**). Specifically, the gene noted in the manuscript, *Ttr*, received p-values close to zero under both the t-test and TN-test.

Revision Figure 8. TN-test analysis on Gli2 KO in the floor and roof plates. **a**, UMAP showing the cells split into two random partitions. **b**, UMAP showing the cells in partition 1 clustered with the leiden algorithm (resolution=0.3). **c**, UMAP showing the cells in partition 2 assigned a cluster label after hyperplanes are learned from the clustering results in panel **b**. **d**, Barplots showing the negative of the log transformed p-values under a t-test and a TN-test in each pairwise cluster comparison. Top 20 genes with lowest t-test p-values are shown. P-values are capped at a minimum of 2.2e-308.

In response to previous reviewer responses 1-7 and 2-4, we have further discussed the identity of the roof plate sub-clusters and followed up with validation experiments. Based on computational analyses, we inferred that the mutant enriched cluster was the choroid plexus, a roof plate-derived cell type. Based on the validation experiments, we believe the Gli2 KO shows slightly lower intensity of *Ttr* expression in the choroid plexus regions, but more cells proportionally expressing *Ttr*, in line with lochNESS results and the suggestion that we are observing mostly differential abundance signals.

Although we were not able to perform additional experiments to include a time course in our dataset, we can incorporate other datasets with a time component in our analysis. In our existing analysis, we have incorporated the E11.5-E13.5 portion of the MOCA dataset when investigating the *Scn11a* GOF mutant and showed that *Scn11a* GOF mutant are more similar to earlier cells in MOCA than wildtype cells (lines 442-450 of original manuscript):

“To investigate this further, we co-embedded *Scn11a* GOF mutant cells with pooled wildtype cells and MOCA cells from the neural tube trajectory. While wildtype cells were distributed near E13.5 cells from MOCA, the *Scn11a* GOF cells were embedded closer to cells from earlier developmental timepoints (**Supplementary Fig. 15d**). As a more systematic approach, we calculated a “time score” for each cell from the MMCA dataset by taking the k-NNs of each MMCA cell in the MOCA dataset and calculating the average of the developmental time of the MOCA cells. The relative time score distributions of *Scn11a* GOF cells and wildtype cells suggest that *Scn11a* GOF cells are significantly delayed in all major trajectories examined (single sided student’s t-test, raw p-value < 0.01; **Supplementary Fig. 15e**).”

Another dataset with a time component is Stereo-seq²⁹, in which sagittal slices are taken from E9.5-E16.5 mice. As discussed in previous responses 1-7, we mapped the three clusters we identify in **Supplementary Fig. 11a** (formerly **Supplementary Fig. 8d**) to recently published spatially-resolved wildtype mouse embryo transcriptomics data (E13.5) using the ML-based Tangram algorithm (**Supplementary Fig. 11e**). We can extend this analysis and map these clusters to neighbouring timepoints (E11.5-E15.5) to infer the physical locations of the potential ancestor and derivative cells of these clusters (**Revision Fig. 9**). In general, Tangram-mapping to these additional Stereo-seq timepoints support our revised annotations of these clusters (cluster 1: floor plate; cluster 2: cortical hem of lateral choroid plexus; cluster 3: lateral and 4th ventricle choroid plexus).

In a parallel project, we are attempting to generate a high-quality reference for the wild-type mouse that has high temporal resolution throughout embryogenesis. We feel that this will be an essential first step for knowing which timepoints to sample for any given mutant, in order to maximise the odds of isolating the right subset of timepoints for understanding any particular mutant/phenotypic aspect in more detail, particularly as it will be impractical to recreate the entire high-resolution time-series on every mutant.

Revision Figure 9. (Left column) Sagittal mouse slices from E11.5-E15.5 coloured by annotation provided by MOSTA data website and (Right columns) Tangram inferred mapping probability of each cluster shown on the right of panel **Supplementary Fig. 11a**. The colour scale is set from 1st percentile to 99th percentile.

3-8: The similarity score between various genotypes to determine mutation-specific and mutant-shared is not clearly described and hard to follow. It would be good for authors to include clear description in the Methods for this score calculation, and what it represents.

We apologise for the ambiguity regarding similarity scores. The similarity score matrix is calculated based on the k-NN graph similar to the lochNESS analysis, but instead of focusing on just one mutant and wild type, the

similarity score is calculated for all mutants and wild types within a background strain. A description of the calculation is included in the **Methods** section (lines 918-930 in the original text), reproduced below:

We can extend the lochNESS analysis, which is computed on each mutant and its corresponding wildtype mice, to compute “similarity scores” between all pairs of individual embryos from the same background strain. We consider all embryos in the same background in a main trajectory as a dataset. For each dataset, we take define a “similarity score” between $cell_n$ and $embryo_j$ as:

$$similarity\ score_{cell_n, embryo_j} = \frac{\# of\ cells\ from\ embryo_i\ in\ kNNs\ of\ cell_n}{k} / \frac{\# of\ cells\ from\ embryo_j\ in\ dataset}{N}$$

Where N is the total number of cells in the dataset and $k = \frac{\sqrt{N}}{2}$. We take the mean of the similarity scores across all cells in the same embryo, resulting in an embryo similarity score matrix where entries are:

$$similarity\ score_{embryo_i, embryo_j} = \frac{1}{n_i} \sum_{n=1}^{n_i} similarity\ score_{cell_n, embryo_i}$$

Where n_i is the number of cells in $embryo_i$.

To summarise, for each cell, we calculated a score for each genotype, resulting in a cell-by-genotype matrix. We then calculated the average of these scores in each genotype, resulting in a genotype-by-genotype matrix which is visualised as in the form of heat maps as in **Fig. 3** and **Supplementary Fig. 15** (former **Supplementary Fig. 10**).

3-9: Overall how sure are authors that cells are accurately assigned to the experimental condition (mutant or wildtype)? What quality metrics are used in this case and how many “cells” are found to be ambiguous/misassigned and therefore removed?

Relating to our discussion in response to Comment 3-2, the first round of indexing is to specifically label all cells from the respective embryo (embryo ID). Single embryo cells are labelled, before they are pooled and split for the next round of indexing to achieve the complexity of labelling needed for library construction. Therefore, a mix-up between the embryo IDs is highly unlikely. To filter out reads that are likely noise, demultiplexed reads were filtered based on the RT index and ligation index ($ED < 2$, including insertions and deletions) (see **Methods** sub-section: “Processing of sequencing reads”) as well as cells were filtered out, which were poorly assigned to embryo IDs (see **Methods** sub-section: “Whole mouse embryo analysis”). We confirmed the correct assignment of embryos to ID using pseudo-bulk analysis of the mouse backgrounds. In addition, the high purity (zero mis-assigned cells) of sci-RNA-seq has been validated in a previous sci-RNA-seq experiments; in these experiments, human and mouse cells were barcoded in different wells for reverse transcription and the species of all recovered cells match with their expected barcodes⁵⁰.

3-10: It would be highly desirable, in my opinion, to integrate the dissociated data generated in this study to the spatially-resolved wildtype mouse data captured by the sci-Space (10.1126/science.abb9536 from this manuscript's senior authors' labs) and stereo-seq (10.1016/j.cell.2022.04.003) and examine the relative enrichment or depletion of mutants in terms of the physical locations in the mouse embryo.

This is a great suggestion. In several parts of the manuscript, we now perform spatial mapping using Tangram, integrating Stereo-seq data as suggested. The detailed results of such spatial mapping have been discussed in previous responses to comments 1-10, 1-11 and 2-4. Briefly, the analyses helped us to: 1) identify the physical location of lateral plate & intermediate mesoderm (the previous “intermediate mesoderm”) subclusters of enriched or depleted in the *Atp6v0a2* KO, *Atp6v0a2* R755Q and *Gorab* KO mutants, 2) confirm that the ‘branch’ of cells in the mesenchyme trajectory specific to Sox9 Reg. INV mutant map to the brain/spinal cord, and 3) the two roof plate clusters with bimodal patterns in the Gli2 KO were mapped to neighbouring locations in the

developing brain, resolving their annotation and guiding our validation experiments. We added a description on our spatial mapping analysis to the **Methods** section:

Spatial mapping with Tangram

We computationally map our dataset onto a spatially resolved transcriptomics dataset, the mouse organogenesis spatiotemporal transcriptomics atlas (MOSTA) generated with Stereo-seq²⁹. The atlas has a total of 53 sagittal sections from C57BL/6 mouse embryos at from E9.5 to E16.5 in 1 day intervals, and we obtained one section from the most relevant E13.5 data (E13.5_E1S1.MOSTA.h5ad) from the data sharing website associated with the manuscript: <https://db.cngb.org/stomics/mosta/download/>. To map the cells for each single cell cluster on the spatially resolved transcriptomics dataset, we used a machine learning-based method called Tangram⁵¹. Briefly, Tangram is a computational tool that uses a Bayesian approach to infer the spatial locations of cells in a single-cell transcriptomics dataset based on their transcriptomic profiles and the spatial patterns of gene expression in the spatially resolved dataset. The relevant subset of the MMCA data was preprocessed in *Scanpy*, but the metadata was inherited from the results generated in the “Cell clustering and annotation” section above. We used Tangram with default parameters to estimate the spatial coordinates of cells from each cluster in the single cell dataset and visualised results on the coordinates provided by MOSTA. We trained the Tangram model in ‘gpu’ mode using a NVIDIA A100 GPU. Overall, Tangram provided a powerful method for mapping the cells from the single cell RNA-seq dataset onto MOSTA, enabling us to infer the spatial locations of different cell clusters of interest within the tissue.

Furthermore, in previous response 1-11, we integrated our dataset with sci-space to further investigate a group of ‘neural-like’ mesenchyme cells, with results that are complementary/supportive of the Stereo-seq analysis referenced above. We added a description on our integration analysis with sci-space in the **Methods** section:

Integration and spatial mapping with sci-space data

We integrated our dataset with a spatial transcriptomics dataset on mid-gestational mice (E14.5), based on the sci-space method⁴⁸, in which a subset of transcriptionally profiled nuclei have known physical locations in sagittal sections within which they were mapped prior to sc-RNA-seq. We used anchor-based integration as implemented by Seurat for a co-embedding of a subset of MMCA and sci-space. For cells in the subset of MMCA, we find the nearest neighbour in sci-space data in the integrated co-embedding, and plot the location of the neighbouring sci-space cell where it is known.

Minor

3-11: Figure 1d - more useful ordering boxplots by median cell number, typo Dmrt1 "boudary" which appears throughout manuscript and web app

Thank you for pointing this out. We have updated **Fig. 1d** and fixed the typo.

Figure 1d. The number of cells profiled per embryo for each strain. The centre lines show the medians; the box limits indicate the 25th and 75th percentiles; the replicates are represented by the dots. The genotypes are listed by the median cell number in ascending order.

3-12: Figure 1e and elsewhere - 3D UMAP is not particularly visual here, either only show 2D or include grids as in Figure 1b

Thanks for the suggestion. We have added grids in all the 3D UMAPs shown in the figures.

3-13: Supp Figure 5a - add a statistical measure of departure in the cell type distribution, e.g. chi-square test of the perturbation condition and it's corresponding genetic background WT.

The adjusted p-values by Chi-squared test on cell compositions for individual mutant type and its corresponding genetic background wildtype have been added in **Supplementary Fig. 5a**.

Supplementary Figure 5a. Cell composition across 13 major trajectories of embryos from different wildtype or mutant strains. Cells from all replicates for each strain were pooled for this visualisation. The adjusted p-value by Chi-squared test on cell compositions for individual mutant type and its corresponding genetic background wildtype has been added above.

3-14: Figure 2a - barplot on top more useful if presented on a log-scale.

Thanks. We have updated the barplot to log2-scale on the y-axis.

3-15: Figure 4c - axes needed

Thanks. We have added axes to this and other UMAP embeddings.

References:

1. Despang, A. *et al.* Functional dissection of the Sox9-Kcnj2 locus identifies nonessential and instructive roles of TAD architecture. *Nat. Genet.* **51**, 1263–1271 (2019).
2. Stottmann, R. W., Tran, P. V., Turbe-Doan, A. & Beier, D. R. Ttc21b is required to restrict sonic hedgehog activity in the developing mouse forebrain. *Dev. Biol.* **335**, 166–178 (2009).
3. Yadav, N. *et al.* Specific protein methylation defects and gene expression perturbations in coactivator-associated arginine methyltransferase 1-deficient mice. *Proc. Natl. Acad. Sci. U. S. A.* **100**, 6464–6468 (2003).
4. Mo, R. *et al.* Specific and redundant functions of Gli2 and Gli3 zinc finger genes in skeletal patterning and development. *Development* **124**, 113–123 (1997).
5. Leipold, E. *et al.* A de novo gain-of-function mutation in SCN11A causes loss of pain perception. *Nature Genetics* vol. 45 1399–1404 Preprint at <https://doi.org/10.1038/ng.2767> (2013).
6. Schwabe, G. C. *et al.* Ror2 knockout mouse as a model for the developmental pathology of autosomal recessive Robinow syndrome. *Dev. Dyn.* **229**, 400–410 (2004).
7. Chan, W. L. *et al.* Impaired proteoglycan glycosylation, elevated TGF- β signaling, and abnormal osteoblast differentiation as the basis for bone fragility in a mouse model for gerodermia osteodysplastica. *PLOS Genetics* vol. 14 e1007242 Preprint at <https://doi.org/10.1371/journal.pgen.1007242> (2018).
8. Fischer, B. *et al.* Further characterization of ATP6V0A2-related autosomal recessive cutis laxa. *Hum. Genet.* **131**, 1761–1773 (2012).
9. Ringel, A. R. *et al.* Promoter Repression and 3D-Restructuring Resolves Divergent Developmental Gene Expression in TADs. *SSRN Electronic Journal* Preprint at <https://doi.org/10.2139/ssrn.3947354>.
10. Rajderkar, S. *et al.* Topologically Associating Domain Boundaries are Commonly Required for Normal Genome Function. Preprint at <https://doi.org/10.1101/2021.05.06.443037>.
11. Kvon, E. Z. *et al.* Progressive Loss of Function in a Limb Enhancer during Snake Evolution. *Cell* vol. 167 633–642.e11 Preprint at <https://doi.org/10.1016/j.cell.2016.09.028> (2016).
12. Tran, P. V. *et al.* THM1 negatively modulates mouse sonic hedgehog signal transduction and affects retrograde intraflagellar transport in cilia. *Nat. Genet.* **40**, 403–410 (2008).

13. Ben-Yosef, T., Asia Batsir, N., Ali Nasser, T. & Ehrenberg, M. Retinal dystrophy as part of TTC21B-associated ciliopathy. *Ophthalmic Genet.* **42**, 329–333 (2021).
14. Kvon, E. Z. *et al.* Progressive Loss of Function in a Limb Enhancer during Snake Evolution. *Cell* **167**, 633–642.e11 (2016).
15. Ding, Q. *et al.* Diminished Sonic hedgehog signaling and lack of floor plate differentiation in Gli2 mutant mice. *Development* **125**, 2533–2543 (1998).
16. Jacob, J. & Briscoe, J. Gli proteins and the control of spinal-cord patterning. *EMBO Rep.* **4**, 761–765 (2003).
17. Matise, M. P., Epstein, D. J., Park, H. L., Platt, K. A. & Joyner, A. L. Gli2 is required for induction of floor plate and adjacent cells, but not most ventral neurons in the mouse central nervous system. *Development* **125**, 2759–2770 (1998).
18. Kietzman, H. W., Everson, J. L., Sulik, K. K. & Lipinski, R. J. The Teratogenic Effects of Prenatal Ethanol Exposure Are Exacerbated by Sonic Hedgehog or Gli2 Haploinsufficiency in the Mouse. *PLoS One* **9**, e89448 (2014).
19. Herbert, J. *et al.* Transthyretin: a choroid plexus-specific transport protein in human brain. The 1986 S. Weir Mitchell award. *Neurology* **36**, 900–911 (1986).
20. Roof plate mediated morphogenesis of the forebrain: New players join the game. *Dev. Biol.* **413**, 145–152 (2016).
21. Patterning of the Dorsal Telencephalon and Cerebral Cortex by a Roof Plate-Lhx2 Pathway. *Neuron* **32**, 591–604 (2001).
22. Dani, N. *et al.* A cellular and spatial map of the choroid plexus across brain ventricles and ages. *Cell* **184**, 3056–3074.e21 (2021).
23. Cohen, M. *et al.* Ptch1 and Gli regulate Shh signalling dynamics via multiple mechanisms. *Nat. Commun.* **6**, 1–12 (2015).
24. Iulianella, A., Sakai, D., Kurosaka, H. & Trainor, P. A. Ventral neural patterning in the absence of a Shh activity gradient from the floorplate. *Dev. Dyn.* **247**, 170–184 (2018).
25. Persson, M. *et al.* Dorsal-ventral patterning of the spinal cord requires Gli3 transcriptional repressor activity. *Genes Dev.* **16**, 2865–2878 (2002).

26. Broom, E. R., Gilthorpe, J. D., Butts, T., Campo-Paysaa, F. & Wingate, R. J. T. The roof plate boundary is a bi-directional organiser of dorsal neural tube and choroid plexus development. *Development* **139**, 4261–4270 (2012).
27. Grove, E. A., Tole, S., Limon, J., Yip, L. & Ragsdale, C. W. The hem of the embryonic cerebral cortex is defined by the expression of multiple Wnt genes and is compromised in Gli3-deficient mice. *Development* **125**, 2315–2325 (1998).
28. Matise, M. P., Epstein, D. J., Park, H. L., Platt, K. A. & Joyner, A. L. Gli2 is required for induction of floor plate and adjacent cells, but not most ventral neurons in the mouse central nervous system. *Development* **125**, 2759–2770 (1998).
29. Spatiotemporal transcriptomic atlas of mouse organogenesis using DNA nanoball-patterned arrays. *Cell* **185**, 1777–1792.e21 (2022).
30. Zeidler, M. *et al.* NOCICEPTR: Gene and microRNA Signatures and Their Trajectories Characterizing Human iPSC-Derived Nociceptor Maturation. *Adv. Sci.* **8**, e2102354 (2021).
31. Branco, M. A., Dias, T. P., Cabral, J. M. S., Pinto-do-Ó, P. & Diogo, M. M. Human multilineage pro-epicardium/foregut organoids support the development of an epicardium/myocardium organoid. *Nat. Commun.* **13**, 1–18 (2022).
32. Lotto, J. *et al.* Single-Cell Transcriptomics Reveals Early Emergence of Liver Parenchymal and Non-parenchymal Cell Lineages. *Cell* **183**, 702–716.e14 (2020).
33. Shimada, T., Ross, A. C., Muccio, D. D., Brouillette, W. J. & Shealy, Y. F. Regulation of hepatic lecithin:retinol acyltransferase activity by retinoic acid receptor-selective retinoids. *Arch. Biochem. Biophys.* **344**, 220–227 (1997).
34. Arora, R., Metzger, R. J. & Papaioannou, V. E. Multiple roles and interactions of Tbx4 and Tbx5 in development of the respiratory system. *PLoS Genet.* **8**, e1002866 (2012).
35. Aros, C. J., Pantoja, C. J. & Gomperts, B. N. Wnt signaling in lung development, regeneration, and disease progression. *Communications Biology* **4**, 1–13 (2021).
36. Pabst, O., Zweigerdt, R. & Arnold, H. H. Targeted disruption of the homeobox transcription factor Nkx2-3 in mice results in postnatal lethality and abnormal development of small intestine and spleen. *Development* **126**, 2215–2225 (1999).

37. Milewicz, D. M. *et al.* De novo ACTA2 mutation causes a novel syndrome of multisystemic smooth muscle dysfunction. *Am. J. Med. Genet. A* **152A**, 2437–2443 (2010).
38. Abrams, J. *et al.* Graded effects of unregulated smooth muscle myosin on intestinal architecture, intestinal motility and vascular function in zebrafish. *Dis. Model. Mech.* **9**, 529–540 (2016).
39. Halim, D. *et al.* Loss-of-Function Variants in MYLK Cause Recessive Megacystis Microcolon Intestinal Hypoperistalsis Syndrome. *Am. J. Hum. Genet.* **101**, 123–129 (2017).
40. Haworth, K., Samuel, L., Black, S., Kirilenko, P. & Latinkic, B. Liver Specification in the Absence of Cardiac Differentiation Revealed by Differential Sensitivity to Wnt/ β Catenin Pathway Activation. *Front. Physiol.* **10**, 155 (2019).
41. Mai, H. *et al.* Whole mouse body histology using standard IgG antibodies. Preprint at <https://doi.org/10.1101/2023.02.17.528921>.
42. Borchering, N. *et al.* Mapping the immune environment in clear cell renal carcinoma by single-cell genomics. *Commun Biol* **4**, 122 (2021).
43. Liberzon, A. *et al.* The Molecular Signatures Database (MSigDB) hallmark gene set collection. *Cell Syst* **1**, 417–425 (2015).
44. Coricor, G. & Serra, R. TGF- β regulates phosphorylation and stabilization of Sox9 protein in chondrocytes through p38 and Smad dependent mechanisms. *Sci. Rep.* **6**, 38616 (2016).
45. Haller, R. *et al.* Notch1 signaling regulates chondrogenic lineage determination through Sox9 activation. *Cell Death Differ.* **19**, 461–469 (2012).
46. Akiyama, H. *et al.* Interactions between Sox9 and beta-catenin control chondrocyte differentiation. *Genes Dev.* **18**, 1072–1087 (2004).
47. Hino, K. *et al.* Master regulator for chondrogenesis, Sox9, regulates transcriptional activation of the endoplasmic reticulum stress transducer BBF2H7/CREB3L2 in chondrocytes. *J. Biol. Chem.* **289**, 13810–13820 (2014).
48. Srivatsan, S. R. *et al.* Embryo-scale, single-cell spatial transcriptomics. *Science* **373**, 111–117 (2021).
49. Hernández, R. *et al.* Differentiation of Human Mesenchymal Stem Cells towards Neuronal Lineage: Clinical Trials in Nervous System Disorders. *Biomol. Ther.* **28**, 34–44 (2020).
50. Cao, J. *et al.* Comprehensive single-cell transcriptional profiling of a multicellular organism. *Science* **357**,

661–667 (2017).

51. Biancalani, T. *et al.* Deep learning and alignment of spatially-resolved whole transcriptomes of single cells in the mouse brain with Tangram. Preprint at <https://doi.org/10.1101/2020.08.29.272831>.

Reviewer Reports on the First Revision:

Referees' comments:

Referee #1 (Remarks to the Author):

The authors have been attentive to reviewers comments and accordingly the manuscript is much improved. Particularly important improvements are in looking at wild-type variability and calculating some confidence in the data. It is instructive just how many controls need to be looked at to manage some bias in the data. This sort of finding should not be underplayed as it provides real instructive lessons for the user.

Nothing can be done at this point by the selection of mutants, my view has not changed that more thought on the choices would have given more weight to the study.

The authors have cleared up several misunderstandings. However, the authors discussion of Gli2 and the roof plate is still confusing. At the time the author examine the data, there is a roof plate and I believe there is no evidence that the roof plate is abnormal? However, the authors report abnormalities in cell populations that shared a relationship with the roof plate earlier in developmental time - for example. a potential common progenitor gave current (time of analysis) roof plate cell type and choroid plexus cell type. The only conclusion to be drawn then is a possible relationship (not directly addressed by any study in this paper) of Gli2 at an early time, directly or indirectly, impacting roof plate derivatives

Referee #2 (Remarks to the Author):

The authors have addressed my concerns. My biggest worry from the initial version of the manuscript related to the possibility of mouse strain-driven technical artifacts. The authors have taken this concern to heart and performed a number of additional analysis that now convincingly show that while strain of origin **does** make a big difference, this can be addressed by using appropriately considered strain-specific controls. The new Figure 2a, which adopts this change, has reassuringly been cleaned up considerably, and now more clearly illustrates the perturbation-specific changes.

The authors have also performed extensive analyses (e.g. the new supplemental fig 6) to explore how other technical variables contributed to their cell type compositional signals. This analysis has reassured the reader that when strain-of-origin is appropriately controlled for, the likely artifact signals are mitigated substantially.

The authors have made substantial improvements to the description and benchmarking of their "lochness" scores.

The in vivo followups related to the diverging phenotypes of the two roofplate clusters is really interesting and is substantially strengthened with the additional layers of information related to the

two types of choroid plexus.

Referee #3 (Remarks to the Author):

I thank the authors for their careful work in addressing my comments. The amended genotype-trajectory analysis and addition of mapping to spatial reference data has greatly improved the manuscript. I have a few remaining comments and questions that I think should be addressed, below.

- Regarding the lochNESS score, I see that the lochNESS metric is calculated for each cell and excluding cells from the same embryo/replicate. My initial concern is with the comparison against the pooled wildtype as a homogenous group, rather than quantifying any normal variation that can be attributed to sampling cells from embryos, e.g. by calculating lochNESS_L, where L is for each wildtype embryo. As it currently is defined, lochNESS is a single 'point-estimate' for this local enrichment, and it is difficult to understand how variable this estimate is given the cell sampling in the experiment, and therefore how it should be interpreted.

- As an additional question to the above, I'm not so sure on the choice of the pooled wildtype of all 4 background strains as the dataset. Given the broader differences among strains, does it not make sense to calculate lochNESS where the wildtype is matched in strain for the given mutant? Such an approach could lower the bias of different background strains, but may also increase the variance due to lower total cell numbers.

- Comparison with LISI - observing the high concordance of absolute lochNESS score with LISI makes a lot of sense, and I appreciate the added benefit of the sign component in lochNESS which indicates enrichment-depletion as opposed to mixing-notmixing. Describing this relationship among these metrics and distinguishing factors of lochNESS metric would be useful to the readers.

- Updated Figure 2a - it's great to see more robust results here. The key difference is in treating not only wildtype mice as the statistical control group for a given genotype, but the combination of wildtype mice alongside all other genotypes collected for that strain. The observation of less 'block-like' structure in the genotype-trajectory logratio heatmap, and fewer potential artefacts in sub-trajectories is consistent with the intuition that more data in the control group could increase robustness in the results, as you are better able to quantify variation when using all available data. The only caveat of such an analysis is the underlying assumption that the cell type proportions of non-wildtype genotypes are roughly consistent as a whole to that of wildtype. I think that for the current experimental study with a broad range of genotypes and cell type phenotypes, this is a safe enough underlying assumption, and indeed could be explicitly checked via a "wildtype vs non-wildtype" comparison with no discernible differences. However, for any additional studies that are not so varied in terms of the non-wildtype genotypes, such a strategy could reduce the power to detect biologically relevant genotype-trajectory relationships. This point could be discussed in the manuscript and would be helpful for readers wanting to replicate such approaches in their own studies.

Author Rebuttals to First Revision:

We thank the referees and the editors for their careful consideration of our revisions. In this document, the second round of comments are shown in blue and our responses in black.

Referee #1 (Remarks to the Author):

1-0 The authors have been attentive to reviewers comments and accordingly the manuscript is much improved. Particularly important improvements are in looking at wild-type variability and calculating some confidence in the data. It is instructive just how many controls need to be looked at to manage some bias in the data. This sort of finding should not be underplayed as it provides real instructive lessons for the user.

We thank the referee for their constructive feedback and comments that absolutely helped us to improve the manuscript. We fully agree that this aspect should not be underplayed, and in the revised **Discussion**, we explicitly emphasise the importance of controls. The relevant text is replicated below.

“We emphasise that the concurrent analysis of many mutants proved essential to the contextualization of particular observations, *i.e.* to understand how specific or non-specific any apparent deviation really was, against a background of dozens of genotypes and over 100 embryos.”

“Our mouse mutant cell atlas (MMCA) has limitations. First, we only profiled 4 replicates per mutant at a single developmental time point. Based on a simulation analysis of the analytical approach that considers cell proportions only, four replicates of each mutant is likely sufficient to detect modest changes in abundant cell types (*e.g.* a 10% change for cell types at 10% abundance) but only large changes in rarer cell types (*e.g.* a 25% change in cell types at 1% abundance; **Supplementary Fig. 7**).”

“Fourth, our results emphasise the importance of a well-matched control; although data from our wildtype embryos could be re-used as control data for future studies of additional mutants, that risks batch effects, and a safer strategy would be to always include a well-matched, “in-line” wild-type control while profiling mutant embryos.”

1-1 Nothing can be done at this point by the selection of mutants, my view has not changed that more thought on the choices would have given more weight to the study.

Thank you for the comment. Hindsight is twenty-twenty, as they say, but looking forward, with all that we have learned through this current study, we are putting careful thought into the selection of coherent sets of mutants for our expanded application of this approach.

1-2 The authors have cleared up several misunderstandings. However, the authors discussion of Gli2 and the roof plate is still confusing. At the time the author examine the data, there is a roof plate and I believe there is no evidence that the roof plate is abnormal? However, the authors report abnormalities in cell populations that shared a relationship with the roof plate earlier in developmental time - for example. a potential common progenitor gave current (time of analysis) roof plate cell type and choroid plexus cell type. The only conclusion to be drawn then is a possible

relationship (not directly addressed by any study in this paper) of Gli2 at an early time, directly or indirectly, impacting roof plate derivatives

Thank you for your insightful comment. We indeed did not examine changes in the roof plate itself but after annotating the roof plate subclusters further in depth, we detected changes in the descendant structures, namely the choroid plexus and the cortical hem. As to which time point these alterations occur due to changes in Gli2 expression is not discernible from our data. To further clarify this point, we've added a brief section into the discussion (additions given blue background):

Previous text (line 717-726): “The utility of pursuing whole embryo sc-RNA-seq was also demonstrated by an unexpected finding of both a depleted and an enriched cell population of roof plate cells in the *Gli2* KO mutant. The “dorsalization” of the neural tube in the absence of SHH signalling is well-described^{28,52,99} and was confirmed in our histological analysis of this line (**Supplementary Fig. 12**). However, the roof plate in *Gli2* KO mice has been described as normal, based on traditional analyses of dorso-ventral patterning by examination of transverse sections of the developing neural tube⁵². In contrast, whole embryo sc-RNA-seq uncovered that derivatives of the roof plate also depict changes in composition (a primary finding) and tissue development (based on secondary validation) in the mutant, illustrating how this approach can potentially yield new insight into even well-studied developmental pathways.”

Now: “The utility of pursuing whole embryo sc-RNA-seq was also demonstrated by an unexpected finding of both a depleted and an enriched cell population of roof plate cells derivatives in the *Gli2* KO mutant. The “dorsalization” of the neural tube in the absence of SHH signalling is well-described^{18,38,53} and was confirmed in our histological analysis of this line (**Supplementary Fig. 12**). However, there have been no described changes in the roof plate or its derivatives to date in *Gli2* KO mice³⁸. In contrast, whole embryo sc-RNA-seq uncovered that derivatives of the roof plate depict changes in composition (a primary finding) and tissue development (based on secondary validation) in the mutant, illustrating how this approach can potentially yield new insight into even well-studied developmental pathways. However, due to our dataset only capturing one timepoint, whether Gli2 misexpression causes the structural change directly in the derivative tissue or earlier during roof plate formation, remains elusive.”

Referee #2 (Remarks to the Author):

2-0 The authors have addressed my concerns. My biggest worry from the initial version of the manuscript related to the possibility of mouse strain-driven technical artifacts. The authors have taken this concern to heart and performed a number of additional analysis that now convincingly show that while strain of origin *does* make a big difference, this can be addressed by using appropriately considered strain-specific controls. The new Figure 2a, which adopts this change, has reassuringly been cleaned up considerably, and now more clearly illustrates the perturbation-specific changes.

The authors have also performed extensive analyses (e.g. the new supplemental fig 6) to explore how other technical variables contributed to their cell type compositional signals. This analysis

has reassured the reader that when strain-of-origin is appropriately controlled for, the likely artifact signals are mitigated substantially.

The authors have made substantial improvements to the description and benchmarking of their "lochness" scores.

The in vivo followups related to the diverging phenotypes of the two roofplate clusters is really interesting and is substantially strengthened with the additional layers of information related to the two types of choroid plexus.

We thank the referee for their constructive feedback and comments that absolutely helped us to improve the manuscript.

Referee #3 (Remarks to the Author):

3-0 I thank the authors for their careful work in addressing my comments. The amended genotype-trajectory analysis and addition of mapping to spatial reference data has greatly improved the manuscript. I have a few remaining comments and questions that I think should be addressed, below.

Thank you for your positive feedback, as well as for your constructive comments which very much helped us to strengthen the manuscript. We have sought to address your remaining comments as outlined below.

3-1 - Regarding the lochNESS score, I see that the lochNESS metric is calculated for each cell and excluding cells from the same embryo/replicate. My initial concern is with the comparison against the pooled wildtype as a homogenous group, rather than quantifying any normal variation that can be attributed to sampling cells from embryos, e.g. by calculating lochNESS_L, where L is for each wildtype embryo. As it currently is defined, lochNESS is a single 'point-estimate' for this local enrichment, and it is difficult to understand how variable this estimate is given the cell sampling in the experiment, and therefore how it should be interpreted.

Thank you for your comment clarifying the question regarding the variability of lochNESS. The current implementation of lochNESS is indeed simply a per cell quantification of local enrichment of mutant cells and does not utilise the individual wildtype embryos to account for variability. This is because lochNESS is calculated on a KNN graph and benefits from having more cells in the calculation. For reference, in the focused analysis of Gli2 KO in the roof plate and floor plate, there were a total of 1753 cells across all mutant embryos (331 cells total, range 67 to 200 cells per embryo) and wildtype embryos (1422 cells, range 28 to 168 cells per embryo). Given the limited number of wildtype cells per embryo for individual cell types, we believe the current implementation was the most suitable for our dataset, but we could imagine other datasets where calculation of lochNESS_L would be more suitable. We have implemented lochNESS_L in our demo script on a demo dataset containing Gli2 KO cells and wildtype cells in the haematopoiesis

trajectory. For this demo, we observe consistently between lochNESS and lochNESS_L (Pearson's $r > 0.5$) except for one wildtype embryo (#44) (**Reviewer Fig. 1**).

Reviewer Figure 1. Assessing the variability of lochNESS on a demo dataset containing *Gli2* KO cells and wildtype cells in the Haematopoiesis trajectory. **a**, UMAP embedding of cells in the haematopoiesis trajectory from the *Gli2* KO mutant, coloured by sub-trajectory. **b**, same as in **a**, but coloured by lochNESS. **c**, heatmap showing the correlation between lochNESS and lochNESS_L, where lochNESS_L is calculated with each wildtype sample L as reference. **d**, same as **a**, but coloured by lochNESS_L.

We have not added this analysis to the paper for reasons of space (although we are happy to if you think that we should). We have added the following notes to the **Methods** section:

“Additionally, if the numbers of cells are sufficient, one set of lochNESS can be calculated for each wildtype sample separately and the variability between samples can be considered.”

3-2 - As an additional question to the above, I'm not so sure on the choice of the pooled wildtype of all 4 background strains as the dataset. Given the broader differences among strains, does it not make sense to calculate lochNESS where the wildtype is matched in strain for the given mutant? Such an approach could lower the bias of different background strains, but may also increase the variance due to lower total cell numbers.

Thank you for bringing up this question. Indeed there was a tradeoff between bias due to differences in background strain (if we use the current version, a pooled wildtype) and bias due to low numbers of cells (if we use only wildtype samples from the matching background strain).

We have tested both versions and eventually elected to show the current version with a pooled wildtype, mostly due to the low number of cells in some of the trajectories of interest (as noted in

our response to 3-1). Overall, we selected an implementation using the pooled wildtype as a homogeneous group for reference based on various factors (the number of genotypes, the number of samples per genotype, the number of cells per sample, the number of cells in the cell type/tissue of interest).

For example, if we compare our current lochNESS implementation for the neural tube sub-trajectories of the *Ttc21b* KO and *Gli2* KO mutants (currently shown in **Fig. 3c**) with lochNESS calculated using only the matching background strain (**Reviewer Fig. 2**). With the matching background strain, we can detect the more severe losses in *Ttc21b* KO retinal neuron trajectory and *Gli2* KO floor plate trajectory, but miss the more subtle changes in the *Gli2* KO roof plate trajectory.

Reviewer Figure 2. Comparison between different lochNESS implementations. **a**, Distribution of lochNESS in the neural tube sub-trajectories of the *Ttc21b* KO and *Gli2* KO mutants, where the reference is a pooled wildtype. Dashed boxes highlight the shifted distributions of the retinal neuron sub-trajectory of the *Ttc21b* KO mutant and the floor plate and roof plate sub-trajectories of the *Gli2* KO mutant. **b**, Distribution of lochNESS in the neural tube sub-trajectories of the *Ttc21b* KO and *Gli2* KO mutants, where the reference is the background matched wildtype.

We have not added this analysis to the paper for reasons of space (although we are happy to if you think that we should). But as the best implementation will vary for every dataset based on these factors, we have added the following note to the **Methods** section.

“Currently we calculate lochNESS using a pooled wildtype combining all 4 background strains to include larger numbers of cells in constructing the KNN graph. If the numbers of cells are sufficient, a wildtype from the matched background strain can be used.”

3-3 - Comparison with LISI - observing the high concordance of absolute lochNESS score with LISI makes a lot of sense, and I appreciate the added benefit of the sign component in lochNESS which indicates enrichment-depletion as opposed to mixing-notmixing. Describing this relationship among these metrics and distinguishing factors of lochNESS metric would be useful to the readers.

Thank you for the comment. We have added the figure comparing lochNESS and LISI as **Supplementary Fig. 10f**, reproduced below.

Supplementary Figure 10f. Scatterplots showing the concordance of lochNESS and LISI of cells from the G4 mutants in various main trajectories. More extreme lochNESS (indicating separation between mutant and wildtype) is associated with LISI scores approaching one (indicating non-mixing).

Since we need to cut down by a very large number of words, we are unfortunately unable to add the description into the main text, but we have added the following description in the **Methods** section:

“LochNESS shares conceptual similarities with batch correcting measurement scores like LISI, which quantifies the amount of mixing in a cell’s neighbourhood by counting the number of batches represented in the neighbourhood. As a direct comparison, we calculated LISI on each mutant with a pooled wildtype reference in PCA space. We calculated LISI with a dynamic perplexity based on the dataset size (perplexity = $\text{floor}(0.5 * \sqrt{N/3})$, $K = 3 * \text{perplexity}$), similar to our strategy for determining the neighbourhood size for lochNESS. Focusing on the G4 mutants as an example, the results show a correlation between LISI and lochNESS, where LISI values close to 1 correspond to the more extreme positive or negative values of lochNESS as expected (**Supplementary Fig. 10f**). LochNESS has several conceptual advantages compared to LISI. First,

lochNESS can easily determine whether the mutant sample is enriched or depleted in an area that is not well mixed using the sign of the value (positive = enrichment, negative = depletion), while LISI can only separate mixed (scores approaching 2) vs. separated (scores approaching 1). Second, lochNESS can be easily extended to comparisons between multiple samples, while LISI is relatively restricted to pairwise comparisons. Third, lochNESS considers a dataset specific neighbourhood size and baseline proportions.”

3-4 - Updated Figure 2a - it's great to see more robust results here. The key difference is in treating not only wildtype mice as the statistical control group for a given genotype, but the combination of wildtype mice alongside all other genotypes collected for that strain. The observation of less 'block-like' structure in the genotype-trajectory logratio heatmap, and fewer potential artefacts in sub-trajectories is consistent with the intuition that more data in the control group could increase robustness in the results, as you are better able to quantify variation when using all available data. The only caveat of such an analysis is the underlying assumption that the cell type proportions of non-wildtype genotypes are roughly consistent as a whole to that of wildtype. I think that for the current experimental study with a broad range of genotypes and cell type phenotypes, this is a safe enough underlying assumption, and indeed could be explicitly checked via a "wildtype vs non-wildtype" comparison with no discernible differences. However, for any additional studies that are not so varied in terms of the non-wildtype genotypes, such a strategy could reduce the power to detect biologically relevant genotype-trajectory relationships. This point could be discussed in the manuscript and would be helpful for readers wanting to replicate such approaches in their own studies.

Thank you for your kind note and insightful suggestions. The reviewer is correct and we indeed document some broad differences in cell type composition between WT strains at the level of major trajectories. To evaluate whether these or more subtle differences impact our analyses, we compared wildtype and non-wildtype samples, ignoring the differences of individual mutant types, in each of the three backgrounds (C57BL/6, G4, and FVB). We performed the same analysis as before, and the results are shown in **Reviewer Fig. 3**.

Reviewer Figure 3. Cell type composition changes and significance for wildtype vs non-wildtype comparisons. Each dot represents a comparison between wildtype and non-wildtype in a trajectory in a background strain. The x-axis represents the log2 transformed ratio (non-wildtype vs. wildtype) and the y-axis represents the $-\log_{10}$ transformed FDR (by beta-binomial regression).

The results show that most of the comparisons are not significant, indicating that our assumption is generally reasonable. For those examples which are significant ($\text{fdr} < 0.05$), some of them may be due to the small number of cells (e.g. endocardium trajectory, $n = 1654$ cells, 0.1% of the whole dataset), while others have small \log_2 transformed ratio or their fdr are close to 0.05 (e.g. spinal cord excitatory neuron trajectory, $\text{fdr} = 0.049$).

Furthermore, we added this paragraph in the **Discussion** section.

“It is important to note that our cell-composition analysis, which includes both wildtype and mutant cells from the same strain to generate a pooled reference, assumes that the cell type proportions of non-wildtype genotypes are roughly consistent, at least on the whole, with those of wildtype cells. This assumption may be more problematic in studies of biologically related mutants.”